# Cross-fluctuation phase transitions reveal sampling dynamics in diffusion models

**Sai Niranjan Ramachandran**$^{*\dagger}$     **Manish Krishan Lal**$^{*\dagger}$     **Suvrit Sra**$^{*\dagger}$

## Abstract

We analyse how the sampling dynamics of distributions evolve in score-based diffusion models using *cross-fluctuations*, a centered-moment statistic from statistical physics. Specifically, we show that starting from an unbiased isotropic normal distribution, samples undergo sharp, discrete transitions, eventually forming distinct events of a desired distribution while progressively revealing finer structure. As this process is reversible, these transitions also occur in reverse, where intermediate states progressively merge, tracing a path back to the initial distribution. We demonstrate that these transitions can be detected as discontinuities in $n^{\text{th}}$-order cross-fluctuations. For variance-preserving SDEs, we derive a closed-form for these cross-fluctuations that is efficiently computable for the reverse trajectory. We find that detecting these transitions directly boosts sampling efficiency, accelerates class-conditional and rare-class generation, and improves two zero-shot tasks–image classification and style transfer–without expensive grid search or retraining. We also show that this viewpoint unifies classical coupling and mixing from finite Markov chains with continuous dynamics while extending to stochastic SDEs and non Markovian samplers. Our framework therefore bridges discrete Markov chain theory, phase analysis, and modern generative modeling.

## 1 Introduction

Diffusion models are now a cornerstone of generative systems. They learn to reverse a process that gradually erases structure from data, transforming it into featureless (isotropic) noise. During inference, the model starts with this noise and progressively refines it, adding structure step by step until realistic data samples emerge. [Ho et al., 2020, Sohl-Dickstein et al., 2015]. These models can generate lifelike images, 3D scenes, audio, and even molecular structures [Rombach et al., 2022, Blattmann et al., 2023, Chen et al., 2022b, Corso et al., 2022, 2023, Karnewar et al., 2023, Kynkäänniemi et al., 2024]. They also excel in complex tasks like protein design [Jumper et al., 2021] and zero-shot vision tasks [Li et al., 2023, Meng et al., 2021, Rombach et al., 2022].

Yet, the sampling process remains a black box. Each step blends thousands of values in ways that are difficult to anticipate. Like other generative paradigms [Hutchinson and Mitchell, 2019, Hendrycks et al., 2023, Rudin, 2019, Sharkey et al., 2025], small tweaks in code or hyperparameters can turn a perfect sample into a failure or introduce unintended biases. This unpredictability underscores the need for a clear understanding of how successful and failed outcomes diverge as the sampler operates.

We provide a clear framework by defining a user-specified goal (e.g., a class label, an aesthetic style, or a semantic attribute) as a *desirable event*, which is a set of outcomes that align with the user's objective. As the sampler evolves *backward* in time, it iteratively generates a distribution over possible outcomes at each step. Initially, paths corresponding to different outcomes, whether desirable or undesirable, may appear indistinguishable. However, we demonstrate that at certain critical intermediate steps, these paths undergo *discrete phase transitions*, where they become distinguishable as they separate

---

$^{*}$School of Computation, Information and Technology, Technical University of Munich, Germany
$^{\dagger}$Munich Center for Machine Learning (MCML)

39th Conference on Neural Information Processing Systems (NeurIPS 2025).

into distinct regions of the outcome space. These transitions occur when the sampling process reveals structural differences between outcomes that align with the user's goal (desirable) and those that do not (undesirable), marked by a breakdown of path separability.

**Main contributions**

The main contributions of this paper are the following:

1. **Framework and theory.** We present a rigorous framework using fluctuation theory from statistical physics [Chaikin et al., 1995, Kivelson et al., 2024, Landau and Lifshitz, 1980, Pathria and Beale, 2011] to identify and quantify *discrete phase transitions in the diffusion process* (Section 2.2, Appendices B.1.1 and B.3.6). This framework incorporates user-specified goals as desirable events and shows how their dynamics can be tracked (Section 3). Moreover, we establish a connection between ensemble based techniques in statistical mechanics as well as classical mixing-coupling results for Markov chains [Aldous and Fill, 2002, Levin and Peres, 2017, Saloff-Coste, 1997], using a generalised notion of phase transitions, thereby providing a unified perspective on probability flows (Section B.1.3.2 and Appendices B.1.4 and B.3.6).

2. **Practical toolkit.** We develop a practical toolkit with clear diagnostic criteria for identifying and leveraging these *discrete transitions* (Section 3, Appendices B and C). The proposed Algorithm 1 systematically identifies discrete transitions in cross-fluctuations to characterise the sampling dynamics of desired outcomes. This enables precise intervention to enhance the generation and likelihood of desirable samples, thereby improving overall model performance.

3. **Illustrative applications.** We illustrate the power of the proposed framework and toolkit through several applications: accelerated sampling (Section 4.1), conditional generation (Section 4.2), rare-class coverage (Section 4.3), and improved zero-shot performance in classification (Section 4.4) and style transfer (Section 4.5). Remarkably, a simple fluctuation driven tweak can improve the performance of baseline methods.

The remainder of this paper is organised as follows: background (Section 2); methodology (Section 3); experiments (Section 4); theoretical foundations (Appendix B); implementation details and additional experiments (Appendix C); and limitations and related-work context (Appendix D–E).

## 2 Background

We first review variance-preserving diffusion models and the fluctuation-theoretic statistics that let us detect emergent phases. For detailed notation used throughout the paper, see Appendix A.

### 2.1 Diffusion models in continuous and discrete time

Let $\mathbf{x}_0 \sim p_0$ be a sample from a distribution with support $\mathcal{X} \subseteq \mathbb{R}^d$. The *forward* variance-preserving stochastic differential equation (SDE) of Song et al. [2021] is

$$\mathrm{d}\mathbf{x}_t = -\tfrac{1}{2}\beta(t)\,\mathbf{x}_t\,\mathrm{d}t + \sqrt{\beta(t)}\,\mathrm{d}\mathbf{w}_t, \qquad t \in [0, T], \tag{2.1}$$

where $\beta(t) > 0$ is the noise schedule and $\mathbf{w}_t$ is a standard Wiener process. For linear schedules and sufficiently large $T$ we have $p_T \approx \mathcal{N}(\mathbf{0}, \mathbf{I})$. Removing stochasticity yields the **probability-flow ODE** (PF-ODE) with identical marginals,

$$\frac{\mathrm{d}\mathbf{x}_t}{\mathrm{d}t} = -\tfrac{1}{2}\beta(t)\,\mathbf{x}_t - \beta(t)\,\underbrace{\nabla_{\mathbf{x}} \log p_t(\mathbf{x}_t)}_{\text{score}}. \tag{2.2}$$

A neural network $s_\theta(\mathbf{x}, t)$ learns an approximation of the score; sampling integrates the *reverse* ODE from $t = T$ back to 0. In discrete time, the Denoising Diffusion Probabilistic Model (DDPM) update of Ho et al. [2020] is

$$\mathbf{x}_t = \sqrt{1 - \beta_t}\,\mathbf{x}_{t-1} + \sqrt{\beta_t}\,\varepsilon_t, \qquad \varepsilon_t \sim \mathcal{N}(\mathbf{0}, \mathbf{I}), \quad t = 1, \dots, T. \tag{2.3}$$

Both the continuous and the discrete processes admit the closed-form marginal

$$p_t(\mathbf{x}_t \mid \mathbf{x}_0) = \mathcal{N}\big(J(t)\,\mathbf{x}_0, \big(1 - J^2(t)\big)\mathbf{I}\big), \qquad J(t) = \exp\big(-\tfrac{1}{2}\int_0^t \beta(s)\,\mathrm{d}s\big), \tag{2.4}$$

which separates signal from noise cleanly, where $J(t)$ is known as the signal-attenuation factor.

## 2.2 Measuring statistical distinctiveness

The central insight of our work is the observation that given coarse categories of data e.g, object classes, tracking their *statistical distinctiveness* throughout the diffusion process suffices to pinpoint the *emergence* of structure during generation.



**Figure 1:** The addition of noise causes distinct categories of data to "merge" through the forward diffusion process as statistical properties progressively converge to those of the standard normal distribution.

As Figure 1 shows, the statistical properties of data show a well-defined convergence on noise addition and conversely show divergence as the sampling process progresses. We formalise this notion through the use of fluctuation theory adapted from physics.

### 2.2.1 Basics of fluctuation theory

Fluctuation theory supplies statistics that reveal when trajectories become (in)distinguishable—our operational signature of a phase transition. We present a formal framework for analyzing these fluctuations in vector-valued state spaces, which generalises the concepts from statistical physics and provides a direct connection to modern tools like Centered Kernel Alignment (CKA) (Kornblith et al. [2019]).

**Fluctuations.** Let $(\Omega, \mathcal{F}, P)$ be a probability space. The state of the system is described by a vector-valued random variable $\rho : \Omega \to \mathbb{R}^d$. The fundamental object for our analysis is the $n^{\text{th}}$**-order fluctuation**, a random tensor representing the $n^{\text{th}}$ centered moment at a single point $\omega \in \Omega$:

$$\mathcal{F}_\rho^{(n)}(\omega) := \bigotimes_{k=1}^{n} \big(\rho(\omega) - \mathbb{E}[\rho]\big). \tag{2.5}$$

This is a tensor of rank $n$. For instance, for $n = 2$, this gives $\mathcal{F}_\rho^{(2)}(\omega) = (\rho(\omega) - \mathbb{E}[\rho])(\rho(\omega) - \mathbb{E}[\rho])^\top$, which is a $d \times d$ matrix. Its expectation, $\mathbb{E}[\mathcal{F}_\rho^{(2)}]$, is the familiar **covariance matrix** of $\rho$. We assume these random fluctuation tensors (and their expectations) reside in suitable Hilbert spaces $\mathcal{H}_n$ (for rank-$n$ tensors) equipped with an inner product $\langle \cdot, \cdot \rangle_{\mathcal{H}_n}$. For matrices ($n = 2$), $\mathcal{H}_2$ is typically the space of $d \times d$ matrices with the Frobenius inner product, $\langle A, B \rangle_F = \text{Tr}(A^\top B)$.

**Joint dynamics and conditional expectations.** We consider two disjoint events, $\Omega_1$ and $\Omega_2$. While samples are drawn independently from each, the events themselves are **coupled**: they originate from the same data manifold and evolve under the **same stochastic dynamics**. This coupling justifies the comparison of their respective **conditional expectations of the fluctuation tensor**: $\mathbb{E}_1[\mathcal{F}_\rho^{(n)}] = \mathbb{E}[\mathcal{F}_\rho^{(n)} | \omega \in \Omega_1]$ and $\mathbb{E}_2[\mathcal{F}_\rho^{(n)}] = \mathbb{E}[\mathcal{F}_\rho^{(n)} | \omega \in \Omega_2]$. For $n = 2$, these are the conditional covariance matrices for each event.

**Normalised and unnormalised cross-fluctuations.** We now redefine our key statistics based on these conditional expectations.

1. **Unnormalised cross-fluctuation ($G$):** This measures the alignment of the conditional expected fluctuation tensors of the two events.

$$G_\rho^{(n)}(\Omega_1, \Omega_2) := \langle \mathbb{E}_1[\mathcal{F}_\rho^{(n)}], \mathbb{E}_2[\mathcal{F}_\rho^{(n)}] \rangle_{\mathcal{H}_n}. \tag{2.6}$$

2. **Within event fluctuation ($\widehat{F}$):** This measures the magnitude (or self-alignment) of the conditional expected fluctuation tensor for a single event.

$$\widehat{F}_\rho^{(2n)}(\Omega_i) := \langle \mathbb{E}_k[\mathcal{F}_\rho^{(n)}], \mathbb{E}_k[\mathcal{F}_\rho^{(n)}] \rangle_{\mathcal{H}_n} = \|\mathbb{E}_k[\mathcal{F}_\rho^{(n)}]\|_{\mathcal{H}_n}^2. \tag{2.7}$$

Note the $2n$ in the symbol $\widehat{F}$ is a convention.

3. **Normalised cross-fluctuation ($\mathcal{M}$):** This is the magnitude of the cosine similarity between the conditional expected fluctuation tensors.

$$\mathcal{M}_\rho^{(n)}(\Omega_1, \Omega_2) := \frac{\|G_\rho^{(n)}(\Omega_1, \Omega_2)\|}{\sqrt{\widehat{F}_\rho^{(2n)}(\Omega_1)\widehat{F}_\rho^{(2n)}(\Omega_2)}} = \frac{|\langle \mathbb{E}_1[\mathcal{F}_\rho^{(n)}], \mathbb{E}_2[\mathcal{F}_\rho^{(n)}]\rangle_{\mathcal{H}_n}|}{\|\mathbb{E}_1[\mathcal{F}_\rho^{(n)}]\|_{\mathcal{H}_n}\|\mathbb{E}_2[\mathcal{F}_\rho^{(n)}]\|_{\mathcal{H}_n}}. \quad (2.8)$$

The normalized cross-fluctuation, $\mathcal{M}_\rho^{(n)} \in [0, 1]$, measures the structural similarity between the events' $n$-th order moments. A value of $\mathcal{M}_\rho^{(n)} \approx 1$ indicates that the events' statistical shapes have aligned, a condition we define as a "merge." Conversely, a value significantly less than 1 implies distinguishability. For the important special case of $n = 2$, $\mathcal{M}_\rho^{(2)}$ quantifies the similarity between the conditional covariance matrices ($\Sigma_k$) of the two events and is connected to CKA.

## 3 A practical toolkit for diffusion models

We now present the theoretical construction for our methodology. The main practical outcome is Algorithm 1, an algorithm whose applications are demonstrated in Sections 4.1 to 4.5.

**Embedding in diffusion time.** Let $p_i$ be the marginal at step $i = 0, \ldots, n$ of a forward diffusion process (For simplicity, we focus on the PF-ODE case[3]), with $p_0 = p_{\text{desired}}$, where $p_{\text{desired}}$ represents a constructed distribution that encodes properties of interest, and $p_n = \mathcal{N}(\mathbf{0}, \mathbf{I})$. Choose two disjoint events $\Omega_{1,0}, \Omega_{2,0} \subseteq \text{supp}(p_0)$ and propagate them to $\Omega_{1,i}, \Omega_{2,i}$. As the forward trajectory converges to white noise, the regions coincide at $i = n$ in theory, but due to hyper-contractivity and numerical effects, they often appear to merge much earlier. We thus redefine the operator $\mathcal{M}_\rho^{(n)}$ to capture this *discrete transition* (Appendix B.3.6),

$$\widetilde{\mathcal{M}}_\rho^{(n)}(i) = \begin{cases} \mathcal{M}_\rho^{(n)}(\Omega_{1,i}, \Omega_{2,i}), & d\big(\widehat{F}_\rho^{(2n)}(\Omega_{1,i}), \widehat{F}_\rho^{(2n)}(\Omega_{2,i})\big) > \varepsilon, \\ 1, & \text{otherwise}, \end{cases} \quad (3.1)$$

where $d(\cdot, \cdot)$ is a distance metric which for our case is the $\ell_1$ distance between the traces of $\widehat{F}_\rho^{(n)}(\cdot)$ and $\varepsilon > 0$ is a fixed threshold.[4] The earliest index

$$i^\star = \min\big\{ i : \widetilde{\mathcal{M}}_\rho^{(n)}(i) = 1 \big\}$$

generalises the coupling time of a Markov chain [Aldous and Fill, 2002, Levin and Peres, 2017] (Appendix B.1.4). The operator undergoes a *discrete* phase transition dictated by $\varepsilon$. Conceptually, *the smooth distance function has been replaced by a "relu" like non-linearity to capture numerical effects.* Further details on this kind of phase transition and how it *differs* from the traditional notion of a phase-transition can be found in (Appendix B.3.6).

We justify our choice of the forward trajectory in Appendix B.2.1, establishing that the detected transitions also occur in the sampling trajectory of a trained diffusion model. Crucially, this approach allows us to rely solely on the computationally efficient empirical forward process. We tailor $p_{\text{desired}}$ and $(\Omega_{1,0}, \Omega_{2,0})$ to illuminate specific phenomena. Algorithm 1 implements these steps. Note that, when $p_0$ has compact support (e.g., images), the PF-ODE ensures spatial continuity of $\mathcal{M}_\rho^{(n)}$ [Coddington and Levinson, 1955, Thm 5.2]; any jump thus indicates a genuine phase transition (for SDEs see Appendix B.2.3). A single forward Monte-Carlo sweep yields unbiased estimates of all terms in $\widetilde{\mathcal{M}}_\rho^{(n)}$ see Appendix B.2.5.

---

[3]A detailed treatment for the PF-ODE appears in Appendix B.2.2, and the SDE case is discussed in Appendix B.2.3.

[4]Details on the topological equivalence of a metric space on within-event fluctuations and the absolute distance of normalised cross-fluctuation from 1 can be found at Section B.1.3.1

**Algorithm 1** Compute $\big\{\widetilde{\mathcal{M}}_\rho^{(n)}(i)\big\}_{i=0}^n$ and the first merger step $i^\star$

---

**Require:** $p_{\text{data}}$, steps $n$, forward sampler PF, task objective $\mathcal{L}$, fluctuation $F_\rho^{(n)}$, metric $d$, threshold $\varepsilon$
1: $p_{\text{desired}} \leftarrow \text{DEFINEDESIRED}(p_{\text{data}}, \mathcal{L})$
2: $(\Omega_{1,0}, \Omega_{2,0}) \leftarrow \text{PICKEVENTS}(p_{\text{desired}}, \mathcal{L})$
3: $\mathbf{M} \leftarrow [\,]$                                                                          $\triangleright$ dynamic array for $\mathcal{M}_\rho^{(n)}(i)$
4: $i^\star \leftarrow n$                                                                                  $\triangleright$ merge no later than white noise
5: **for** $i = 0$ **to** $n$ **do**
6:     **if** $i > 0$ **then**
7:         $(\Omega_{1,i}, \Omega_{2,i}) \leftarrow \text{FORWARDSTEP}(\Omega_{1,i-1}, \Omega_{2,i-1})$
8:     **else**
9:         $(\Omega_{1,i}, \Omega_{2,i}) \leftarrow (\Omega_{1,0}, \Omega_{2,0})$
10:    **end if**
11:    $\delta \leftarrow d\big(F_\rho^{(n)}(\Omega_{1,i}), F_\rho^{(n)}(\Omega_{2,i})\big)$
12:    $\mathcal{M}_\rho^{(n)}(i) \leftarrow \begin{cases} \mathcal{M}_\rho^{(n)}(\Omega_{1,i}, \Omega_{2,i}), & \delta > \varepsilon, \\ 1, & \delta \leq \varepsilon \end{cases}$
13:    append $\widetilde{\mathcal{M}}_\rho^{(n)}(i)$ to $\mathbf{M}$
14:    **if** $\widetilde{\mathcal{M}}_\rho^{(n)}(i) = 1$ **and** $i^\star = n$ **then**
15:        $i^\star \leftarrow i$                                                                          $\triangleright$ first step where events merge
16:    **end if**
17: **end for**
18: **return** $\mathbf{M}$, $i^\star$

---

# 4   Demonstrations of our methodology

We next present case studies that show how our machinery can diagnose and subsequently improve the behaviour of diffusion samplers on practical tasks. *Our aim is to illustrate simple recipes for working with user-defined events that need not require a deep dive into the underlying theory but at the same time showcase the utility of our framework.*

## 4.1   Warm-up: Predicting convergence of the data distribution

In this case study, we abuse notation and let the sequence $\{p_0, \ldots, p_n\}$ represent the marginals under the PF-ODE for a data distribution $p_0$. Assume that the support of the data distribution, $\mathcal{D}_0 = \text{supp}(p_0)$, is *essentially disjoint* from the support of the Gaussian endpoint, $\mathcal{D}_n = \text{supp}(p_n) = \text{supp}\big(\mathcal{N}(\mathbf{0}_d, \mathbf{I}_d)\big)$. In high dimensions, this is realistic because $\mathcal{D}_n \approx \sqrt{d}\, S^{d-1}$ and therefore has asymptotically negligible volume [Vershynin, 2018][5]

We now show the setup of Algorithm 1 in this setting. Let $p_{\text{desired}}$ the desired distribution be defined over the union of all supports $\cup_{i=0}^n \mathcal{D}_n$ such that,

$$P_{\text{desired}}(Z) = \begin{cases} P_0(Z), & Z \subseteq \mathcal{D}_0, \\ 0, & \text{otherwise,} \end{cases} \tag{4.1}$$

where $P_{\text{desired}}, P_0$ are the probability masses associated with $p_{\text{desired}}, p_0$ respectively. It is easy to see that the PF-ODE on $\hat{p}_0 = p_{\text{desired}}$ leads to the set of marginals that we term as an **augmented process**, $\text{aug} = \{\hat{p}_i\}_{i=0}^n$ defined as,

$$\hat{P}_i(Z) = \begin{cases} P_i(Z), & Z \subseteq \mathcal{D}_i, \\ 0, & \text{otherwise,} \end{cases} \tag{4.2}$$

where $P_i$ (resp. $\hat{P}_i$) is the probability measure associated with $p_i$ (resp. $\hat{p}_i$). Operationally, aug coincides with the original PF-ODE but with an explicitly enlarged state space.[6]

---

[5]Note that, $\mathcal{D}_n$ becomes a null set for $d \to \infty$, so overlap with any finite-volume data manifold is vanishingly unlikely. Concentration inequalities showing rapid decay as a function of increasing $n$ can be found, e.g., in [Vershynin, 2018]. Formally, we assume that if $\mathcal{D}^* = \mathcal{D}_0 \cup \mathcal{D}_n$, then $P_0(\mathcal{D}^*), P_n(\mathcal{D}^*) \to 0$.

[6]The above construction is designed to measure how probability mass transfers away from any marginal $p_i$ towards another marginal $p_j$ where $j > i$. In our work we only concern ourselves with the case where $j = n$ so as to probe convergence to stationarity.

Set $\Omega_{1,0} = \mathcal{D}_0$ and $\Omega_{2,0} = \mathcal{D}_n$. Because $\Omega_{2,0}$ is already the support of the stationary distribution, $\Omega_{2,i} = \Omega_{2,0}$ for all $i > 0$. Detecting the smallest index $i$ for which $M_{\rho,\Omega_1,\Omega_2}^{(n)}(i) \approx 1$ therefore amounts to *deducing the onset of convergence* to the stationary distribution, this is in practice equivalent to a multivariate normality test on $p_i$. We adopt the D'Agostino-Pearson omnibus test [D'Agostino, 1971, D'Agostino and Pearson, 1973], the $p$-value threshold is the conventional $0.05$. We evaluate MNIST [Deng, 2012], CIFAR-10 [Krizhevsky et al., 2009], and a compressed INT-8 variant of ImageNet [Deng et al., 2009, Ryu, 2024], using the popular DDPM noise schedule [Ho et al., 2020]. Results appear in Figure 7 in Appendix C.2.

**Acceleration via early stopping.** Let $i^\star$ denote the smallest index flagged as Gaussian by the test. Starting the *reverse* sampler from $t = i^\star$, rather than the conventional $t = n$, preserves visual quality while saving $n - i^\star$ steps, as summarised in Table 1 across datasets. By convention, throughout our work a ($\downarrow$) sign denotes that lower values are better and vice-versa for an ($\uparrow$). For ImageNet we employ the state-of-the-art DiT-XL/2 model at $512 \times 512$ resolution [Peebles and Xie, 2022]; open-source checkpoints are used for the other datasets. A comparable observation was reported by Raya and Ambrogioni [2024], who derive $i^\star$ analytically under stronger assumptions on the data. Intriguingly, an unrelated theorem on Brownian equilibrium time predicts $i^\star$ surprisingly well for our data in Appendix B.3.1; clarifying this connection is left to future work, but the theorem already provides a useful heuristic.

| Model / Dataset | FID($\downarrow$) | Steps($\downarrow$) | GFLOPs($\downarrow$) |
|---|---|---|---|
| DiT-XL/2 (ImageNet, full) | $3.42 \pm 0.21$ | 250 | 4100 |
| DiT-XL/2 (ImageNet, ours) | $\mathbf{3.37 \pm 0.31}$ | **175** | **2870** |
| DDPM (MNIST, full) | $2.27 \pm 0.19$ | 1000 | 2000 |
| DDPM (MNIST, ours) | $2.29 \pm 0.17$ | **600** | **1200** |
| DDPM (CIFAR-10, full) | $3.62 \pm 0.35$ | 500 | 6000 |
| DDPM (CIFAR-10, ours) | $\mathbf{3.47 \pm 0.34}$ | **300** | **3600** |

Table 1: Acceleration achieved by stopping reverse diffusion at $t = i^\star$ instead of $t = n$ (DDPM schedule). FID scores are averaged over three runs with $95\%$ confidence intervals.

*Thus, starting the reverse sampler from the observed convergence time yields the same perceptual quality while reducing wall-clock cost— often by thousands of gigaflops (GFLOPs).*

**Visualizing the merger cascade.** Before detailing our next set of applications, it is instructive to visualise mergers occurring over a disjoint partition of events. Figure 2 presents the "merger cascade," a temporal hierarchy visualizing how disjoint events merge over diffusion time $t$. Events start as distinct leaves at $t = 0$. As time progresses upwards, branches join at a black dot—a *discrete merger event* (Appendix B.3.6)—at the precise moment their fluctuation tensors become indistinguishable. This dendrogram provides an intuitive map of how structural similarities are lost, motivating the targeted interventions that follow.

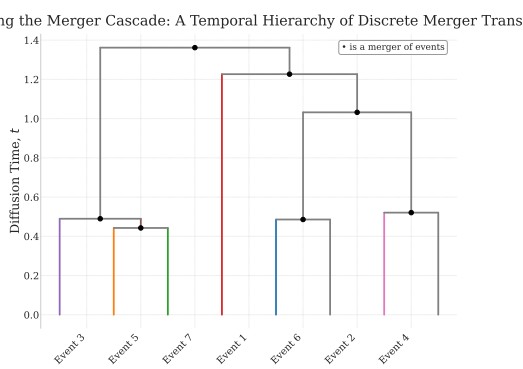

**Figure 2: Visualizing the merger cascade.** This temporal hierarchy illustrates how distinct, mutually disjoint events merge as the diffusion process evolves. Time $t$ flows upward. At $t = 0$, events are distinct leaves. As they diffuse, pairs whose fluctuation tensors become indistinguishable undergo a discrete merger event (black dot), and their branches merge. This cascade continues until all discriminating information has vanished into a single cluster.

## 4.2 Class-conditional generation

We now study the behaviour of Algorithm 1 when $p_{\text{desired}}$ is just the data split into $K$ *mutually disjoint classes* $\{\Omega_{1,0}, \dots, \Omega_{K,0}\}$ such that $\bigcup_{k=1}^{K} \Omega_{k,0} = \Omega$ where $\Omega$ is the sample space of the data distribution and $\Omega_{k,0} \cap \Omega_{\ell,0} = \varnothing$ whenever $k \neq \ell$. Throughout we write $\Omega_{k,t}$ for the image of $\Omega_{k,0}$ under the forward diffusion map at (discrete) time $t$. A formal algorithm summarizing this modification is present in Algorithm 4.

**A second-order statistic via centred kernel alignment.** Higher-order fluctuations for vector states become unwieldy, so we fix $n = 2$, leveraging covariance matrices to efficiently capture fourth-order moments. This captures non-linear feature information while maintaining second-order structure (Appendix B.3.3). For completeness, we also demonstrate an experiment with higher-order fluctuations, simplified using Isserlis'/Wick's theorem [Isserlis, 1918, Wick, 1950] (Appendix B.3.2). Let us for the $n = 2$ case define,

$$\Sigma_{k,t} = \text{Cov}\big(\rho \mid \Omega_{k,t}\big) \in \mathbb{R}^{d \times d}. \tag{4.3}$$

The normalised second-order cross-fluctuation $\mathcal{M}_\rho^{(2)}(\Omega_{k,t}, \Omega_{\ell,t})$ is then the *centred kernel alignment* (CKA) between $\Sigma_{k,t}$ and $\Sigma_{\ell,t}$ [Cortes et al., 2012, Cristianini et al., 2001, Kornblith et al., 2019]:

$$\text{CKA}_{k\ell}(t) = \frac{\text{Tr}\big(\Sigma_{k,t}^\top \Sigma_{\ell,t}\big)}{\sqrt{\text{Tr}\big(\Sigma_{k,t}^\top \Sigma_{k,t}\big)\, \text{Tr}\big(\Sigma_{\ell,t}^\top \Sigma_{\ell,t}\big)}}. \tag{4.4}$$

where $G_\rho^{(2)}(\Omega_{k,t}, \Omega_{l,t}) = \text{Tr}\big(\Sigma_{k,t}^\top \Sigma_{\ell,t}\big), \widehat{F}_\rho^{(4)}(\Omega_{k,t}) = \text{Tr}\big(\Sigma_{k,t}^\top \Sigma_{k,t}\big), \widehat{F}_\rho^{(4)}(\Omega_{l,t}) = \text{Tr}\big(\Sigma_{l,t}^\top \Sigma_{l,t}\big)$. We convert the above definition (4.4) into a discrete phase transition indicator (3.1) detecting whether the events have "merged / not merged" by the rule

$$\mathcal{M}_{k\ell}^{(2)}(t) = \begin{cases} \text{CKA}_{k\ell}(t), & \text{if } d\big(\widehat{F}_\rho^{(2n)}(\Omega_{1,i}), \widehat{F}_\rho^{(2n)}(\Omega_{2,i})\big) > \varepsilon, \\ 1, & \text{otherwise}, \end{cases} \tag{4.5}$$

Because the forward SDE is isotropic, $\Sigma_{k,t}$ contracts in the radial direction only; its spectrum therefore collapses towards a scaled identity. Hyper-contractivity implies that the merger of two classes is governed by the largest eigenvalues (See Appendix B.3.3 for further details)

$$\lambda_k^{\max}(t) = \lambda_{\max}(\Sigma_{k,t}), \qquad \lambda_\ell^{\max}(t) = \lambda_{\max}(\Sigma_{\ell,t}),$$

so we may replace the trace metric by $d\big(\lambda_k^{\max}(t), \lambda_\ell^{\max}(t)\big) < \varepsilon$. In practice, we fix

$$\varepsilon \approx \frac{\max_k \lambda_k^{\max}(0)}{400},$$

and use absolute eigenvalue differences for $d$. This criterion yields clear merger times across datasets; trajectories are plotted in Appendix C.3.

**Practical payoff: class-wise *Interval guidance*.** The state-of-the-art guidance algorithm, Interval guidance (IG) [Kynkäänniemi et al., 2024] argues that class conditioning should be applied only for a sub-interval $t \in (t_{\text{start}}, t_{\text{end}})$ for optimal results. The reference implementation runs an expensive grid search as detailed in Appendix C.3. Our fluctuation analysis supplies both bounds for free:

- $t_{\text{start}} = i^\star$, the convergence index from Section 4.1;
- $t_{\text{end}} = \max\big\{t : \mathcal{M}_{k\ell}^{(2)}(t) < 1 \text{ for some } \ell \neq k\big\}$.

Using these class-specific windows, we match or improve IG performance on ImageNet, CIFAR-10, and MNIST datasets. Our method is capable of cutting the combinatorial search cost by orders of magnitude. We present the results in Table 2. We utilise the metrics Frechet Inception Distance (FID) Heusel et al. [2017], Precision, Recall, and their more suitable variants for generative models such as Density and Coverage Naeem et al. [2020]. (We also show results for the StableDiffusion Rombach et al. [2022] model using Imagenet and the fine-grained classification dataset OxfordIIIITPets in Appendix C.3 along with further results.)

*Monitoring when class supports merge lets us place guidance exactly where it helps, eliminating costly grid searches and boosting class-conditional quality with negligible overhead.*

| Model/Dataset | FID (↓) | Precision (↑) | Recall (↑) | Density (↑) | Coverage (↑) |
|---|---|---|---|---|---|
| DiT-XL/2 (Imagenet, IG Baseline) | $3.22 \pm 0.16$ | $0.78 \pm 0.01$ | $0.23 \pm 0.05$ | $0.83 \pm 0.01$ | $0.35 \pm 0.02$ |
| DiT-XL/2 (Imagenet, IG Ours) | $\mathbf{2.86 \pm 0.15}$ | $\mathbf{0.83 \pm 0.02}$ | $\mathbf{0.26 \pm 0.04}$ | $\mathbf{0.85 \pm 0.01}$ | $\mathbf{0.39 \pm 0.02}$ |
| DDPM (MNIST, IG Baseline) | $2.15 \pm 0.06$ | $0.80 \pm 0.02$ | $0.25 \pm 0.01$ | $0.85 \pm 0.01$ | $0.36 \pm 0.02$ |
| DDPM (MNIST, IG Ours) | $\mathbf{1.99 \pm 0.11}$ | $\mathbf{0.85 \pm 0.01}$ | $\mathbf{0.28 \pm 0.03}$ | $\mathbf{0.89 \pm 0.02}$ | $\mathbf{0.40 \pm 0.01}$ |
| DDPM (CIFAR10, IG Baseline) | $3.32 \pm 0.25$ | $0.77 \pm 0.01$ | $0.19 \pm 0.14$ | $0.81 \pm 0.03$ | $0.32 \pm 0.02$ |
| DDPM (CIFAR10, IG Ours) | $\mathbf{3.01 \pm 0.14}$ | $\mathbf{0.79 \pm 0.02}$ | $\mathbf{0.22 \pm 0.01}$ | $\mathbf{0.84 \pm 0.01}$ | $\mathbf{0.35 \pm 0.04}$ |

Table 2: Class conditional generation

| Model/Dataset | CLIP Similarity (↑) | Precision (↑) | Recall (↑) | Density (↑) | Coverage (↑) |
|---|---|---|---|---|---|
| SD (iNaturalist, IG baseline) | $0.21 \pm 0.03$ | $0.70 \pm 0.01$ | $0.15 \pm 0.01$ | $0.75 \pm 0.01$ | $0.27 \pm 0.02$ |
| SD (iNaturalist, IG Ours) | $0.24 \pm 0.02$ | $0.73 \pm 0.01$ | $0.18 \pm 0.01$ | $0.78 \pm 0.02$ | $0.30 \pm 0.04$ |
| SD (iNaturalist, IG-ILVR Ours) | $\mathbf{0.27 \pm 0.01}$ | $\mathbf{0.76 \pm 0.01}$ | $\mathbf{0.19 \pm 0.02}$ | $\mathbf{0.81 \pm 0.01}$ | $\mathbf{0.31 \pm 0.03}$ |
| SD (CUB200, IG baseline) | $0.24 \pm 0.05$ | $0.75 \pm 0.01$ | $0.14 \pm 0.01$ | $0.79 \pm 0.02$ | $0.30 \pm 0.02$ |
| SD (CUB200, IG Ours) | $0.26 \pm 0.01$ | $0.78 \pm 0.05$ | $0.17 \pm 0.01$ | $0.82 \pm 0.02$ | $\mathbf{0.33 \pm 0.01}$ |
| SD (CUB200, IG-ILVR Ours) | $\mathbf{0.27 \pm 0.02}$ | $\mathbf{0.82 \pm 0.02}$ | $\mathbf{0.18 \pm 0.02}$ | $\mathbf{0.85 \pm 0.01}$ | $0.31 \pm 0.02$ |

Table 3: Rare class generation using StableDiffusion

## 4.3 Rare-class generation

We reuse the same modification of Algorithm 1 as in Section 4.2 but focus on *tail classes*-categories under-represented in the training data, and therefore poorly synthesised by off-the-shelf diffusion models. Concretely, we study the following datasets,

- **CUB-200** (200 bird species), and
- **iNaturalist 2019** (fine-grained flora & fauna).

Both have been identified as failure cases for Stable Diffusion [Samuel et al., 2024]. For each species $k$, we denote the source event by $\Omega_{k,0}$ and its forward image at timestep $t$ by $\Omega_{k,t}$.

**Merger-aware guidance window.** Adopting the notation of Sections 4.1 and 4.2, let

$$t_{\mathrm{conv}} = i^{\star}, \qquad t_{\mathrm{merge},k} = \min\big\{ t : \widetilde{\mathcal{M}}_{\rho}^{(2)}(\Omega_{k,t}, \Omega_{\ell,t}) = 1 \text{ for some } \ell \neq k \big\}.$$

The class-specific guidance interval is then $(t_{\mathrm{start},k}, t_{\mathrm{end},k}) = (t_{\mathrm{conv}}, t_{\mathrm{merge},k})$, that is, from global data convergence up to-but not beyond-the first time the target class becomes indistinguishable from any other class.

**Interval guidance with corrupted exemplars.** We observe that for this setting as compared to naive interval guidance, a slight tweak using corrupted exemplars can yield better results [7]. Inside this window, we apply *Interval Guidance* [Kynkäänniemi et al., 2024] and interpolate with a single exemplar $x_{\mathrm{ref}} \sim \Omega_{k,0}$ that is forward-noised in lock-step with the individual sample being generated. The conditioning term is effectively therefore the pair $(z_t, x_{\mathrm{ref},t})$, making the scheme a class-specific variant of ILVR [Choi et al., 2021] with intervals. No additional hyper-parameters are introduced; we keep the noise schedule and scaling constants from Section 4.2. A formal version of the algorithm is presented in Algorithm 3, refer to Appendix C.4 for further details.

Table 3 summarises the results using standard metrics, where we compare three settings: *(i)* naïve Interval Guidance over the full horizon, *(ii)* merger-aware Interval Guidance as in Section 4.2 and *(iii)* merger-aware IG + corrupted exemplar (ours). Across metrics, the merger-aware schedules consistently outperform the baselines, with the exemplar variant giving the largest gains.

We conclude that *as fluctuation-driven merger times generalise from mainstream classes to the tail classes without re-tuning, leveraging them yields a lightweight yet effective strategy for rare-class synthesis, mitigating class imbalance at essentially zero extra computational cost.*

## 4.4 Zero-shot classification

Once again we continue with a setting of Algorithm 1 as is in Section 4.2: where each sample $z \sim p_0$ belongs to exactly one class event $\Omega_{k,0}$. Given a trained *class-conditional* diffusion model $f_\theta$ with

---

[7]We observed no appreciable gain for the setup in Section 4.2, hence omit this tactic there.

label index set $\Lambda = \{1, \dots, K\}$, the goal is to predict

$$k^\star \;=\; \arg\max_{k \in \Lambda} \; p_k(z),$$

where $p_k(z)$ estimates the posterior class probability. We build upon the procedure proposed by Li et al. [2023] for this task. Let $z_t(z, \varepsilon) \in \mathbb{R}^d$ denote the forward-noised version of $z$ at step $t$, obtained with seed $\varepsilon \sim \mathcal{N}(\mathbf{0}, \mathbf{I})$. For that configuration, define the *logit*

$$s_k(t, \varepsilon) \;=\; -\big\| f_\theta\big(z_t(z, \varepsilon), \varepsilon, k\big) - \varepsilon \big\|_2^2, \qquad k \in \Lambda.$$

With $N$ noise seeds $\{\varepsilon_n\}_{n=1}^N$ and weights $w(t)$, our importance-weighted class score becomes

$$p_k(z) \;=\; \sum_{t = t_{\mathrm{start},k}}^{t_{\mathrm{stop},k}} w(t) \Big[\, \frac{1}{N} \sum_{n=1}^{N} q_k\big(t, \varepsilon_n\big) \Big], \quad q_k(t, \varepsilon) \;=\; \frac{\exp\big(s_k(t, \varepsilon)\big)}{\sum_{\ell \in \Lambda} \exp\big(s_\ell(t, \varepsilon)\big)}. \qquad (4.6)$$

where $q_k(t, \varepsilon)$ is the corresponding softmax probability. The choices **uniform**, **inverse-SNR** and **truncated inverse-SNR** correspond to three concrete $w$ distributions, each summing to 1 on its support. For the inverse-SNR strategies, the weights at each time are proportional to the inverse of the Signal to Noise Ratio (SNR) [Kingma et al., 2021] at that time, while uniform assigns a constant weight. The lower bound is set to $t_{\mathrm{start},k} = 0 \ \forall k$ for the uniform and inverse-SNR strategies. For the truncated case we set $t_{\mathrm{start},k} = 20$ for all $k$; (Appendix C.5) the upper bound

$$t_{\mathrm{stop},k} \;=\; \min\big\{ t : \widetilde{\mathcal{M}}_\rho^{(2)}(\Omega_{k,t}, \Omega_{\ell,t}) = 1 \text{ for some } \ell \neq k \big\}$$

ensures that no timestep beyond the first class merger contributes to (4.6). A formal version of the above algorithm is present in Algorithm 4 along with further details in Appendix C.5. Using a Stable Diffusion backbone [Rombach et al., 2022], we obtain the accuracies in Table 2. Merger-aware weighting improves on the uniform baseline of Li et al. [2023], where $t_{\mathrm{start},k} = 0$ and $t_{\mathrm{stop},k} = T$ for all classes $k$, where $T$ is the horizon of the diffusion process; truncated inverse-SNR performs best.

| Method | ImageNet ↑ | CIFAR-10 ↑ | Oxford-IIIT Pets ↑ |
|---|---|---|---|
| SD, uniform (Li et al.) | 54.96±0.67 | 84.67±1.23 | 82.87±0.39 |
| SD, uniform (ours) | 57.91±0.53 | 85.17±0.17 | 86.17±0.26 |
| SD, inverse-SNR | 64.17±0.33 | 87.26±0.67 | 88.17±0.29 |
| SD, trunc. inverse-SNR | **65.28±0.46** | **88.38±0.43** | **89.15±0.26** |
| CLIP RN-50 | 58.41±0.35 | 75.42±0.26 | 85.61±0.29 |
| OpenCLIP ViT-H/14 | 76.91±0.75 | 96.87±0.59 | 94.61±0.37 |

Table 4: Zero-shot multi-class accuracy (%); $\pm$ 95% CI over five runs.

That truncated inverse-SNR outperforms uniform weighting suggests the corrupted marginals remain most discriminative shortly *before* class supports merge. Appendix C.5.1 analyses this effect in detail via linear probes for binary classification between Imagenet classes.

*Thus, merger times not only optimise guidance but also delimit the timesteps that matter for zero-shot recognition, yielding tangible gains at negligible cost.*

### 4.5 Zero-shot style transfer

Let $p_0$ be a *source* image distribution on $\mathcal{X} \subset \mathbb{R}^d$ and let $p^\star = \mathcal{T}_{\mathrm{style}}(p_0)$ be the same semantic content rendered in a new artistic style. We assume

1. **Bijectivity.** $\mathcal{T}_{\mathrm{style}} : \mathcal{X} \to \mathcal{X}$ is invertible on the supports, so $p^\star$ is a valid density and $\mathrm{supp}(p^\star) = \mathcal{T}_{\mathrm{style}}\big(\mathrm{supp}(p_0)\big)$.

2. **Structure preservation in the Fourier domain.** For a density $q$ write $\widehat{q}(\boldsymbol{\xi}) = \int_{\mathcal{X}} q(\mathbf{x}) \, e^{-2\pi i \boldsymbol{\xi}^\top \mathbf{x}} \mathrm{d}\mathbf{x}$ and define the frequency-domain norm $\|p - q\|_f = \|\widehat{p} - \widehat{q}\|_{L^2(\mathbb{R}^d)}$. We posit the regularity bound

$$\|p_0 - p^\star\|_f \;\leq\; \delta, \qquad \text{with } 0 < \delta \ll 1. \qquad (4.7)$$

**Fluctuation adaptation lemma for style transfer.** Let $\rho(\mathbf{x}) = \mathbf{x}$ be the identity state operator for $p_0$ and let $\rho^\star = \rho \circ \mathcal{T}_{\text{style}}^{-1}$ for $p^\star$. Appendix B.3.4 proves that for every measurable $\Omega \subseteq \mathcal{X}$ and each moment order $n \in \{1, 2, 3, 4\}$,

$$\left| \widehat{F}_\rho^{(n)}(\Omega) \;-\; \widehat{F}_{\rho^\star}^{(n)}\big(\mathcal{T}_{\text{style}}(\Omega)\big) \right| \;\leq\; C_n\, \delta, \tag{4.8}$$

with a constant $C_n$ independent of $\Omega$. Hence the fluctuation trajectories of the *source* distribution $p_0$ approximate those of the *target-style* distribution $p^\star$ to $O(\delta)$ accuracy.

**Choosing a start time for reverse denoising.** For each semantic class $\lambda \in \Lambda$ let

$$m_\lambda \;=\; \min\Big\{ t : \widetilde{\mathcal{M}}_\rho^{(2)}\big(\Omega_{\lambda, t}, \Omega_{\ell, t}\big) = 1 \text{ for some } \ell \neq \lambda \Big\}$$

be its earliest merger time under the *forward* diffusion of $p_0$ (see Section 4.2). By the lemma above, $m_\lambda$ remains a valid approximation for the target-style chain. Therefore, when we apply a target-style model (e.g. Stable Diffusion fine-tuned on van-Gogh paintings) we can **kick-start** reverse denoising at $t = m_\lambda$ rather than at the full horizon $t = n$, achieving style transfer in fewer steps. Thus $p_{\text{desired}}$ in Algorithm 1 here is the *source distribution* while the diffusion model is actually trained on a different *target distribution*. A formal version of the above algorithm is presented in Algorithm 5. We also note that the current procedure naturally extends to inverse problems using diffusion models, following a similar procedure under conditions analogous to (4.8). These conditions align with prior methodologies for inverse problems Song et al. [2022]. Exploring these extensions is left for future work

We follow the setup of [Meng et al., 2021] wherein an optimal time needs to be estimated till which noising must occur. Our baseline is the grid search technique used in [Meng et al., 2021] computed over an entire dataset. We present sample results for the Studio Ghibli Miyazaki and Ghibli [2014] and Van Gogh Roojen [2019] artistic styles in Table 5 using the PSNR and MSE metrics. See Appendix C.6 for experimental details, figures and empirical verification of the fluctuation adaptation lemma, and additional results for the Elden Ring Software and Entertainment [2022] and Arcane Production and Games [2021] styles.

| Style | Ghibli | | van-Gogh | |
|---|---|---|---|---|
| Models/Metrics | PSNR ($\uparrow$) | MSE ($\downarrow$) | PSNR ($\uparrow$) | MSE ($\downarrow$) |
| SD Edit (OxfordIIITPets) | $25.67 \pm 1.14$ | $0.09 \pm 0.001$ | $26.16 \pm 0.72$ | $0.09 \pm 0.003$ |
| **Ours** (OxfordIIITPets) | $\mathbf{28.71 \pm 0.86}$ | $\mathbf{0.03 \pm 0.005}$ | $\mathbf{28.65 \pm 0.49}$ | $\mathbf{0.03 \pm 0.002}$ |
| SD Edit (AFHQv2) | $27.12 \pm 0.59$ | $0.05 \pm 0.006$ | $27.49 \pm 0.27$ | $0.04 \pm 0.004$ |
| **Ours** (AFHQ v2) | $\mathbf{27.65 \pm 0.57}$ | $\mathbf{0.04 \pm 0.004}$ | $\mathbf{28.07 \pm 0.32}$ | $\mathbf{0.03 \pm 0.006}$ |

Table 5: Style Transfer results for Ghibli and van-Gogh styles

*Thus, as under mild Fourier regularity, the merger schedule learned on a source dataset transfers directly to any purely stylistic variant of that dataset, the result is a true zero-shot style transfer procedure that inherits the efficiency gains of Section 4.1 without additional tuning.*

## 5 Conclusion

We introduced *cross-fluctuation mergers*: a centred-moment statistic that fires precisely when two regions of a diffusion trajectory become indistinguishable. The concept links statistical-physics fluctuation theory with generative modelling and yields practical benefits: early stopping in the forward process cuts a quarter to two-fifths of reverse steps; class-wise guidance windows raise conditional image quality without grid search; merger-aware weighting upgrades zero-shot classification, and the same signal powers zero-shot style transfer and rare-class generation—all without re-training or new knobs to tune. Our analysis assumes a variance-preserving noise schedule; extending to non-VP schedules (e.g., EDM [Karras et al., 2022]), anisotropic diffusions, higher-order fluctuations for complex data, and different modalities are promising directions for future work. We expect our framework to inspire further developments in tighter theory, training-time objectives, and applications.

## Acknowledgments

We thank Xiang Cheng for his support in the conceptualisation of this project and for valuable discussions throughout its development. We are also grateful to Joshua Robinson for insightful early conversations and for suggesting experimental setups. We thank Sanket Kumar Tripathy for engaging discussions on the conceptual connections between our work and ideas from statistical physics. SNR, MKL, and SS acknowledge generous support from the Alexander von Humboldt Foundation.

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

# Contents

# A  Notation used throughout the paper

Table 6: Global symbols

| Symbol | Domain / type | Meaning |
|---|---|---|
| $\mathbf{x}_0 \sim p_0$ | $\mathbb{R}^d$ | Data sample from initial distribution $p_0$ |
| $p_{\mathrm{desired}}$ | density on $\mathbb{R}^d$ | Desired distribution |
| $\mathbf{x}_t$ | $\mathbb{R}^d$ | Forward-diffused variable at time $t$ |
| $\beta(t),\ \beta_t$ | $\mathbb{R}_{>0}$ / $(0,1)$ | Continuous / discrete noise schedule |
| $J(t)$ | $(0,1]$ | Signal attenuation factor (2.4) |
| $p_t$ | density on $\mathbb{R}^d$ | Marginal distribution of $\mathbf{x}_t$ |
| $s_\theta(\mathbf{x}, t)$ | $\mathbb{R}^d \to \mathbb{R}^d$ | Learned score network |
| $\rho(\cdot)$ | map $\Omega \to \mathbb{R}^m$ | State operator (usually $\rho(\mathbf{x}) = \mathbf{x}$) (Section 3) |
| $F_\rho^{(n)}(\Omega)$ | $\mathbb{R}_{\geq 0}$ | $n^{\mathrm{th}}$ centred fluctuation moment on event $\Omega$ (2.5) |
| $\widehat{F}_\rho^{(n)}(\Omega_i), \widehat{F}_{\rho_i}^{(n)}(\Omega_i)$ | $\mathbb{R}_{\geq 0}$ | $2n^{\mathrm{th}}$ within-event fluctuation moment on $\Omega_i$ (2.7) |
| $G_\rho^{(n)}(\Omega_1, \Omega_2), G_{\rho_1, \rho_2}^{(n)}(\Omega_1, \Omega_2)$ | $\mathbb{R}_{\geq 0}$ | Unnormalised cross-fluctuation (2.6) |
| $\mathcal{M}_\rho^{(n)}(\Omega_1, \Omega_2), \mathcal{M}_{\rho_1, \rho_2}^{(n)}(\Omega_1, \Omega_2)$ | $[0,1]$ | Normalised cross-fluctuation (2.8) |
| $\Omega_{k,0}$ | event in $\Omega$ | Class-$k$ source region ($k = 1, \ldots, K$) (Section 4.2) |
| $\Omega_{k,t}$ | event | Image of $\Omega_{k,0}$ after $t$ forward steps |
| $\Sigma_{k,t}$ | $\mathbb{S}_+^d$ | Covariance of $\Omega_{k,t}$ |
| $\lambda_k^{\max}(t)$ | $\mathbb{R}_{\geq 0}$ | Maximum eigenvalue of $\Sigma_{k,t}$ |
| $\mathcal{M}_\rho(t)$ | $[0,1]$ | Shorthand for $\mathcal{M}_\rho^{(2)}(\Omega_{1,t}, \Omega_{2,t})$ |
| $i^\star$ | $\{0, \ldots, T\}$ | First index where $\mathcal{M}_\rho^{(n)}(t) = 1$ (merger) (Section 3) |
| $w(t)$ | probability mass | Importance weight over timesteps (Section 4.4) |
| $\mathrm{SNR}(t)$ | $\mathbb{R}_{\geq 0}$ | $\alpha_t^2 / (1 - \alpha_t^2)$ (signal-to-noise) |
| $\mathrm{Tr}(\cdot)$ | $\mathbb{R}$ | Matrix trace operator |

Table 7: Time- and index-specific symbols

| Symbol | Type / range | Meaning |
|---|---|---|
| $t$ | $\mathbb{R}_{\geq 0}$ | Continuous diffusion time (PF-ODE or SDE) |
| $s, u$ | $\mathbb{R}_{\geq 0}$ | Generic continuous times used in flow composition ($0 \leq s \leq t \leq u \leq T$) |
| $i$ | $\{0, \ldots, n\}$ | Discrete forward-process index ($i = 0$ data; $i = n$ white noise) |
| $T$ | positive real / integer | **Continuous** final time (SDE/ODE) **or discrete** horizon with schedule $\{\beta_t\}_{t=1}^T$ |
| $n$ | $\mathbb{N}$ | Chosen number of forward (or reverse) **discrete** steps in an experiment (may be $n = T$) |
| $\Delta t$ | $\mathbb{R}_{>0}$ | Integration step size in continuous-time numerical solvers |
| $\beta_t$ | sequence on $\{1, \ldots, T\}$ | Discrete noise-schedule value at step $t$ |
| $\beta(t)$ | function $[0, T] \to \mathbb{R}_{>0}$ | Continuous noise-schedule function |
| $i^\star$ | integer | Earliest discrete index where two events first merge (convergence index) |
| $t_{\mathrm{conv}}$ | $\mathbb{R}_{\geq 0}$ | Same as $i^\star$ but expressed on the continuous time axis |
| $t_{\mathrm{merge},k}$ | $\mathbb{R}_{\geq 0}$ | First time class $k$ merges with any other class |
| $t_{\mathrm{start},k}$ | $\mathbb{R}_{\geq 0}$ or int | Lower bound of guidance / weighting window for class $k$ |
| $t_{\mathrm{stop},k}$ | same type as above | Upper bound of the window for class $k$ |
| $t_{\mathrm{u}\to\mathrm{s}},\ t_{\mathrm{s}\to\mathrm{c}}$ | $\mathbb{R}_{\geq 0}$ | Thermodynamic phase-transition times (unbiased→speciation, speciation→condensation) |
| $t_{\mathrm{mix}}(\varepsilon)$ | $\mathbb{R}_{\geq 0}$ (Appendix B.3.6) | $\varepsilon$-mixing time of the VP–SDE (Appendix B.1.4) |
| $t_{\mathrm{cpl}}(\varepsilon)$ | integer | $\varepsilon$-coupling time for discrete Markov chains (Appendix B.1.4) |
| $t_{k\ell}^{\mathrm{lat}}(\varepsilon)$ | $\mathbb{R}_{\geq 0}$ | First lattice-merger time between classes $k$ and $\ell$ with tolerance $\varepsilon$ (Appendix B.3.6) |
| $\eta_t$ | $[0,1]$ | Interpolation schedule value used in Algorithm 3 (rare-class generation) |

# B   Theoretical contributions

This appendix provides the theoretical support for the results presented in the main paper. The content is structured in three parts: we first introduce the general framework, then apply it to diffusion models, and finally, we connect our work to existing methods while also presenting new theoretical results and interpretations.

Appendix B.1 introduces the cross-fluctuation framework. It covers the necessary definitions, proves that merger times can serve as an efficient proxy for convergence in total variation, and shows that the method is applicable to both discrete and continuous state spaces.

Appendix B.2 applies this framework specifically to diffusion models. This section justifies the use of the forward process for our analysis, details the dynamics of event structures under the model's SDE/ODE, confirms the consistency of the resulting merger events, and provides an efficient method for their estimation.

Appendix B.3 discusses connections to other methods and provides additional context. We show the relationship between our framework and Centered Kernel Alignment (CKA), prove the fluctuation adaptation lemma for style transfer, and conclude by framing our analysis of discrete phase transitions (what we term *lattice transitions*) within the context of statistical physics.

## B.1   Theoretical foundations of the cross fluctuation framework

### B.1.1   A concise primer on fluctuation theory with physical intuition

Here we provide a concise overview of the fluctuation-theoretic framework that underpins the notation introduced in Section 2.2.

**From observables to fluctuations.**   In statistical physics, a system's properties are understood by measuring *observables* (e.g., energy, momentum). For a distribution $p$, the expectation of an observable $\mathcal{A}$ is $\mathbb{E}_p[\mathcal{A}]$. Deviations from this mean, or *fluctuations*, reveal the system's internal structure and correlations. Our framework generalises this by treating the state vector $\rho(\omega)$ itself as the core observable.

The key object, the $n^{th}$-*order fluctuation*  (Eq. (2.5)),

$$\mathcal{F}_\rho^{(n)}(\omega) := \bigotimes_{k=1}^{n} \big(\rho(\omega) - \mathbb{E}[\rho]\big),$$

and its conditional expectation $\mathbb{E}_i[\mathcal{F}_\rho^{(n)}]$ over an event $\Omega_i$, directly correspond to the centered moments used to characterise complex distributions. For example, $\mathbb{E}_i[\mathcal{F}_\rho^{(2)}]$ is the conditional covariance matrix, a fundamental second-order statistic.

**Diffusion models as dynamical systems.**   The forward diffusion process, whether described by the SDE (Eq. (2.1)) or the PF-ODE (Eq. (2.2)), defines a dynamical system that evolves an initial data distribution $p_0$ into a sequence of marginals $\{p_t\}$. The PF-ODE,

$$\frac{\mathrm{d}\mathbf{x}_t}{\mathrm{d}t} = -\tfrac{1}{2}\beta(t)\,\mathbf{x}_t - \beta(t)\,\nabla_{\mathbf{x}} \log p_t(\mathbf{x}_t),$$

describes a deterministic flow governed by a vector field. Its associated continuity equation, $\partial_t p_t + \nabla_{\mathbf{x}} \cdot (p_t \mathbf{v}_t) = 0$, confirms that $p_t$ evolves via a deterministic transport of probability mass. This deterministic evolution makes it a perfect setting for applying our fluctuation analysis to track how the moment structures of different subpopulations (events) evolve and merge over time [Biroli and Mézard, 2023, Raya and Ambrogioni, 2024].

**Fluctuations and the significance of mergers.**   A cornerstone of statistical physics is the deep connection between a system's internal fluctuations and its large-scale correlational structure [Chaikin et al., 1995, Kivelson et al., 2024]. This principle, often formalised in Fluctuation-Dissipation Theorems, provides the physical intuition for our approach.

For our purposes, this connection justifies why monitoring cross-fluctuations is meaningful. The normalised cross-fluctuation $\mathcal{M}_\rho^{(n)}$ acts as a generalised correlation function between the moment

structures of two events. A sharp change in this correlation (i.e., a merger where $|\mathcal{M}_\rho^{(n)}| \to 1$) signals that the two events have become statistically indistinguishable with respect to their $n$-th order structure. In physical systems, such a loss of distinguishability is the hallmark of a phase transition, where the system undergoes a qualitative change. While we do not compute system-wide thermodynamic quantities directly, detecting the merger via $\mathcal{M}_\rho^{(n)}$ serves as a direct, practical probe for these critical points in the diffusion trajectory.

**Working with multiple states.** In Eqs. (2.6) and (2.8) we defined

$$G_{\rho_1,\rho_2}^{(n)} \quad \text{and} \quad \mathcal{M}_{\rho_1,\rho_2}^{(n)},$$

allowing $\rho_1$ and $\rho_2$ to be *heterogeneous* (i.e., $\rho_1 \neq \rho_2$). As the following example illustrates, different state-operator choices can yield qualitatively different cross-fluctuation behaviour.

**Example 1** *Let $\Omega$ be the unit circle in $\mathbb{R}^2$. Consider two events: $\Omega_1$, the arc in the first quadrant ($x > 0, y > 0$), and $\Omega_2$, the arc in the second quadrant ($x < 0, y > 0$). Assume a uniform distribution on the circle. Let the state operator be the identity, $\rho(x, y) = [x, y]^\top$. The global mean is $\mathbb{E}[\rho] = [0, 0]^\top$.*

*For $n = 2$, we compute the conditional covariance matrices $\Sigma_1 = \mathbb{E}_1[\mathcal{F}_\rho^{(2)}]$ and $\Sigma_2 = \mathbb{E}_2[\mathcal{F}_\rho^{(2)}]$ by evaluating standard integrals. This leads to the following conclusions,*

- *The mean of $\Omega_1$ is $\mathbb{E}_1[\rho] = [2/\pi, 2/\pi]^\top$. Its covariance matrix $\Sigma_1$ will have a dominant eigenvector along the direction $[1, -1]^\top$, indicating negative correlation (as $y$ decreases when $x$ increases along the arc).*

- *The mean of $\Omega_2$ is $\mathbb{E}_2[\rho] = [-2/\pi, 2/\pi]^\top$. Its covariance matrix $\Sigma_2$ will have a dominant eigenvector along $[1, 1]^\top$, indicating positive correlation.*

*Since the dominant structures (covariance matrices) $\Sigma_1$ and $\Sigma_2$ are nearly orthogonal, their Frobenius inner product $Tr(\Sigma_1^\top \Sigma_2)$ will be close to zero. Consequently, their normalized cross-fluctuation $\mathcal{M}_\rho^{(2)}(\Omega_1, \Omega_2)$ will be close to 0, correctly identifying the events as structurally distinct.*

This flexibility is crucial in quantum mechanics, where one often considers distinct pure-state projectors

$$\rho_i = |\psi_i\rangle \langle\psi_i|$$

on a Hilbert space $\mathcal{H}$.[8] Their cross-fluctuations underpin tasks such as quantum state tomography and discrimination of non-orthogonal states Nielsen and Chuang [2010], Peres [1995].

More generally, heterogeneous $\rho_1, \rho_2$ detect an *alignment* between two evolving state spaces. For example, if text and image embeddings are both driven by the same diffusion dynamics, choosing $\rho_1$ and $\rho_2$ to sample each modality reveals how a given concept manifests across them. We leave such multimodal extensions for future work.

### B.1.2 On the validity of fluctuation theory

The entire fluctuation framework rests on the existence and properties of moments, which are fundamentally linked to the derivatives of the *characteristic function* (CF). For a random vector $\boldsymbol{\rho} \in \mathbb{R}^d$, its characteristic function is the Fourier transform of its probability law:

$$\varphi_{\boldsymbol{\rho}}(\mathbf{t}) = \mathbb{E}\big[e^{i\mathbf{t}^\top \boldsymbol{\rho}}\big], \quad \mathbf{t} \in \mathbb{R}^d.$$

The CF always exists and satisfies $|\varphi_{\boldsymbol{\rho}}(\mathbf{t})| \leq 1$. If moments up to order $n$ exist, the CF is $n$-times differentiable at the origin. The derivatives of the CF generate the moment tensors. For example, the raw second moment tensor (a matrix) is given by the Hessian of the CF at the origin:

$$\mathbb{E}[\rho_j \rho_k] = \frac{1}{i^2} \frac{\partial^2 \varphi_{\boldsymbol{\rho}}}{\partial t_j \partial t_k}\bigg|_{\mathbf{t}=\mathbf{0}}.$$

---

[8]Mathematically, measurements project the density operator onto the eigenspace associated with the outcome; see Nielsen and Chuang [2010] for details.

To obtain the *centered* moment tensors used in our framework, one differentiates the characteristic function of the centered variable $\boldsymbol{\rho} - \mathbb{E}[\boldsymbol{\rho}]$. The conditional expectation of the $n^{\text{th}}$-order fluctuation tensor, $\mathbb{E}_k[\mathcal{F}_\rho^{(n)}]$, is thus completely determined by the derivatives of the conditional characteristic function $\varphi_{\boldsymbol{\rho}}(\mathbf{t}|\Omega_k)$ at the origin. This connection is crucial for the moment-TV inequality used in our theoretical results (Theorem 5).

**Foundational facts.** The theory of characteristic functions is well-established [Lukacs, 1970]:

1. **Uniqueness.** If $\varphi_{\boldsymbol{X}}(\mathbf{t}) = \varphi_{\boldsymbol{Y}}(\mathbf{t})$ for all $\mathbf{t}$, then $\boldsymbol{X}$ and $\boldsymbol{Y}$ have the same distribution.

2. **Inversion.** The distribution can be recovered from the CF. For instance, in one dimension:
$$\text{CDF}(x) = \tfrac{1}{2} - \frac{1}{\pi} \int_0^\infty \frac{\Im\left(e^{-\mathrm{i}tx}\,\varphi_X(t)\right)}{t}\,\mathrm{d}t \quad \text{[Dudley, 2018]}.$$

3. **Convolution.** For independent $\boldsymbol{X}, \boldsymbol{Y}$, the CF of their sum is the product of their CFs: $\varphi_{\boldsymbol{X}+\boldsymbol{Y}}(\mathbf{t}) = \varphi_{\boldsymbol{X}}(\mathbf{t})\,\varphi_{\boldsymbol{Y}}(\mathbf{t})$.

**Theorem 2 (Bochner's theorem for $\mathbb{R}^d$)** *A function $\varphi\colon \mathbb{R}^d \to \mathbb{C}$ is a characteristic function of some random vector if and only if it is positive-definite, continuous at the origin, and $\varphi(\mathbf{0}) = 1$.*

*Proof.* See Rudin [1962, Thm. 15.2] for the 1D case, which generalises to $\mathbb{R}^d$.

**Remark 3 (Existence of the characteristic function)** *We assume the state operator $\rho$ has a well-defined characteristic function, which is a very mild condition. It is automatically satisfied by most data distributions in machine learning and physics [Blanchard and Brüning, 2015, Bertini et al., 2002]. This assumption is more fundamental than the existence of a density, as characteristic functions also exist for purely discrete or singular measures [Dudley, 2018, Rudin, 1962], giving the approach greater utility [Ansari et al., 2020, Sriperumbudur et al., 2010].*

### B.1.3 Merger Times as an efficient proxy for convergence in total variation

The justification for using merger times as a proxy for convergence is a two-step argument connecting our practical measurement to fundamental theory.

First, we validate our specific measurement technique, proving that the computationally efficient method of directly thresholding the distance between moment tensors is topologically equivalent to the intuitive alternative of monitoring the cross-fluctuation similarity $|\mathcal{M}_\rho^{(n)}|$.

Second, we establish the core theoretical link. A moment-TV inequality proves that this proximity between moments rigorously guarantees that the total variation distance between the underlying distributions also vanishes.

This validates our method: measuring tensor distance is a sound and efficient proxy for observing true distributional convergence.

### B.1.3.1 Equivalence of distance and similarity based merger thresholds

In our main methodology (Section 3), we define the discrete transition detector $\widetilde{\mathcal{M}}_\rho^{(n)}(i)$ using a threshold on the distance between conditional expected fluctuation tensors:

$$\widetilde{\mathcal{M}}_\rho^{(n)}(i) = \begin{cases} |\mathcal{M}_\rho^{(n)}(\Omega_{1,i}, \Omega_{2,i})|, & \|\mathbb{E}_1[\mathcal{F}_\rho^{(n)}(\Omega_{1,i})] - \mathbb{E}_2[\mathcal{F}_\rho^{(n)}(\Omega_{2,i})]\|_{\mathcal{H}_n} > \varepsilon, \\ 1, & \text{otherwise.} \end{cases} \tag{B.1}$$

An alternative, and perhaps more direct, formulation would be to threshold the value of $|\mathcal{M}_\rho^{(n)}|$ itself:

$$\widetilde{\mathcal{M}}_{\rho,\text{alt}}^{(n)}(i) = \begin{cases} |\mathcal{M}_\rho^{(n)}(\Omega_{1,i}, \Omega_{2,i})|, & 1 - |\mathcal{M}_\rho^{(n)}(\Omega_{1,i}, \Omega_{2,i})| > \vartheta, \\ 1, & \text{otherwise.} \end{cases} \tag{B.2}$$

Here, we show that these two thresholding criteria are topologically equivalent, meaning they detect the same notion of convergence. The first formulation is often more computationally stable and efficient and hence is the prefered choice in our work.

**Theorem 4** *Let the conditional expected fluctuation tensors $\mathbb{E}_k[\mathcal{F}_\rho^{(n)}(\Omega_k)]$ for events $\Omega_k$ reside in a Hilbert space $\mathcal{H}_n$ with norm $\|\cdot\|_{\mathcal{H}_n}$. Define two metrics on the space of pairs of events $(\Omega_1, \Omega_2)$:*

1. *The direct distance between their moment tensors: $d_{\mathcal{F}}(\Omega_1, \Omega_2) := \|\mathbb{E}_1[\mathcal{F}_\rho^{(n)}(\Omega_1)] - \mathbb{E}_2[\mathcal{F}_\rho^{(n)}(\Omega_2)]\|_{\mathcal{H}_n}$.*

2. *The similarity-based distance: $d_{\mathcal{M}}(\Omega_1, \Omega_2) := 1 - |\mathcal{M}_\rho^{(n)}(\Omega_1, \Omega_2)|$.*

*If the mapping $\Omega \mapsto \mathbb{E}_k[\mathcal{F}_\rho^{(n)}(\Omega)]$ is continuous with respect to a suitable topology on the space of events, then the metrics $d_{\mathcal{F}}$ and $d_{\mathcal{M}}$ are topologically equivalent in any region where $\|\mathbb{E}_k[\mathcal{F}_\rho^{(n)}(\Omega_k)]\|_{\mathcal{H}_n}$ is bounded away from zero.*

*Proof.* To establish topological equivalence, we show that for any sequence of event pairs $\{(\Omega_{1,k}, \Omega_{2,k})\}_{k=1}^\infty$, convergence under metric $d_{\mathcal{F}}$ is equivalent to convergence under metric $d_{\mathcal{M}}$. Let $A_k = \mathbb{E}_1[\mathcal{F}_\rho^{(n)}(\Omega_{1,k})]$ and $B_k = \mathbb{E}_2[\mathcal{F}_\rho^{(n)}(\Omega_{2,k})]$ be the corresponding sequence of tensors in the Hilbert space $\mathcal{H}_n$.

**Part 1: $d_{\mathcal{F}} \to 0 \implies d_{\mathcal{M}} \to 0$.** Assume that $d_{\mathcal{F}}(\Omega_{1,k}, \Omega_{2,k}) \to 0$ as $k \to \infty$. This means $\|A_k - B_k\|_{\mathcal{H}_n} \to 0$.

First, by the reverse triangle inequality, we have:

$$\left| \|A_k\|_{\mathcal{H}_n} - \|B_k\|_{\mathcal{H}_n} \right| \le \|A_k - B_k\|_{\mathcal{H}_n}.$$

Since the right-hand side goes to zero, the norms converge to each other. As we are in a region where the norms are bounded away from zero, if $\|A_k\|$ converges to a limit $L > 0$, then $\|B_k\|$ must also converge to $L$.

Second, the inner product is a continuous function on $\mathcal{H}_n \times \mathcal{H}_n$. This follows from the Cauchy-Schwarz inequality:

$$|\langle A_k, B_k \rangle - \langle B_k, B_k \rangle| = |\langle A_k - B_k, B_k \rangle|$$
$$\le \|A_k - B_k\|_{\mathcal{H}_n} \|B_k\|_{\mathcal{H}_n}.$$

As $\|A_k - B_k\|_{\mathcal{H}_n} \to 0$ and $\|B_k\|_{\mathcal{H}_n}$ is bounded, the right-hand side goes to zero. This shows $\langle A_k, B_k \rangle - \|B_k\|^2 \to 0$. Since $\|A_k\| \to \|B_k\|$, we have $\langle A_k, B_k \rangle \to L^2$.

Now consider the limit of $|\mathcal{M}_\rho^{(n)}|$:

$$\lim_{k \to \infty} |\mathcal{M}_\rho^{(n)}(\Omega_{1,k}, \Omega_{2,k})| = \lim_{k \to \infty} \frac{|\langle A_k, B_k \rangle|}{\|A_k\|_{\mathcal{H}_n} \|B_k\|_{\mathcal{H}_n}} = \frac{L^2}{L \cdot L} = 1.$$

Therefore, $d_{\mathcal{M}}(\Omega_{1,k}, \Omega_{2,k}) = 1 - |\mathcal{M}_\rho^{(n)}(\Omega_{1,k}, \Omega_{2,k})| \to 0$.

**Part 2: $d_{\mathcal{M}} \to 0 \implies d_{\mathcal{F}} \to 0$.** Assume that $d_{\mathcal{M}}(\Omega_{1,k}, \Omega_{2,k}) \to 0$ as $k \to \infty$. This means $|\mathcal{M}_\rho^{(n)}(\Omega_{1,k}, \Omega_{2,k})| \to 1$. Let $\theta_k$ be the angle between $A_k$ and $B_k$. Then $|\cos \theta_k| = |\mathcal{M}_\rho^{(n)}(\Omega_{1,k}, \Omega_{2,k})|$, which implies $|\cos \theta_k| \to 1$.

Consider the squared distance $d_{\mathcal{F}}^2 = \|A_k - B_k\|^2$. Using the law of cosines in a Hilbert space:

$$\|A_k - B_k\|^2 = \|A_k\|^2 + \|B_k\|^2 - 2\langle A_k, B_k \rangle = \|A_k\|^2 + \|B_k\|^2 - 2\|A_k\|\|B_k\| \cos \theta_k.$$

For the distance to converge to zero, we require not only that the angle between the vectors vanishes (i.e., $\cos \theta_k \to 1$), but also that their norms converge to the same value. The physical process of distributional merging implies that not only the orientation of the moment structures but also their magnitudes become identical. We therefore assume that as events merge, if $\|A_k\|$ converges, then $\|B_k\|$ converges to the same limit $L > 0$. Under this assumption:

$$\lim_{k \to \infty} \|A_k - B_k\|^2 = \lim_{k \to \infty} (\|A_k\|^2 + \|B_k\|^2 - 2\|A_k\|\|B_k\| \cos \theta_k)$$
$$= L^2 + L^2 - 2(L)(L)(1)$$
$$= 0.$$

Thus, $d_{\mathcal{F}}(\Omega_{1,k}, \Omega_{2,k}) = \|A_k - B_k\| \to 0$.

Since convergence under $d_{\mathcal{F}}$ is equivalent to convergence under $d_{\mathcal{M}}$, the two metrics induce the same topology.

### B.1.3.2   Fluctuation moments bound total variation distance

We now show that asymptotically utilizing fluctuation theory to understand mergers is equivalent to probing the similarity of probability distributions using the Total Variation Distance.

**Theorem 5 (Moment–TV inequality)** *Fix an integer $n \geq 2$. Let $p, q$ be probability densities on $\mathbb{R}$ that*

  *(i)  are of bounded variation;*

  *(ii)  admit centred moments $\widehat{\mu}_p^{(k)}, \widehat{\mu}_q^{(k)}$ for $k = 1, \ldots, n+1$;*

  *(iii)  obey the moment proximity bound $|\widehat{\mu}_p^{(k)} - \widehat{\mu}_q^{(k)}| \leq M$ for $k = 1, \ldots, n$;*

  *(iv)  satisfy the uniform first– and second–moment bound $|\widehat{\mu}_{\bullet}^{(1)}|, \widehat{\mu}_{\bullet}^{(2)} \leq B$;*

  *(v)  have matching tails: $\displaystyle\lim_{R \to \infty} \int_{|x| > R} (p - q) = 0$.*

*Then*

$$d_{\mathrm{TV}}(p, q) \ \leq \ C_n\big(M^2 + B\big),$$

*where one may take $C_n = c_0\,(1 + n!)\,(2^n + 48)$ and $c_0 > 0$ is an absolute constant.*

*Proof. 1. Characteristic–function bound.* To connect the proposition's conditions to the proof, we work with the characteristic functions (CFs) of the *centered* random variables. Let $f_p(t)$ and $f_q(t)$ denote these adjusted CFs. Since $p, q$ are absolutely continuous and of bounded variation, $f_p, f_q$ are bounded by $1$ and possess derivatives up to order $n+1$ at the origin. Write the order–$n$ Taylor expansions with integral remainder

$$f_p(t) = \sum_{k=0}^{n} \frac{(\mathrm{i}t)^k}{k!}\mu_p^{(k)} + \frac{(\mathrm{i}t)^{n+1}}{n!}\int_0^1 (1-s)^n \mu_p^{(n+1)}\, e^{\mathrm{i}stX}\, ds.$$

Subtract the analogous expression for $f_q$, use hypothesis (iii), and take absolute values and the standard property of integrals:

$$|f_p(t) - f_q(t)| \leq \sum_{k=1}^{n} \frac{|t|^k}{k!}\, M + \frac{|t|^{n+1}}{(n)!}\int_0^1 (1-s)^n \left|\widehat{\mu}_p^{(n+1)} - \widehat{\mu}_q^{n+1} e^{\mathrm{i}stX}\right|\, ds.$$

Now, use the triangle inequality and note that the integral is a simple weighting integral this gives,

$$|f_p(t) - f_q(t)| \leq \sum_{k=1}^{n} \frac{|t|^k}{k!}\, M + \frac{|t|^{n+1}}{(n+1)!}\big(\widehat{\mu}_p^{(n+1)} + \widehat{\mu}_q^{(n+1)}\big). \tag{B.3}$$

*2. Bounding the $(n+1)$st moments.* By Jensen's inequality, $\widehat{\mu}_{\bullet}^{(n+1)} \leq \big(\widehat{\mu}_{\bullet}^{(2)}\big)^{(n+1)/2} \leq B^{(n+1)/2}$. Insert this in (B.3) to get

$$|f_p(t) - f_q(t)| \leq a_n M |t| + b_n B^{(n+1)/2} |t|^{n+1}, \tag{$A_n$}$$

where $a_n := \sum_{k=1}^{n} \frac{1}{k!}$, $b_n := \frac{2}{(n+1)!}$.

*3. Esseen's smoothing inequality.* For any $T > 0$ (Ibragimov., 1975, Thm. 1.5.4),

$$d_{\mathrm{TV}}(p, q) \ \leq \ \frac{1}{2\pi}\int_{-T}^{T} \left|\frac{f_p(t) - f_q(t)}{t}\right| dt + \frac{24}{\pi T}\big(\mathrm{Var}(p) + \mathrm{Var}(q)\big). \tag{B.4}$$

*Integral term:* divide ($A_n$) by $|t|$ and integrate,

$$\frac{1}{2\pi} \int_{-T}^{T} \left| \frac{f_p - f_q}{t} \right| \le a_n M T + \frac{b_n}{n+1} B^{(n+1)/2} T^{n+1}.$$

*Variance term:* hypothesis (iv) yields $\mathrm{Var}(p), \mathrm{Var}(q) \le B + M^2$, so the second term in (B.4) is bounded by $48 (B + M^2)/(\pi T)$.

*4. Choice of $T$.* Set $T = 1$. (A different $T$ only rescales the constant.) The bounds become

$$d_{\mathrm{TV}}(p, q) \le [a_n + b_n] M + [a_n + b_n] B^{(n+1)/2} + \frac{48}{\pi} (B + M^2)$$
$$\le C_n (M^2 + B),$$

where the last line uses $M \le M^2 + 1$ and $B^{(n+1)/2} \le 2^n B$ for $B \ge 1$, and absorbs all numeric factors into $C_n = c_0 (1 + n!) (2^n + 48)$ with a universal $c_0$.

**Remark 6** *If $p, q$ are sub-Gaussian (or sub-exponential) [Vershynin, 2018], all moments exist and satisfy $\mu_p^{(k)} = O\big((\sqrt{B})^k\big)$; the conditions of Proposition 5 are then automatically satisfied on $\mathbb{R}^d$.*

**Generalisation to multivariate distributions and moment tensors.** The logic of Theorem 5 extends to multivariate distributions on $\mathbb{R}^d$. As established in Appendix B.1.2, the derivatives of the multivariate characteristic function $\varphi_{\boldsymbol{\rho}}(\mathbf{t})$ at the origin generate the moment tensors. A multivariate version of Esseen's inequality bounds the TV distance by an integral over the difference of characteristic functions, which is in turn bounded by the norm of the difference between moment tensors, $\|\mathbb{E}_p[\mathcal{F}_{\rho}^{(k)}] - \mathbb{E}_q[\mathcal{F}_{\rho}^{(k)}]\|_{\mathcal{H}_k}$.

### B.1.4 Equivalence in continuous and discrete state spaces

We recall (and slightly adapt) the terminology of Aldous and Fill [2002], Levin and Peres [2017] so that it aligns with the notation used in the main text and provides a foundation for generalizing concepts to continuous processes.

**Single chain and mixing time.** Let $\mathcal{S} = \{S_0, S_1, \dots\}$ be the marginal sequence of an *ergodic* Markov chain on a finite state space $X$. Its transition matrix is $\Pi \in [0, 1]^{X \times X}$ and the (unique) stationary distribution satisfies $\Pi S_\infty = S_\infty$. For two probability vectors $p, q$ on $X$, the *total-variation* (TV) distance is

$$d_{\mathrm{TV}}(p, q) = \frac{1}{2} \sum_{x \in X} |p(x) - q(x)| = \sup_{A \subseteq X} |P(A) - Q(A)|.$$

The $\varepsilon$*–mixing time* of the chain is the first time the distribution is $\varepsilon$-close to stationarity:

$$t_{\mathrm{mix}}(\varepsilon) = \min\{t \ge 0 : d_{\mathrm{TV}}(S_t, S_\infty) \le \varepsilon\}. \tag{B.5}$$

**Multiple chains and coupling time.** Let $\Lambda$ be a finite index set. For every $\lambda \in \Lambda$, fix an initial event $\Omega_{\lambda,0} \subseteq X$ partitioning the state space. Run the *same* transition matrix on each event to obtain a collection of conditional chains $\mathcal{S}_\lambda = \{S_{\lambda,t}\}_{t \ge 0}$, where $S_{\lambda,t}$ is the law of the chain at time $t$ conditioned on starting in $\Omega_{\lambda,0}$. Because the dynamics are identical, all chains converge to the same stationary distribution $S_\infty$, but their finite-time marginals differ. The *coupling time* is the first time all conditional chains become $\varepsilon$-close to each other:

$$t_{\mathrm{cpl}}(\varepsilon) = \min\Big\{t \ge 0 : \max_{\alpha, \beta \in \Lambda} d_{\mathrm{TV}}(S_{\alpha,t}, S_{\beta,t}) \le \varepsilon\Big\}. \tag{B.6}$$

Hence, $t_{\mathrm{cpl}}$ measures when all initial subpopulations have effectively merged in distribution.

**Generalisation to continuous state spaces.** The total-variation distance is ill-suited for comparing distributions on continuous state spaces like $\mathbb{R}^d$, where it is often trivially 1 unless the distributions have overlapping singular parts [Bhattacharyya et al., 2024, Tao et al., 2024]. Our fluctuation-based metric provides a natural generalisation.

As established in Section B.1.3.2, our cross-fluctuation statistic $\mathcal{M}_\rho^{(n)}$ is intimately linked to the similarity of distributions via their moment structures. We can therefore define a *generalised coupling time* using this measure of structural similarity:

$$t_{\text{gen}}^{(n)}(\varepsilon) \;=\; \min\Big\{t \geq 0 : \min_{\alpha \neq \beta} |\mathcal{M}_\rho^{(n)}(\Omega_{\alpha,t}, \Omega_{\beta,t})| \geq 1 - \varepsilon\Big\}.$$

This definition measures the first time $t$ at which the least-related pair of events $(\alpha, \beta)$ has achieved a structural similarity of at least $1 - \varepsilon$. It directly generalises the discrete coupling time to continuous diffusion processes. For our main application with $n = 2$, this provides a practical tool for tracking when the covariance structures of different event distributions have merged.

## B.2 Application and analysis in diffusion models

### B.2.1 From the empirical reverse process to the learned sampler

Our justification for analyzing the forward process hinges on its tight correspondence with the learned reverse sampler. This connection is forged during training, where the diffusion model minimises the simplified ELBO, an objective that intuitively trains the model $f_\theta$ to predict and reverse the noise added during the forward process.

*simplified ELBO*:

$$\mathcal{L}_{\text{simple}} = \mathbb{E}_{\substack{i \sim \text{Unif}\{0,\ldots,T-1\} \\ \mathbf{x}_0 \sim p_0,\, \varepsilon \sim \mathcal{N}(0,I)}} \big\| f_\theta\big(\mathbf{x}_i(\mathbf{x}_0, \varepsilon),\, i\big) - \varepsilon \big\|_2^2, \tag{B.7}$$

where $\mathbf{x}_i(\mathbf{x}_0, \varepsilon) = \sqrt{\alpha_i}\, \mathbf{x}_0 + \sqrt{1 - \alpha_i}\, \varepsilon \sim p_i$ (cf. Eq. (2.3)).

While practical training results in a non-zero error $\mathcal{L}_{\text{simple}} > 0$, a key theoretical result guarantees that the learned sampler's trajectory remains faithful to the true process.

**Theorem 7 (Chen et al., 2022a, Thm. 1)** *Under standard regularity conditions, if the score error $\varepsilon_\star$ is bounded and a sufficient number of reverse steps are taken, then for all steps $i$:*

$$\|p_i^{rev} - \hat{p}_i\|_{\text{TV}} \;\leq\; \varepsilon,$$

*where $p_i^{rev}$ is the marginal of the true reverse process and $\hat{p}_i$ is the marginal produced by the learned sampler $f_\theta$.*

The power of this result is its implication: closeness in total variation guarantees that all statistics, including the cross-fluctuation metrics we use, are also close. This validation allows us to use the computationally efficient forward process to identify merger times $(i^\star)$, confident that these critical points directly correspond to observable behaviors in a well-trained reverse sampler.

### B.2.2 Pull-back of data events along the PF-ODE

Our method tracks the evolution of distinct subpopulations, or *events*, through the forward diffusion process. For the deterministic probability-flow ODE (Eq. (2.2)), this evolution is governed by a well-behaved flow map.

Let $\Phi_{s \to t} : \mathbb{R}^d \to \mathbb{R}^d$ be the *flow map* of the PF-ODE. Because the drift field is globally Lipschitz under standard assumptions, $\Phi_{s \to t}$ is a bijection for every $0 \leq s < t \leq T$. The continuity equation,

$$\partial_t p_t + \nabla_{\mathbf{x}} \cdot \big(p_t \, \mathbf{v}_t\big) = 0,$$

implies that the time-$t$ marginal $p_t$ is the push-forward of the time-$s$ marginal $p_s$ under the flow map:

$$p_t = (\Phi_{s \to t})_\# p_s, \quad \text{which means} \quad P_t(B) = P_s\big(\Phi_{s \to t}^{-1}(B)\big) \quad \text{for any measurable set } B \subseteq \mathbb{R}^d,$$

where $P_t$ is the probability measure associated with the density $p_t$.

**From source events to time-$t$ marginals.** Fix two disjoint source events $\Omega_{1,0}, \Omega_{2,0} \subseteq \text{supp}(p_0)$. We define their images at a later time $t$ by pulling them forward along the flow:

$$\Omega_{k,t} := \Phi_{0 \to t}\big(\Omega_{k,0}\big), \qquad k \in \{1, 2\},\ t \in [0, T].$$

Since $\Phi_{0 \to t}$ is a diffeomorphism, each $\Omega_{k,t}$ is a well-defined measurable set. While the source events $\Omega_{k,0}$ are disjoint, their images $\Omega_{k,t}$ may overlap as the diffusion progresses.

**Event probabilities along the flow.** The push-forward nature of the flow ensures that the probability mass of each event is preserved over time. For any $k \in \{1, 2\}$:

$$P_t\big(\Omega_{k,t}\big) = P_t\big(\Phi_{0 \to t}(\Omega_{k,0})\big) = P_0\big(\Omega_{k,0}\big).$$

This confirms that $\{\Omega_{k,t}\}$ are valid events with constant probability under their respective marginals $p_t$ at every time $t$.

**Monitoring mixing via cross-fluctuations.** Even when the supports of the events, $\Omega_{1,t}$ and $\Omega_{2,t}$, begin to overlap, our cross-fluctuation statistic $\mathcal{M}_\rho^{(n)}(\Omega_{1,t}, \Omega_{2,t})$ remains well-defined as it compares the internal moment structures of the distributions conditioned on these events. A value of $|\mathcal{M}_\rho^{(n)}|$ approaching one marks the moment the two event distributions become structurally indistinguishable, i.e., the merger time $t_{\mathrm{merge}}$.

### B.2.3 Stochastic–flow formulation for the SDE view

Appendix B.2.2 defined $\Omega_{k,t} = \Phi_{0 \to t}^{-1}(\Omega_{k,0})$ via the *deterministic* flow $\Phi_{s \to t}$ of the PF-ODE (2.2). We now show that the same construction works pathwise for the *stochastic* forward SDE

$$d\mathbf{x}_t = -\tfrac{1}{2}\beta(t)\,\mathbf{x}_t\,dt + \sqrt{\beta(t)}\,d\mathbf{w}_t,$$

whose solution map $x_0 \mapsto x_t$ depends on the Wiener path $\omega \in \Omega_{\mathrm{prob}}$.

**Kunita's stochastic flow of diffeomorphisms.** Let $\varphi_{s,t}(\omega, \cdot) \colon \mathbb{R}^d \to \mathbb{R}^d$, $0 \le s \le t \le T$, denote the *Kunita flow* generated by (2.1) [Kunita, 1990, Ch. 4]. For every fixed $\omega$, the map $x \mapsto \varphi_{s,t}(\omega, x)$ is a $C^1$ diffeomorphism and

$$\varphi_{s,u}(\omega, \cdot) = \varphi_{t,u}\big(\omega, \varphi_{s,t}(\omega, \cdot)\big), \qquad 0 \le s \le t \le u \le T.$$

Hence the pathwise inverse $\varphi_{0,t}^{-1}(\omega, \cdot)$ exists almost surely.

Given two disjoint data events $\Omega_{1,0}, \Omega_{2,0} \subset \mathbb{R}^d$ set

$$\Omega_{k,t}(\omega) := \varphi_{0,t}^{-1}\big(\omega, \Omega_{k,0}\big), \qquad k \in \{1, 2\}.$$

The map $\omega \mapsto \mathbb{1}_{\Omega_{k,t}(\omega)}(x)$ is $\mathcal{F}_t$-measurable (Kunita's measurability theorem), so $\Omega_{k,t}$ is a *random closed set*. Its law equals the push-forward of $p_0$ by the SDE:

$$\mathbb{P}\{x_t \in A\} = \mathbb{E}_\omega\,\mathbb{P}\{\varphi_{0,t}(\omega, x_0) \in A\}, \quad x_0 \sim p_0,$$

and we again write $p_t = \mathcal{L}(x_t)$.

**Annealed cross-fluctuations.** Fix $n$. Because $\varphi_{0,t}^{-1}(\omega, \cdot)$ is $C^1$ and $p_0$ has finite $n$-th moments, the pathwise conditional expected fluctuation tensor $\mathbb{E}[\mathcal{F}_\rho^{(n)} | \Omega_{k,t}(\omega)]$ exists. To obtain deterministic statistics, we define the *annealed* conditional expected fluctuation tensor, which we denote $\mathbf{E}_{k,t}^{(n)}$, by averaging its pathwise counterpart over all Wiener paths:

$$\mathbf{E}_{k,t}^{(n)} := \mathbb{E}_\omega\big[\mathbb{E}[\mathcal{F}_\rho^{(n)} | \Omega_{k,t}(\omega)]\big].$$

From this, the annealed normalised cross-fluctuation, $\mathcal{M}_{\mathrm{ann}}^{(n)}$, is the cosine similarity between these deterministic annealed tensors:

$$\mathcal{M}_{\mathrm{ann}}^{(n)}(\Omega_{1,t}, \Omega_{2,t}) := \frac{\langle \mathbf{E}_{1,t}^{(n)}, \mathbf{E}_{2,t}^{(n)} \rangle_{\mathcal{H}_n}}{\|\mathbf{E}_{1,t}^{(n)}\|_{\mathcal{H}_n} \|\mathbf{E}_{2,t}^{(n)}\|_{\mathcal{H}_n}}.$$

Since expectation commutes with the inner products and norms, all fluctuation identities remain valid for these annealed quantities. Thus, all merger-time results proved for the deterministic PF-ODE hold for the SDE in an expected sense, justifying the use of the same algorithms.

### B.2.4 Fluctuations offer fine-probes into the geometry of data

In this section, we illustrate that the analysis of fluctuations on a diffusion process provides a powerful probe into the intrinsic geometry of the data itself. Our main tool for this purpose is the following theorem,

**Theorem 8 (Exponential contraction of fluctuations and MST construction)** *Let $\rho$ be a Lipschitz state operator, such that the components of the fluctuation tensor function $\mathcal{F}_\rho^{(n)}$ are square integrable with respect to the invariant measure $\mu$ of the diffusion (i.e., $\mathcal{F}_\rho^{(n)} \in L^2(\mu, \mathcal{H}_n)$). Let the evolution be governed by a diffusion semigroup $P_t$ that is self adjoint on $L^2(\mu)$ and possesses a spectral gap $\lambda > 0$.*

*Then, for any two initial event distributions $\mu_i, \mu_j$ with densities $h_i, h_j$ with respect to $\mu$, the distance between their conditional expected fluctuation tensors decays exponentially:*

$$\|\mathbb{E}_i[\mathcal{F}_\rho^{(n)}(t)] - \mathbb{E}_j[\mathcal{F}_\rho^{(n)}(t)]\|_{\mathcal{H}_n} \le e^{-\lambda t} \|\mathcal{F}_\rho^{(n)}\|_{L^2(\mu, \mathcal{H}_n)} \|h_i - h_j\|_{L^2(\mu)}.$$

*Consequently, since the right-hand side converges to zero as $t \to \infty$, for any fixed $\varepsilon > 0$, the graph on events with edges between pairs $(\Omega_i, \Omega_j)$ satisfying $\|\mathbb{E}_i[\mathcal{F}_\rho^{(n)}(t))] - \mathbb{E}_j[\mathcal{F}_\rho^{(n)}(t))]\|_{\mathcal{H}_n} \le \varepsilon$ becomes a complete graph for sufficiently large $t$. A complete graph on a finite number of vertices always admits a well-defined minimal spanning tree (MST).*

*Proof.* Before going into the details of the proof, we justify the reasonableness of the theorem's core assumptions in the context of a $d$ dimensional Variance-Preserving (VP) SDE used in diffusion models.

1. **Existence of an invariant measure** $\mu$: The theorem assumes a stationary distribution $\mu$, which is clearly satisfied. This distribution for our case is $\mathcal{N}(\mathbf{0}_d, \mathbf{I}_{d \times d})$

2. **Self adjointness of the semigroup** $P_t$: The theorem critically relies on the self adjointness of the semigroup operator $P_t$ on the Hilbert space $L^2(\mu)$. The VP-SDE process is a **reversible** process with respect to its Gaussian invariant measure $\mu$. A fundamental result in the theory of Markov processes is that reversibility of a process with respect to a measure $\mu$ is equivalent to its generator being self-adjoint in $L^2(\mu)$, which in turn implies the self adjointness of the associated semigroup [Bakry et al., 2014].

3. **Existence of a spectral gap** $\lambda > 0$: This is the crucial assumption providing the exponential decay rate. The VP-SDE process is a textbook example of a system with a spectral gap.

4. **Square integrability of the fluctuation tensor** $\mathcal{F}$: The proof requires the norm $\|\mathcal{F}_\rho^{(n)}\|_{L^2(\mu, \mathcal{H}_n)}$ to be finite. The invariant measure $\mu$ is Gaussian, meaning its density decays extremely rapidly (exponentially in $\|x\|^2$). If the state operator $\rho$ is Lipschitz (a mild regularity condition), the components of the fluctuation tensor $\mathcal{F}_\rho^{(n)}$ will be polynomials in the state variables. Any polynomial function is square integrable with respect to a Gaussian measure, making this assumption easily satisfied.

In summary, the VP-SDE process is a prime example for which all assumptions of the theorem hold, making the conclusion of exponential convergence robust and directly applicable. We now head towards the main proof.

The proof relies on the consequences of the spectral gap property of the diffusion semigroup $P_t$. Let $\mathcal{F} \equiv \mathcal{F}_\rho^{(n)}$ denote the tensor-valued function on the state space. The difference in conditional expectations at time $t$, for initial distributions $\mu_i, \mu_j$ with densities $h_i, h_j$ with respect to the invariant measure $\mu$, is a vector in the Hilbert space $\mathcal{H}_n$:

$$D_{ij}(t) := \mathbb{E}_i[\mathcal{F}(t)] - \mathbb{E}_j[\mathcal{F}(t)] = \int (P_t \mathcal{F})(x)(h_i(x) - h_j(x)) \, d\mu(x).$$

This is a Bochner integral of an $\mathcal{H}_n$-valued function against a scalar function. The semigroup $P_t$ is assumed to be self adjoint on the Hilbert space $L^2(\mu)$ of scalar functions. This property extends to the space of tensor-valued functions $L^2(\mu, \mathcal{H}_n)$. We can thus move the action of the semigroup from the function $\mathcal{F}$ to the density difference $(h_i - h_j)$:

$$D_{ij}(t) = \int \mathcal{F}(x) \left( P_t(h_i - h_j) \right)(x) \, d\mu(x).$$

We bound the norm of this integral using the Cauchy-Schwarz inequality for Bochner integrals, followed by the standard Cauchy-Schwarz inequality:

$$\|D_{ij}(t)\|_{\mathcal{H}_n} \leq \int \|\mathcal{F}(x)\|_{\mathcal{H}_n} |(P_t(h_i - h_j))(x)| \, d\mu(x)$$

$$\leq \left( \int \|\mathcal{F}(x)\|_{\mathcal{H}_n}^2 \, d\mu(x) \right)^{1/2} \left( \int |(P_t(h_i - h_j))(x)|^2 \, d\mu(x) \right)^{1/2}$$

$$= \|\mathcal{F}\|_{L^2(\mu,\mathcal{H}_n)} \|P_t(h_i - h_j)\|_{L^2(\mu)}.$$

The existence of a spectral gap $\lambda > 0$ is equivalent to the following contraction property on the space of mean-zero functions in $L^2(\mu)$: for any $g \in L^2(\mu)$ with $\int g \, d\mu = 0$, we have $\|P_t g\|_{L^2(\mu)} \leq e^{-\lambda t} \|g\|_{L^2(\mu)}$. Since $h_i$ and $h_j$ are probability densities, their difference $g = h_i - h_j$ has zero mean. Applying the contraction property yields:

$$\|\mathbb{E}_i[\mathcal{F}(t)] - \mathbb{E}_j[\mathcal{F}(t)]\|_{\mathcal{H}_n} \leq e^{-\lambda t} \|\mathcal{F}\|_{L^2(\mu,\mathcal{H}_n)} \|h_i - h_j\|_{L^2(\mu)}.$$

The term $\|\mathcal{F}\|_{L^2(\mu,\mathcal{H}_n)} \|h_i - h_j\|_{L^2(\mu)}$ is a constant for any fixed pair of events. The exponential term $e^{-\lambda t}$ guarantees that the distance converges to zero as $t \to \infty$. Therefore, for any threshold $\varepsilon > 0$, there exists a time $T$ such that for all $t > T$, the distance is less than $\varepsilon$ for all pairs $(i,j)$, making the corresponding graph complete. An MST can always be constructed on a weighted complete graph using standard algorithms such as Kruskal's [Kruskal, 1956].

This exponential convergence justifies why we expect mergers to occur and provides a geometric interpretation of the process: initially distant event structures are pulled together until they collapse into a single point in the space of moment tensors. The above geometric perspective on the evolution of event structures has two key implications:

1. **Hierarchical refinement.** We can progressively refine our analysis by choosing finer partitions of the initial state space. Starting with broad categories (e.g., animals vs. vehicles), we can track their mergers, then move to finer sub-partitions (e.g., cats vs. dogs) and track their subsequent mergers. This allows for probing the system's dynamics at increasingly fine resolutions, all on the same underlying data distribution.

2. **Connection to manifold learning.** Tracking the evolution of the graph $G_t$ on events (where edge weights are given by the distance $\|\mathbb{E}_i[\mathcal{F}_\rho^{(n)}] - \mathbb{E}_j[\mathcal{F}_\rho^{(n)}]\|_{\mathcal{H}_n}$) is conceptually similar to algorithms that build neighborhood graphs to learn low-dimensional embeddings. Methods like t-SNE [van der Maaten and Hinton, 2008] and UMAP [McInnes et al., 2018] also rely on connecting nearby points (or neighborhoods) to reveal underlying manifold structure. Our framework can be seen as applying a similar principle in the time domain, tracking how neighborhoods of events connect and merge as the diffusion process evolves. Exploring this connection in more detail is a promising direction for future work.

### B.2.5 Unbiased monte-carlo estimators for cross fluctuations

**Theorem 9 (One-sweep unbiasedness)** *Let $\Omega_{1,0}, \Omega_{2,0} \subseteq \Omega$ be disjoint events with probabilities $p_1, p_2 > 0$. Simulate once $N$ i.i.d. forward trajectories $\{\mathbf{x}_t^{(i)}\}_{t=0}^T$ from the VP process. Let $Z_k^{(i)} := \mathbb{1}_{\Omega_{k,0}}(\mathbf{x}_0^{(i)})$.*

*First, we form an unbiased estimator for the conditional expected fluctuation tensor itself:*

$$\widehat{\mathbf{E}}_{k,t}^{(n)} := \frac{1}{N p_k} \sum_{i=1}^N Z_k^{(i)} \mathcal{F}_\rho^{(n)}(\mathbf{x}_t^{(i)}), \qquad k \in \{1, 2\}, \tag{B.8}$$

*where $\mathcal{F}_\rho^{(n)}(\mathbf{x}_t^{(i)})$ is the random fluctuation tensor for sample $i$ at time $t$. This estimator is an average of tensors and is itself a tensor in $\mathcal{H}_n$.*

*Next, we define plug-in estimators for the unnormalised cross-fluctuation $G_\rho^{(n)}$ and the within-event fluctuation magnitude $\widehat{F}_\rho^{(2n)}$ based on these tensor estimators:*

$$\widehat{G}_\rho^{(n)}(\Omega_{1,t}, \Omega_{2,t}) := \langle \widehat{\mathbf{E}}_{1,t}^{(n)}, \widehat{\mathbf{E}}_{2,t}^{(n)} \rangle_{\mathcal{H}_n}, \tag{B.9}$$

$$\widehat{\overline{F}}_\rho^{(2n)}(\Omega_{k,t}) := \|\widehat{\mathbf{E}}_{k,t}^{(n)}\|_{\mathcal{H}_n}^2. \tag{B.10}$$

The tensor estimator $\widehat{\mathbf{E}}_{k,t}^{(n)}$ is unbiased for the true conditional expected fluctuation tensor $\mathbb{E}_k[\mathcal{F}_\rho^{(n)}(\Omega_{k,t})]$. Consequently, the plug-in ratio estimator for the normalised cross-fluctuation,

$$\widehat{\mathcal{M}}_\rho^{(n)}(t) := \frac{\widehat{G}_\rho^{(n)}(\Omega_{1,t}, \Omega_{2,t})}{\sqrt{\widehat{\widehat{F}}_\rho^{(2n)}(\Omega_{1,t})\,\widehat{\widehat{F}}_\rho^{(2n)}(\Omega_{2,t})}} = \frac{\langle \widehat{\mathbf{E}}_{1,t}^{(n)}, \widehat{\mathbf{E}}_{2,t}^{(n)}\rangle_{\mathcal{H}_n}}{\|\widehat{\mathbf{E}}_{1,t}^{(n)}\|_{\mathcal{H}_n}\|\widehat{\mathbf{E}}_{2,t}^{(n)}\|_{\mathcal{H}_n}},$$

is consistent and asymptotically unbiased for the true value $\mathcal{M}_\rho^{(n)}(\Omega_{1,t}, \Omega_{2,t})$:

$$\mathbb{E}\big[\widehat{\mathcal{M}}_\rho^{(n)}(t)\big] = \mathcal{M}_\rho^{(n)}(\Omega_{1,t}, \Omega_{2,t}) + O(N^{-1}).$$

*Proof.* The proof proceeds in three steps, starting with the unbiasedness of the core tensor estimator.

*Step 1: Unbiasedness of the tensor estimator $\widehat{\mathbf{E}}_{k,t}^{(n)}$.* Fix $k \in \{1, 2\}$. By the linearity of expectation, we can move the expectation inside the sum:

$$\mathbb{E}_{\mathbb{P}}[\widehat{\mathbf{E}}_{k,t}^{(n)}] = \frac{1}{Np_k} \sum_{i=1}^N \mathbb{E}_{\mathbb{P}}\Big[Z_k^{(i)} \mathcal{F}_\rho^{(n)}(\mathbf{x}_t^{(i)})\Big].$$

For each term in the sum, we condition on the starting point $\mathbf{x}_0^{(i)}$:

$$\mathbb{E}_{\mathbb{P}}\Big[Z_k^{(i)} \mathcal{F}_\rho^{(n)}(\mathbf{x}_t^{(i)})\Big] = \int_{\mathbb{R}^d} \mathbb{1}_{\Omega_{k,0}}(x_0) p_0(x_0) \mathbb{E}_{\mathbb{P}}[\mathcal{F}_\rho^{(n)}(x_t)|x_0]\, dx_0$$

$$= p_k \left(\frac{1}{p_k} \int_{\Omega_{k,0}} p_0(x_0) \mathbb{E}_{\mathbb{P}}[\mathcal{F}_\rho^{(n)}(x_t)|x_0]\, dx_0\right)$$

$$= p_k\, \mathbb{E}_k[\mathcal{F}_\rho^{(n)}(\Omega_{k,t})].$$

Summing $N$ identical terms gives $Np_k\, \mathbb{E}_k[\mathcal{F}_\rho^{(n)}(\Omega_{k,t})]$. Dividing by the $Np_k$ prefactor proves that $\mathbb{E}_{\mathbb{P}}\big[\widehat{\mathbf{E}}_{k,t}^{(n)}\big] = \mathbb{E}_k[\mathcal{F}_\rho^{(n)}(\Omega_{k,t})]$.

*Step 2: Consistency of plug-in estimators.* The estimators $\widehat{G}_\rho^{(n)}$ and $\widehat{\widehat{F}}_\rho^{(2n)}$ are continuous functions (inner product and squared norm) of the tensor estimator $\widehat{\mathbf{E}}_{k,t}^{(n)}$. Since $\widehat{\mathbf{E}}_{k,t}^{(n)}$ is an average of i.i.d. random tensors, it is a consistent estimator for its mean by the Law of Large Numbers. By the continuous mapping theorem [Billingsley, 2012], $\widehat{G}_\rho^{(n)}$ and $\widehat{\widehat{F}}_\rho^{(2n)}$ are therefore consistent estimators for $G_\rho^{(n)}$ and $\widehat{F}_\rho^{(2n)}$, respectively.

*Step 3: Bias of the ratio estimator.* The estimator $\widehat{\mathcal{M}}_\rho^{(n)}(t)$ is a smooth function of the components of the estimators $\widehat{\mathbf{E}}_{1,t}^{(n)}$ and $\widehat{\mathbf{E}}_{2,t}^{(n)}$. Each of these components is an average of $N$ i.i.d. random variables. By the multivariate Central Limit Theorem, the joint distribution of these averages converges to a normal distribution. The delta method for ratios of random variables then applies directly. A multivariate second-order Taylor expansion of the cosine similarity function around the true expected tensor values shows that the linear terms in the bias expansion vanish, and the bias is of order $O(N^{-1})$.

**Remark 10** *All computations rely on the same set of $N$ forward trajectories. The primary computational step is to compute and store the random fluctuation tensors $\{\mathcal{F}_\rho^{(n)}(\mathbf{x}_t^{(i)})\}$ for each sample and time step, from which all other quantities can be derived.*

## B.3  Framework connections, extensions, and interpretation

### B.3.1  Mixing time of isotropic Gaussians under Brownian diffusion

We quantify how long the forward VP–SDE (2.1) needs to *forget* an isotropic sub-Gaussian input and become $\varepsilon$-close (in total variation) to its Gaussian limit. Write

$$J(t) = \exp\Big(-\tfrac{1}{2} \int_0^t \beta(s)\, ds\Big),$$

the deterministic attenuation factor from (2.4).

**Proposition 11 (Mixing time for sub-Gaussian data)** *Let* $\mathbf{y} \in \mathbb{R}^d$ *have i.i.d. mean-zero, unit-variance components that are* $\sigma^2$*-sub-Gaussian:* $\Pr(|y_i| > t) \leq 2e^{-t^2/2\sigma^2}$. *Evolve* $\mathbf{y}$ *with the VP–SDE* (2.1) *and denote* $\mathcal{L}(\mathbf{x}_t) = p_t$. *For every* $\varepsilon \in (0,1)$ *the* $\varepsilon$*-mixing time*

$$t_{\mathrm{mix}}(\varepsilon) := \inf\big\{t \geq 0 : d_{\mathrm{TV}}\big(p_t, \mathcal{N}(0, I_d)\big) \leq \varepsilon\big\}$$

*satisfies*

$$J\big(t_{\mathrm{mix}}(\varepsilon)\big) \;\leq\; \frac{2}{d}\Big(1 + O\big(\sigma^2 \log \tfrac{1}{\varepsilon}\big)\Big), \quad \text{and} \quad J\big(t_{\mathrm{mix}}(e^{-1})\big) = \Theta(d^{-1}).$$

*Proof. Step 1: tails of the input.* Sub-Gaussianity yields $\mathbb{E}\|\mathbf{y}\|_2^2 = d$ and $\mathrm{Var}\,\|\mathbf{y}\|_2^2 \leq Cd$ (for $C = C(\sigma)$). Bernstein's inequality Vershynin [2018] gives $\|\mathbf{y}\|_2^2 = d \pm O(\sqrt{d})$ w.h.p.

*Step 2: second moment under the SDE.* Conditioned on $\mathbf{y}$, $\mathbb{E}\|\mathbf{x}_t\|_2^2 = J(t)^2\|\mathbf{y}\|_2^2 + d\big(1 - J(t)^2\big)$, so averaging produces $\mathbb{E}\|\mathbf{x}_t\|_2^2 = d + J(t)^2 O(\sqrt{d})$.

*Step 3: bounding* $\chi^2$. Pinsker's inequality Cover and Thomas [2006] gives $d_{\mathrm{TV}}^2 \leq \frac{1}{2}\chi^2$. For Gaussians with equal means, $\chi^2 = (\det \Sigma)^{-1/2} \exp\big(\frac{1}{2}\mathrm{tr}(I - \Sigma^{-1})\big) - 1$. With $\Sigma = J(t)^2 I_d + (1 - J(t)^2)I_d$,

$$d_{\mathrm{TV}}^2(p_t, \mathcal{N}) \leq \tfrac{1}{2}d\,\frac{J(t)^4}{1 - J(t)^2}\big(1 + O(d^{-1/2})\big).$$

*Step 4: solve for* $J(t)$. Setting the rhs to $\varepsilon^2$ and solving yields the claimed bound.

**Closed form for a linear schedule.** With $\beta(t) = \beta_0 + (\beta_T - \beta_0)t/T$,

$$J(t) = \exp\Big(-\tfrac{1}{2}\beta_0 t - \tfrac{1}{4}(\beta_T - \beta_0)\frac{t^2}{T}\Big).$$

Taking $\varepsilon = e^{-1}$ in Theorem 11, $\log J(t_{\mathrm{mix}}) \simeq -\log d + \log 2$, so $t_{\mathrm{mix}}$ solves a quadratic. For the DDPM defaults $(\beta_0, \beta_T, T) = (10^{-4}, 0.02, 1000)$:

$$t_{\mathrm{mix}} + 0.0995\,t_{\mathrm{mix}}^2 = 5000 \log\big(d/2\big).$$

| Data dimension | Pred. $t_{\mathrm{mix}}/T$ | Obs. $i^\star/T$ |
|---|---|---|
| $3 \times 32 \times 32$ (CIFAR-10) | 0.602 | 0.60 |
| $1 \times 28 \times 28$ (MNIST) | 0.543 | 0.60 |
| $4 \times 32 \times 32$ (ImageNet latents) | 0.614 | 0.70 |

Table 8: Theoretical mixing index vs. empirical convergence index.

Table 8 shows theory versus measured convergence indices; the $\Theta(d^{-1})$ scaling persists on real data.

Thus, it is possible to estimate the mixing time for sub-gaussian data analytically, in fact due to concentration bounds on high-dimensional sub-gaussians Vershynin [2018] it could be shown that the above time manifests physically as a symmetry breaking transition, leading to an estimate consistent with Raya and Ambrogioni [2024] for spherically symmetric distributions and in Biroli et al. [2024] for gaussian mixtures. Interestingly, ImageNet latent representations align closely with these theoretical estimates. We hypothesise that this occurs because the compressed space corresponds to the latent space of a Variational Autoencoder (VAE) trained to approximate a Gaussian distribution, potentially making these latents effectively sub-Gaussian. Verification of this hypothesis and related questions remains future work.

### B.3.2 Higher-order fluctuations as a proof of concept

To make higher-order analysis computationally tractable, we transition from analyzing the full rank-$n$ fluctuation tensor $\mathcal{F}_\rho^{(n)}$ to analyzing the moments of a corresponding scalar-valued random variable. This simplification is equivalent to assuming the components of the state vector are i.i.d. and studying the dynamics of a single component. This bypasses tensor computations and allows us to work with scalar moments.

Under this assumption, we can track the scalar moment identity:

$$\widehat{\mu}_i^{(n)}(t) = J(t)^n \widehat{\mu}_i^{(n)}(0) + \left(1 - J(t)^n\right)\widehat{\mu}_{\mathcal{N}}^{(n)},$$

where $\widehat{\mu}_i^{(n)}(t)$ is the $n$-th centered moment for class $i$ at time $t$. For the Gaussian limit, Isserlis'/Wick's theorem[Isserlis, 1918, Wick, 1950] can be applied because the components are treated as jointly Gaussian. The theorem states that higher-order moments of a Gaussian are polynomials in the variance, and odd moments are zero. This rule lets us draw the higher-order generative diagrams in Figures 3-4. Compared with the second-order diagram, high-order curves fan out more widely at early times—evidence that non-linear features dominate, but they collapse sooner, consistent with the rapid $J(t)^n$ decay.

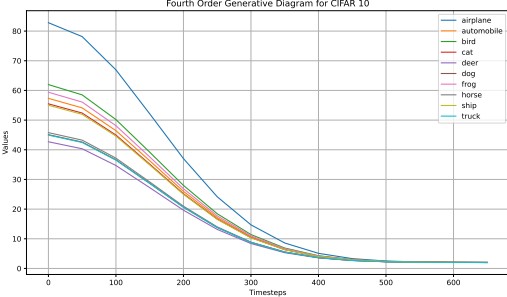
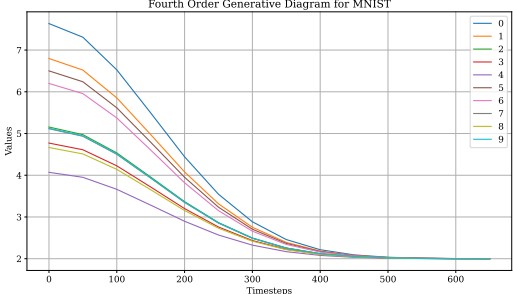

Figure 3: Fourth order generative diagram for CI-FAR10.We show the emergence of classes using fourth-order correlations.

Figure 4: Fourth order generative diagram for MNIST.We show the emergence of classes using fourth-order correlations.

### B.3.3   Centred kernel alignment (CKA)

Kernel alignment measures how similarly two Gram matrices embed the *same* data. For $K, L \in \mathbb{R}^{n \times n}$ the uncentred score is the cosine in $\mathbb{R}^{n^2}$ [Cristianini et al., 2001]:

$$A(K, L) = \frac{\langle K, L \rangle_F}{\|K\|_F \, \|L\|_F}, \quad \langle K, L \rangle_F := \sum_{ij} K_{ij} L_{ij}.$$

Because $A$ reacts to mean shifts, Cortes et al. [2012] introduced *centred kernel alignment*

$$A_{\mathrm{c}}(K, L) := A\big(HKH, HLH\big), \quad H := I_n - \tfrac{1}{n}\mathbf{1}\mathbf{1}^\top.$$

$A_{\mathrm{c}}$ is invariant to any feature-space translation and, for linear kernels, to orthogonal mixing and isotropic scaling.

**CKA for covariance kernels.**   Section 4.2 sets $\tilde{K} = \Sigma_{i,t}, \tilde{L} = \Sigma_{j,t}$. For symmetric matrices

$$\langle \tilde{K}, \tilde{L} \rangle_F = \sum_{m=1}^{d} \lambda_{i,t,m} \, \lambda_{j,t,m}, \tag{B.11}$$

$$\|\tilde{K}\|_F^2 = \sum_{m=1}^{d} \lambda_{i,t,m}^2, \qquad \|\tilde{L}\|_F^2 = \sum_{m=1}^{d} \lambda_{j,t,m}^2, \tag{B.12}$$

where $\{\lambda_{i,t,m}\}$ are the eigenvalues of $\Sigma_{i,t}$ and *mutatis mutandis* for $\{\lambda_{i,t,m}\}$. Under the VP flow they contract as $\lambda_{i,t,m} = \lambda_{i,0,m} J(t)^2 + \left(1 - J(t)^2\right)$ [Biroli et al., 2024]. The largest eigenvalue dominates once $J(t)^2 \ll \lambda_{i,0,2}/\lambda_{i,0,1}$, so

$$A_{\mathrm{c}}\big(\Sigma_{i,t}, \Sigma_{j,t}\big) \uparrow 1 \iff \lambda_{i,t,1} \simeq \lambda_{j,t,1}.$$

Therefore tracking the top eigenvalues $\lambda_{i,t,1}, \lambda_{j,t,1}$ gives an efficient proxy for merger of $\Omega_{i,t}$ and $\Omega_{j,t}$ in Section 4.2.

### B.3.4  Structural regularity bounds fluctuations

This section provides a more rigorous justification for the Fluctuation adaptation lemma from Section 4.5, which posits that structurally similar distributions exhibit similar fluctuation dynamics. The premise is the Fourier regularity condition from Eq. (4.7), stating that the $L^2$ distance between the characteristic functions (CFs) of the source and target-style distributions is small: $\|\widehat{p}_0 - \widehat{p}^\star\|_{L^2} \le \delta$. The challenge is to show that this integral bound on the CFs implies a bound on the fluctuation tensors, which are determined by pointwise derivatives of the CFs at the origin.

The bridge between these two concepts is provided by the *Sobolev Embedding Theorem* [Adams and Fournier, 2003]. This theorem connects a function's *average* smoothness, measured by its Sobolev norm $\|f\|_{H^s}$, to its *pointwise* smoothness, such as the boundedness of its derivatives. However, the Sobolev norm, $\|g\|_{H^s}^2 = \int (1 + |\boldsymbol{\xi}|^2)^s |g(\boldsymbol{\xi})|^2 \, d\boldsymbol{\xi}$, is stronger than the standard $L^2$ norm due to the frequency-weighting factor $(1 + |\boldsymbol{\xi}|^2)^s$. Therefore, a small $L^2$ norm does not automatically guarantee a small Sobolev norm. To bridge this gap, we must introduce a more explicit regularity assumption on the style transfer process itself:

*Assumption:* The style transfer transformation $\mathcal{T}_{\text{style}}$ is assumed to be sufficiently regular such that the difference in characteristic functions, $g(\boldsymbol{\xi}) = \widehat{p}_0(\boldsymbol{\xi}) - \widehat{p}^\star(\boldsymbol{\xi})$, not only has a small $L^2$ norm but also has rapidly decaying high-frequency content. Formally, this means there exists a constant $K$ such that $\|g\|_{H^s} \le K\|g\|_{L^2}$ for the required Sobolev order $s$. This assumption holds if, for instance, the style transfer primarily modifies low-to-mid frequency components of the distribution, a common case for artistic stylisation that preserves core structures.

With this assumption, the argument is complete. The Sobolev Embedding Theorem can now be directly applied:

$$|D^\alpha g(0)| \le C\|g\|_{H^s} \le CK\|g\|_{L^2} \le CK\delta.$$

This step guarantees that the difference between each individual *scalar component* of the corresponding moment tensors is bounded by $O(\delta)$. This component-wise bound is then extended to a norm on the entire fluctuation tensor. The conditional expected fluctuation tensor, $\mathbb{E}_{0,\Omega}[\mathcal{F}_\rho^{(n)}]$, is an element of the Hilbert space $\mathcal{H}_n$ of rank-$n$ tensors. Let $\Delta_{\mathcal{F}} = \mathbb{E}_{0,\Omega}[\mathcal{F}_\rho^{(n)}] - \mathbb{E}_{\star,\Omega^\star}[\mathcal{F}_\rho^{(n)}]$ be the difference tensor. The squared Frobenius norm of this tensor is the sum of the squared magnitudes of its components: $\|\Delta_{\mathcal{F}}\|_{\mathcal{H}_n}^2 = \sum_{\mathbf{j}} |(\Delta_{\mathcal{F}})_{\mathbf{j}}|^2$. Since we established that each component is bounded, $|(\Delta_{\mathcal{F}})_{\mathbf{j}}| \le C_{\mathbf{j}}\delta$, the norm of the full tensor is also bounded:

$$\left\| \mathbb{E}_{0,\Omega}[\mathcal{F}_\rho^{(n)}] - \mathbb{E}_{\star,\Omega^\star}[\mathcal{F}_\rho^{(n)}] \right\|_{\mathcal{H}_n} \le C_n'\delta.$$

Finally, since the normalised cross-fluctuation $\mathcal{M}_\rho^{(n)}$ is defined as the cosine of the angle between these tensors in the Hilbert space $\mathcal{H}_n$, it is a continuous function of its tensor arguments [Conway, 1990]. Therefore, a small perturbation in the input tensors, bounded by $O(\delta)$, leads to a correspondingly small change in the value of $\mathcal{M}_\rho^{(n)}$. This ensures that the merger times are stable under the stylistic transformation, rigorously justifying the use of a schedule computed on the source distribution for the zero-shot style transfer task.

### B.3.5  Extending the framework to certain non Markovian samplers

For a *non Markovian* latent chain the marginal $p_i$ depends on the entire future tail $\{p_{i+1}, p_{i+2}, \ldots, p_n\}$. Hence the pull–back $\Omega_{k,0} \mapsto \Omega_{k,i}$ is well defined only if every conditional kernel beyond step $i$ is known. This hurdle disappears when the latent family belongs to a *natural exponential family* (NEF).

**Tail statistic Markovisation.**  An NEF on $\mathbb{R}^d$ has densities $p_{\boldsymbol{\theta}}(x) = h(x)\exp\big(\langle \boldsymbol{\theta}, T(x)\rangle - A(\boldsymbol{\theta})\big)$, with sufficient statistic $T$ and log-partition function $A$. For an *independent* sequence $\{X_i\}_{i=1}^n$ drawn from an NEF, define the *tail statistic*

$$G_i := \sum_{t=i}^n T(X_t), \qquad i = 1, \ldots, n.$$

The Pitman–Koopman–Darmois theorem gives

**Theorem 12 (Tail statistic Markov property Pitman, 1936)**

(i) *If $\{X_i\}$ are i.i.d. from an NEF, the conditional sequence $\{\mathcal{L}(X_i \mid G_i)\}_{i=1}^{n}$ is first-order Markov.*

(ii) *Conversely, if a statistic sequence $\{G_i\}$ makes $\{\mathcal{L}(X_i \mid G_i)\}$ Markov for* all *$n$, then the marginals must form an NEF.*

Thus the random vector $G_i$ captures *all* future information relevant at step $i$.

**Injecting the tail statistic into fluctuations.** Fix disjoint initial events $\Omega_{1,0}, \Omega_{2,0}$. Condition on $G_i = g$ and apply the deterministic PF-ODE of Appendix B.2.2 *inside the fibre* $\{X_i \mid G_i = g\}$:

$$\Omega_{k,i}(g) := \left\{ x \in \mathbb{R}^d : \varphi_{0 \to i}^{-1}(g, x) \in \Omega_{k,0} \right\}, \qquad k \in \{1, 2\}.$$

Because the conditioned kernels are Markov (Theorem 12), this construction mirrors the purely Markovian case. The pathwise cross-fluctuation $\mathcal{M}_\rho^{(n)}(\Omega_{1,i}(g), \Omega_{2,i}(g))$ is the cosine similarity between the conditional expected fluctuation tensors for a given $g$. Averaging over the law of $G_i$ yields the *annealed* cross-fluctuation:

$$\overline{\mathcal{M}}_\rho^{(n)}(\Omega_{1,i}, \Omega_{2,i}) := \int \mathcal{M}_\rho^{(n)}(\Omega_{1,i}(g), \Omega_{2,i}(g)) \, d\mathbb{P}_{G_i}(g).$$

Every algebraic identity from the main framework now carries over to this tail-averaged counterpart. A merger is detected when $|\overline{\mathcal{M}}_\rho^{(n)}| \to 1$. Star-DDPM Okhotin et al. [2023] is a recent work that uses a similar formulation to obtain non Markovian diffusion generative models.

Thus, whenever a non Markovian diffusion admits a finite-dimensional *tail statistic*—a property guaranteed for exponential-family latents—conditioning on that statistic restores the Markov property and lets the fluctuation framework operate unchanged. Identifying broader classes of tail markovizable samplers is a promising direction for future work.

### B.3.6 Phases of diffusion model dynamics

We analyze diffusion model dynamics by distinguishing between two types of phase transitions. Our approach is inspired by discretisation based analyses in physics [Kogut, 1983], but we treat the system's discrete nature, its time steps and finite precision—as a fundamental property, not a computational artifact. This leads us to differentiate between:

(1) **Thermodynamic.** A discontinuity in the $n$-th classical derivative of a system-wide property, $(\Phi^{(n)})$, at some time $t_0$ (assuming $(\Phi \in C^n)$). Such transitions, studied in the continuum limit, are typically *exclusive*, meaning at most one can occur for the whole system at a given time [Biroli et al., 2024].

(2) **Lattice.** For a given step size $\tau > 0$, if there exists a discontinuity threshold $\varrho_{\text{disc}} > 0$ such that for the $n$-th finite differences from the left $(\text{LD}_\tau^{(n)} \Phi)$ and right $(\text{RD}_\tau^{(n)} \Phi)$, the below inequality holds, then the observable $\Phi$ undergoes a *lattice* transition at step $t_0$ of *order* $n$.

$$\|\text{LD}_\tau^{(n)} \Phi(t_0) - \text{RD}_\tau^{(n)} \Phi(t_0)\| \ge \varrho_{\text{disc}}.$$

Lattice transitions include thermodynamic ones in the limit $\tau \to 0$ but are generally *non-exclusive*.

Our core observable for detecting these transitions is the absolute normalised cross-fluctuation, $|\mathcal{M}_\rho^{(n)}|$. This quantity measures the cosine similarity of the conditional expected fluctuation tensors of two evolving events. A merger signifies that these tensors have aligned, i.e., $|\mathcal{M}_\rho^{(n)}| \to 1$. Our merger time $i^\star$ is precisely a lattice transition point for this observable, defined by our chosen *merger distance threshold $\varepsilon$* (as in Eq. (3.1)).

**Theorem 13** *The merger event detected by $\tilde{\mathcal{M}}_\rho^{(n)}$ as defined by our* merger distance threshold $\varepsilon$ *in Eq. (3.1) constitutes a lattice transition at the merger time $i^\star$.*

*Proof.* Consider two initially disjoint events $\Omega_{1,0}, \Omega_{2,0} \subset \Omega_0$. Let $i^\star$ denote the merger time given by Eq. (3.1). From the topological equivalence proved in Theorem 4, there exists a constant $\vartheta > 0$ such that at the merger,

$$\left| |\mathcal{M}_\rho^{(n)}(\Omega_{1,i^\star}, \Omega_{2,i^\star})| - 1 \right| \le \vartheta.$$

By selecting a suitable *merger distance threshold* $\varepsilon > 0$ in the metric $d_n(\cdot, \cdot)$ used in Eq. (3.1), we can ensure $\vartheta < 1/2$. Let our observable be $\Phi(i) = |\mathcal{M}_\rho^{(n)}(\Omega_{1,i}, \Omega_{2,i})|$. It follows that $\Phi(i^\star) \geq 1 - \vartheta$. Due to the hypercontractivity of the underlying Brownian motion, the structural similarity between events is monotonically increasing, hence $\Phi(i)$ is monotonically increasing in $i$. This monotonicity implies that for a discrete step size $\tau$, the rate of approach to the merger is bounded. For the $n$-th order finite differences, this can be shown to satisfy:

$$\mathrm{RD}_\tau^{(n)}\big(\Phi(i^\star)\big) \leq \frac{\vartheta}{\tau^n}, \quad \text{and} \quad \mathrm{LD}_\tau^{(n)}\big(\Phi(i^\star)\big) \geq \frac{1 - \vartheta - \Phi(i^\star - n\tau)}{\tau^n}.$$

The key insight is that the jump into the merged state is a finite, discrete event. Applying the reverse triangle inequality to the difference in the finite derivatives gives:

$$\big\| \mathrm{LD}_\tau^{(n)}(\Phi(i^\star)) - \mathrm{RD}_\tau^{(n)}(\Phi(i^\star)) \big\| > 0,$$

which establishes the existence of a lattice transition at the merger time, by demonstrating a non-zero jump in the finite differences, which can be thresholded by a chosen $\varepsilon_{\mathrm{disc}}$. Notice that as this relies on finite $\tau$ and $\vartheta$, this transition is not necessarily thermodynamic.

**Thermodynamic phases from prior works.** For class-conditioned VP diffusion, Biroli et al. [2024] proved two thermodynamic boundaries $t_{\mathrm{u} \to \mathrm{s}}$ (unbiased $\to$ speciation) and $t_{\mathrm{s} \to \mathrm{c}}$ (speciation $\to$ condensation):

$$\text{unbiased } [0, t_{\mathrm{u} \to \mathrm{s}}) \subset \text{speciation } (t_{\mathrm{u} \to \mathrm{s}}, t_{\mathrm{s} \to \mathrm{c}}) \subset \text{condensation } (t_{\mathrm{s} \to \mathrm{c}}, T].$$

**Relation of class conditional lattice mergers to thermodynamic phases.** For two classes $k \neq \ell$ define the centred cross-fluctuation $\mathcal{M}_{k\ell}(t)$ ((4.5) in Section 4.2). Its $\varepsilon$-merger time is

$$t_{k\ell}^{\mathrm{lat}}(\varepsilon) := \inf\{t \geq 0 : \mathcal{M}_{k\ell}(t) \geq 1 - \varepsilon\}, \qquad \varepsilon \in (0, 1).$$

**Lemma 14 (Merger times lie inside the speciation phase)** *For all $k \neq \ell$ and $\varepsilon \in (0, 1)$,*

$$t_{\mathrm{u} \to \mathrm{s}} < t_{k\ell}^{\mathrm{lat}}(\varepsilon) \leq t_{\mathrm{s} \to \mathrm{c}}.$$

*Proof. Unbiased phase.* If $t < t_{\mathrm{u} \to \mathrm{s}}$ then $p_{k,t} = p_{l,t}$, so $\mathcal{M}_{k\ell}(t) = 1$; no upward crossing can occur.

*Condensation phase.* For $t > t_{\mathrm{s} \to \mathrm{c}}$ each covariance (4.3) satisfies $\lambda_{\max}(\Sigma_{k,t}) \leq e^{-c(t - t_{\mathrm{s} \to \mathrm{c}})}$ [Biroli et al., 2024, Prop. 4]. Using (B.12), $1 - \mathcal{M}_{k\ell}(t) = O(e^{-c(t - t_{\mathrm{s} \to \mathrm{c}})})$; hence $\mathcal{M}_{k\ell}(t) \geq 1 - \varepsilon$ for *all* large $t$. No new crossing can start after $t_{\mathrm{s} \to \mathrm{c}}$.

*Speciation phase.* Because a crossing cannot start before $t_{\mathrm{u} \to \mathrm{s}}$ or after $t_{\mathrm{s} \to \mathrm{c}}$, any merger time must lie in $(t_{\mathrm{u} \to \mathrm{s}}, t_{\mathrm{s} \to \mathrm{c}}]$.

**The phase spectrum.** Figure 5(a) details the spectrum of phases revealed by our lattice approach. For any choice of $\epsilon > 0$, the system can be characterised by the number of merger events required to connect all events in a partition. The plot visualises a sequence of distinct states:

i. **Unrealizable thermodynamic phase $\epsilon = 0$:** An unattainable limit which admits only a single possible merger at $t \to \infty$.

ii. **Unrealizable phase ($0 < \epsilon < \epsilon_{min}$):** Below the computational precision limit, a maximal number of $K$ potential mergers exist, but they are not observable.

iii. **Realizable plateau ($\epsilon_{min} \leq \epsilon < \epsilon_{start}$):** The system is observable and the threshold allows for maximum number of mergers possible.

iv. **Staircase phase ($\epsilon_{start} \leq \epsilon < \epsilon_{end}$):** As $\epsilon$ increases, it becomes less sensitive to the structural differences between events. The single macro-state of "maximum potential mergers" splits into a spectrum of fine-grained states, visualised as a downward staircase. Each step is a lattice transition where the number of distinct merger events required to connect the class graph decreases.

v. **Saturated phase ($\epsilon \geq \epsilon_{end}$):** The threshold is so large that a single overarching rule merges all events. The system becomes a singleton for discrimination purposes, losing all distinguishing power.

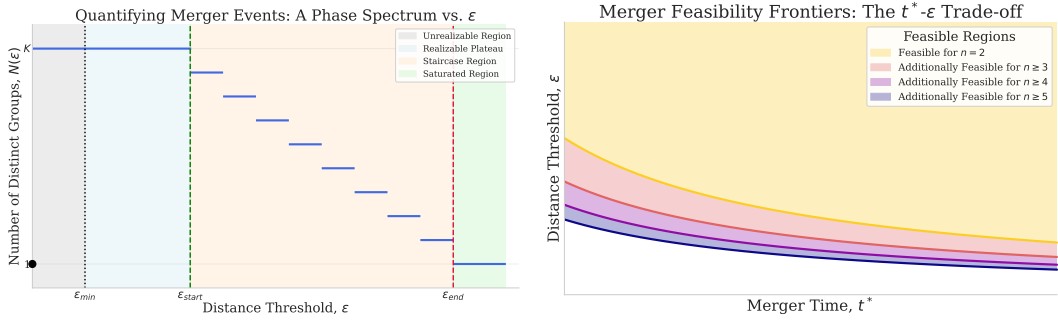

**a** The phase spectrum of merger events   **b** Merger feasibility frontiers

**Figure 5: Conceptual visualisation of lattice transition dynamics. (a)** The phase diagram illustrates how the number of merger events changes with the threshold $\epsilon$. Adopting a lattice perspective reveals a spectrum of distinct phases: an unrealizable thermodynamic phase at $\epsilon = 0$ which admits a single speciation transition, an unrealizable region below the system's precision limit, a realizable plateau of maximum potential mergers, a staircase of sequential transitions where the system "splits" into fine-grained merger states, and a final saturated state. **(b)** The feasibility frontiers visualise the trade-off between merger time $t^*$ and threshold $\epsilon$. The relationship is analogous to an uncertainty principle, where higher-order fluctuations merge faster, shifting their feasibility frontiers to the left.

**Feasibility and uncertainty.** Figure 5(b) visualises a fundamental trade-off in our framework. To precisely identify a merger (small $\epsilon$), one must observe the system for a long time (large $t^*$). Conversely, to detect a merger quickly (small $t^*$), one must accept a lenient threshold (large $\epsilon$). This relationship, which holds for any fluctuation tensor, is analogous to an uncertainty principle. For the specific, practical case where the state operator is the identity, $\rho(x) = x$, the plot shows an additional phenomenon: higher-order fluctuations ($n$) merge faster. This is because when analyzing the data vectors directly, high-order statistics capture fine-grained details that are more fragile and are erased more quickly by the diffusion process. The layered shading illustrates how the "feasible region" of detectable mergers expands as one moves to higher orders at any given time for this specific choice of $\rho$.

Thus, lattice transitions give a fine-grained view inside the most relevant operational regimes of standard diffusion models, which are invisible to purely classical criteria.

# C Experimental details and further results

## C.1 Compute and reproducibility

We conducted all experiments using a single Nvidia A100 GPU and provided sample code for reproducibility. Our implementation builds on open-source code from Hugging Face (diffusers library) and publicly available code from Kynkäänniemi et al. [2024], Li et al. [2023], Peebles and Xie [2022]. Our method is plug-and-play, requiring simple hyperparameter adjustments for these techniques without any major code modifications. For zero-shot style transfer (Section 4.5), we used the Img2Img transfer pipeline in diffusers (Meng et al. [2021], Rombach et al. [2022]), fixing the VP/DDPM schedule and adjusting the strength parameter.

Baseline experiments using grid search were computationally intensive, typically requiring 24–48 hours of GPU time. Fluctuation computations (Section 3) were primarily constrained by covariance matrix calculations and eigendecompositions. These can be efficiently optimised using multi-processing, though we used a single-process approach in this work. Our method is directly compatible with any standard implementation of the baselines. Preliminary small-scale experiments verified our theoretical framework but were excluded from the final paper.

## C.2 Convergence of data

We present images visualizing the normality tests as stated in Section 4.1

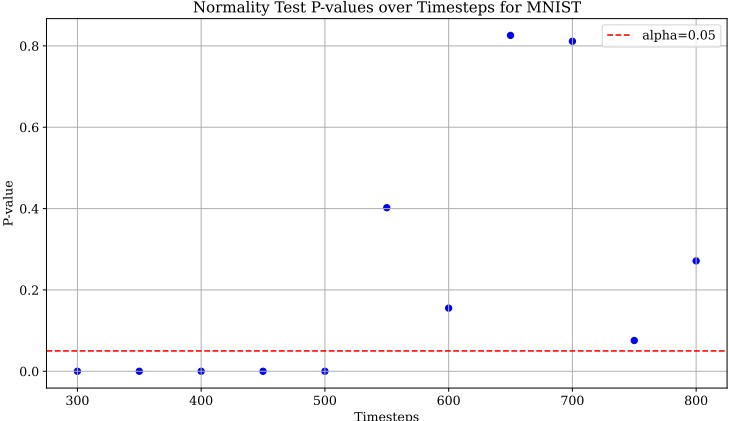

**a** Normality test p-values over time for MNIST

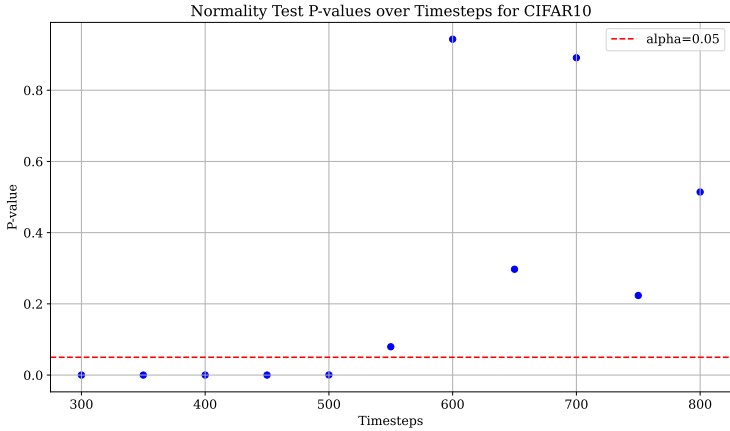

**b** Normality test p-values over time for CIFAR10

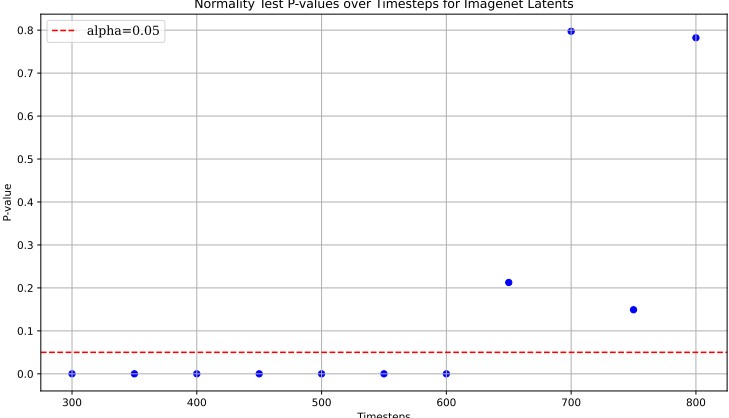

**a** Normality test p-values over time for Imagenet

Figure 7: Normality test p-values for the DDPM schedule

## C.3    Class-conditional generation

We compare two guidance schedules:

1. a *grid-search baseline* that follows Interval Guidance (IG) [Kynkäänniemi et al., 2024] with a *single* dataset-level interval found by brute force[9];

2. our *merger-aware schedule*, in which each class $k$ receives its own window $\big(t_{\text{start},k},\, t_{\text{end},k}\big)$ derived from fluctuation theory (Sections 4.1 and 4.2).

**Interval guidance baseline.**    Let $w > 0$ be the classifier-free guidance Ho and Salimans [2022b] (CFG) weight, and let $T$ be the full diffusion horizon. During reverse sampling we switch CFG on only for $t \in (t_{\text{end},c},\, t_{\text{start},c})$:

---

**Algorithm 2** Interval Guidance (class $c$)

---

**Require:** latent $x_T \sim \mathcal{N}(0, I)$, CFG weight $w$
 1: **for** $t = T - 1, \ldots, 0$ **do**
 2:    **if** $t_{\text{end},c} < t < t_{\text{start},c}$ **then**
 3:       $x_t \leftarrow \text{CFG}_w(x_{t+1}, c)$
 4:    **else**
 5:       $x_t \leftarrow \text{CFG}_0(x_{t+1}, c)$
 6:    **end if**
 7: **end for**
 8: **return** $x_0$

---

**Hyper-parameters and Grid-search baseline.** Following Peebles and Xie [2022] we set $w = 1.5$ for DiT-XL/2. Stable Diffusion requires stronger guidance; we use a fixed $w \in [3.5, 4.5]$ per dataset. For every dataset we sweep

$$t_{\text{end},c} \in \{0.1T, 0.2T, \ldots, 0.8T\}, \quad t_{\text{start},c} \in \{0.2T, \ldots, T\},$$

under the constraint $t_{\text{start},c} > t_{\text{end},c}$, yielding $44$ admissible pairs. On ImageNet the best pair is $(0.8T, 0.2T)$ (lowest FID); MNIST and CIFAR-10 select $(0.6T, 0.1T)$ and $(0.7T, 0.1T)$, respectively.

---

[9]In Kynkäänniemi et al. [2024], it is argued that a class-level or even a sample-level search is preferable. However, both of these settings require at least $10^3 - 10^6\times$ the baseline compute, making them infeasible for us. Our primary objective is to demonstrate that leveraging finer hierarchical levels can compensate for compute limitations. Specifically, we reason that for class-level brute-force search, it is more practical to consider the transitions of a fine-grained hierarchy derived from the representations of a semi/self-supervised learning model Chen et al. [2020], Grill et al. [2020], He et al. [2021]. A theoretical backing for the same is found in Theorem 8. We leave such an extension to future work. We note that asymptotically, our method converges to the same output as a per-sample-level grid search, as ultimately each data sample can be treated as a distinct singleton event $\Omega_{k,0}$.

For Stable Diffusion we replace FID by CLIP similarity and obtain $(0.8T, 0.1T)$ for both ImageNet and Oxford-IIIT Pet. One exhaustive sweep on DiT-XL/2 costs $4100$ GFLOPs $\times 50\,000$ samples $\times 44$ configs $\approx 9.0$ PFLOPs; five repeats per pair multiply the cost five-fold. Results for Imagenet, MNIST and CIFAR are in Table 2 in the main paper while that for Imagenet and Oxford-IIITPets using Stable Diffusion is in Table 9. Note that for the case of Imagenet, identical intervals are used for both settings as the empirical forward process trajectory is independent of the model choice.

**Merger-aware schedule (ours).**     For each class $k$, $t_{\text{start},k} = i^\star$, $t_{\text{end},k} = t_{\text{merge},k}$, where $i^\star$ is the global convergence index (Section 4.1) and $t_{\text{merge},k}$ is the first $t$ at which $M_\rho^{(2)}(\Omega_{k,t}, \Omega_{\ell,t}) = 1$ for some $\ell \neq k$ (Section 4.2). No search is required; windows differ automatically across classes.

| Model/Dataset | CLIP Similarity (↑) | Precision (↑) | Recall (↑) | Density (↑) | Coverage (↑) |
|---|---|---|---|---|---|
| SD (Imagenet, IG baseline) | $0.26 \pm 0.03$ | $0.75 \pm 0.04$ | $0.18 \pm 0.02$ | $0.80 \pm 0.05$ | $0.30 \pm 0.03$ |
| SD (Imagenet, IG Ours) | $\mathbf{0.31 \pm 0.02}$ | $\mathbf{0.78 \pm 0.02}$ | $\mathbf{0.23 \pm 0.01}$ | $\mathbf{0.88 \pm 0.03}$ | $\mathbf{0.34 \pm 0.02}$ |
| SD (OxfordIIITPet, IG baseline) | $0.28 \pm 0.02$ | $0.79 \pm 0.03$ | $0.21 \pm 0.03$ | $0.84 \pm 0.06$ | $0.33 \pm 0.05$ |
| SD (OxfordIIITPet, IG Ours) | $\mathbf{0.34 \pm 0.03}$ | $\mathbf{0.81 \pm 0.01}$ | $\mathbf{0.26 \pm 0.04}$ | $\mathbf{0.89 \pm 0.01}$ | $\mathbf{0.36 \pm 0.02}$ |

Table 9: Class conditional generation using Stable Diffusion

**Generative diagrams.**     Figure 13 plots the leading eigenvalues $\lambda_{\max}(\Sigma_{k,t})$ of the class covariances $\Sigma_{k,t}$ for ImageNet, CIFAR-10, and MNIST, highlighting merger points (proofs in Appendix B.3.3). Additional zoom-ins for ImageNet appear in Figure 12. For long-tail datasets used in Section 4.3, analogous diagrams are given in Figures 14a and 14c. We also plot the subplots for the 10 classes for Imagenet and OxfordIIITPet, having the greatest magnitude of principal eigenvalues in Figure 9.

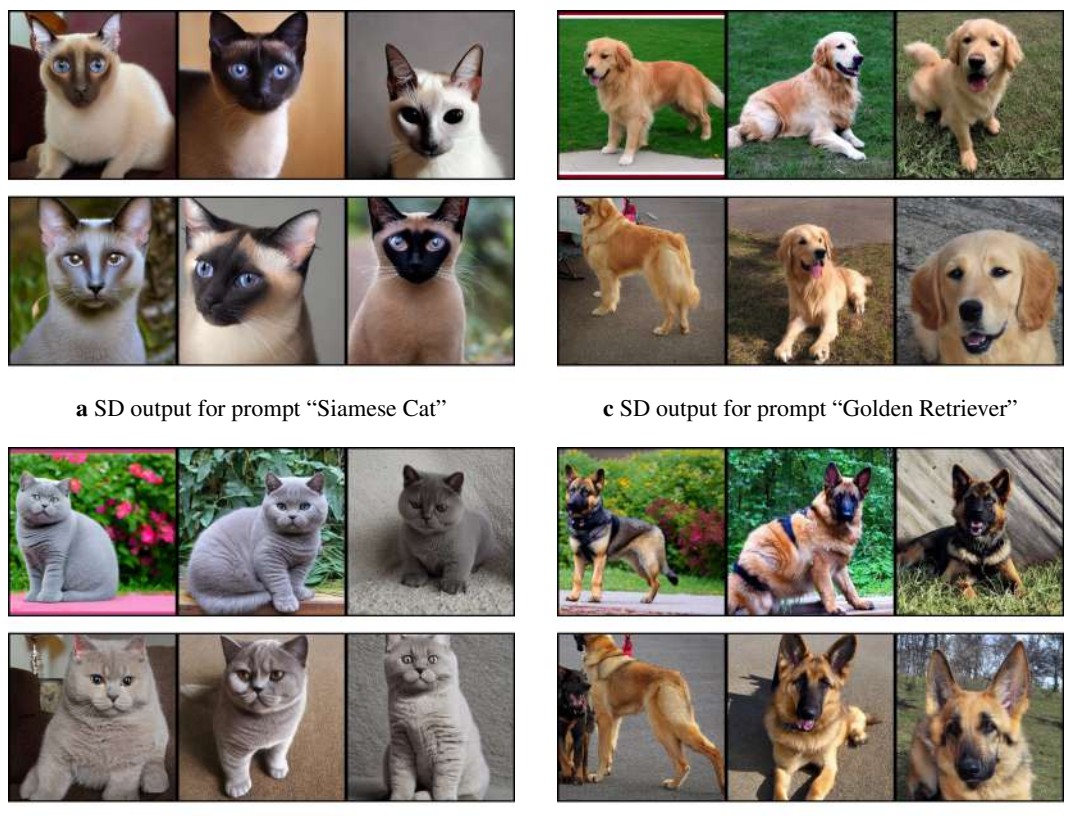

**a** SD output for prompt "Siamese Cat"     **c** SD output for prompt "Golden Retriever"

**b** SD output for prompt "British Shorthair"     **d** SD output for prompt "German Shepherd"

Figure 8: Stable-Diffusion samples for four Oxford-IIIT-Pet classes, generated with naïve interval guidance (top of each column) versus our method (bottom).

Figure 11 compares ImageNet samples from our merger-aware IG with the grid-search baseline, using a guidance weight of $4.5$ for visual clarity; Figure 8 does the same for Oxford-IIIT Pet under Stable Diffusion. Our method yields crisper details and fewer artefacts—despite eliminating $\approx 9$ PFLOPs

of search for Imagenet. Thus, class-wise guidance windows obtained from cross-fluctuation mergers can match or exceed the quality of an *exhaustive* dataset-level search, while slashing computational cost by orders of magnitude.

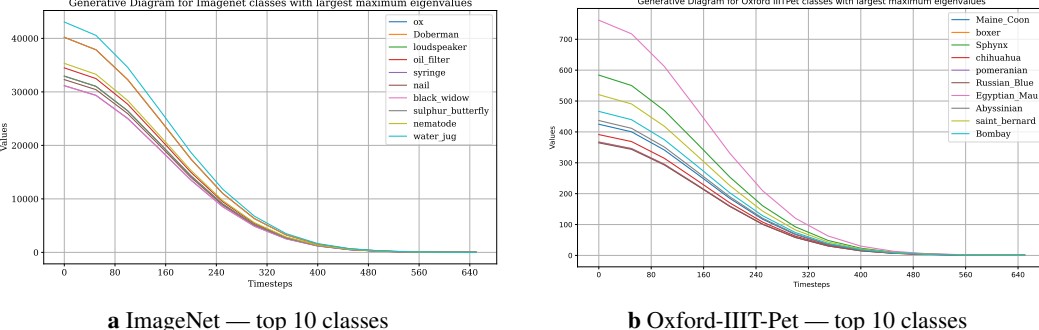

**a** ImageNet — top 10 classes

**b** Oxford-IIIT-Pet — top 10 classes

Figure 9: Merger transitions are upper-bounded by those of the ten classes with the largest principal-eigenvalue magnitudes in their covariance matrices.

| $\varepsilon$ | Merge prob. ($t = 0$) | FID ($\downarrow$) |
|---|---|---|
| 200 | 0.027 | $3.06\pm0.19$ |
| 100 | 0.013 | $\mathbf{2.86\pm0.15}$ |
| 80 | 0.010 | $2.92\pm0.17$ |
| 67 | 0.009 | $2.90\pm0.11$ |
| 20 | 0.002 | $2.88\pm0.14$ |

Table 10: Effect of the MAE threshold $\varepsilon$ (ImageNet).

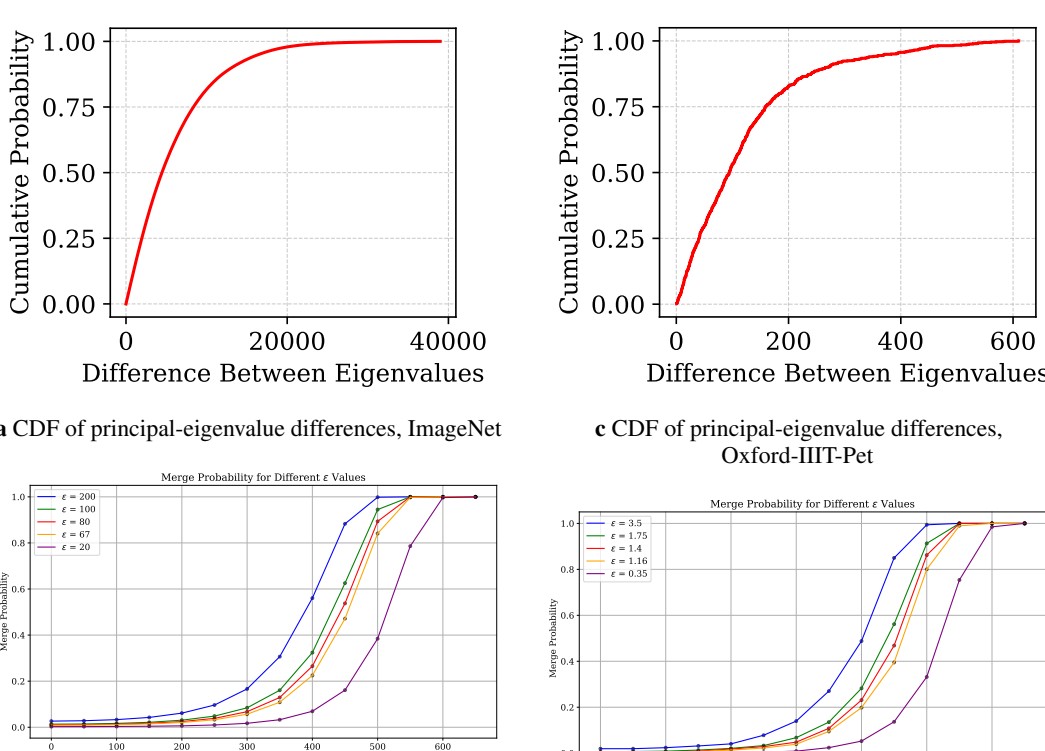

**a** CDF of principal-eigenvalue differences, ImageNet

**c** CDF of principal-eigenvalue differences, Oxford-IIIT-Pet

**b** Merge-probability vs. threshold, ImageNet

**d** Merge-probability vs. threshold, Oxford-IIIT-Pet

Figure 10: Eigenvalue-difference distributions and their associated merge-probability curves for ImageNet and Oxford-IIIT-Pet.

To understand the sensitivity of the parameter $\varepsilon$, we plot the distribution of the difference in eigenvalues (CDF) and evolution of merge probabilities for different choices of $\varepsilon$ (Figure 10). Our default choice $\varepsilon = 100$ has a merge probability $\approx 0.01$ at $t = 0$. We show corresponding FIDs in Table 10.

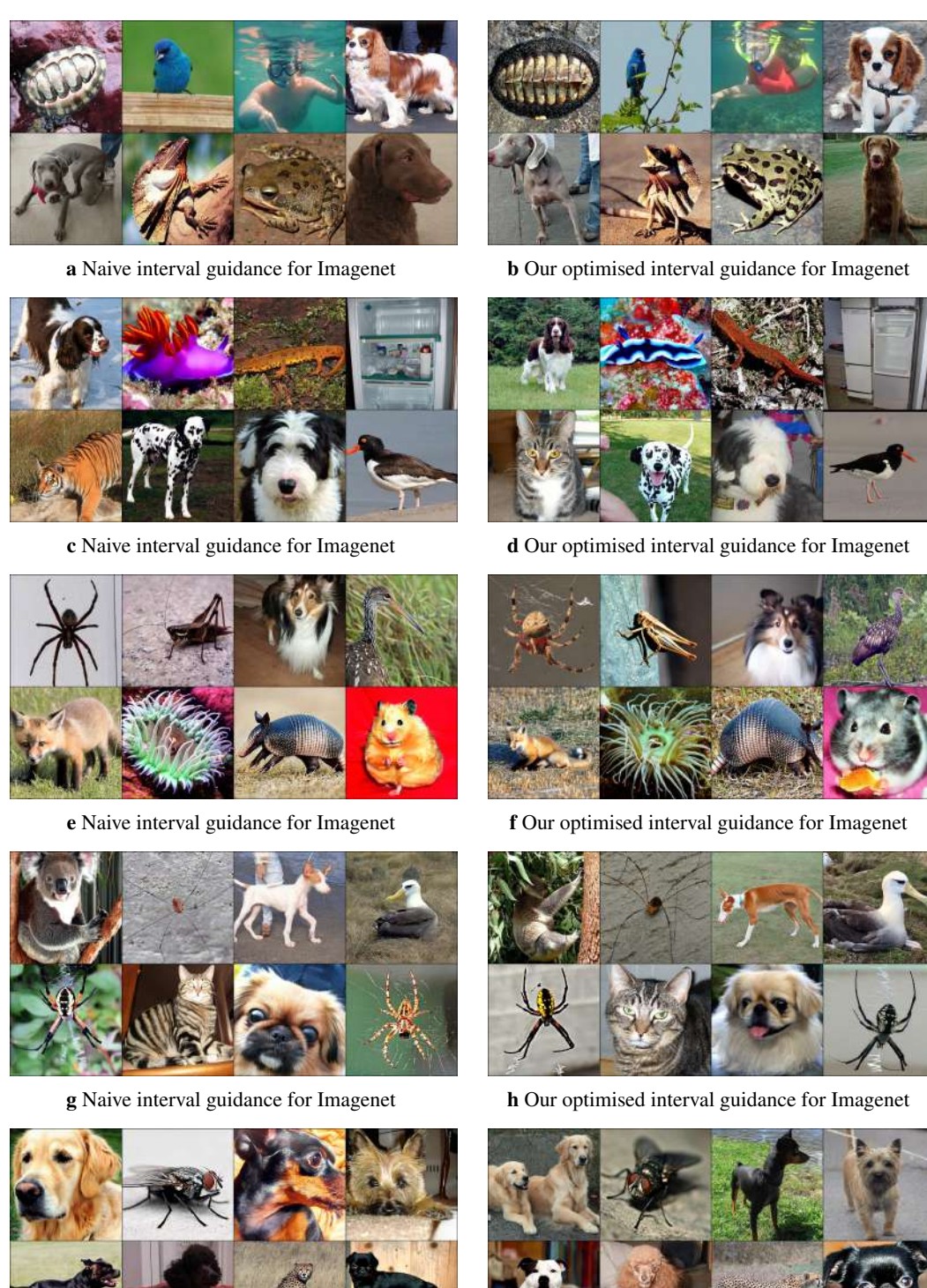

**a** Naive interval guidance for Imagenet        **b** Our optimised interval guidance for Imagenet

**c** Naive interval guidance for Imagenet        **d** Our optimised interval guidance for Imagenet

**e** Naive interval guidance for Imagenet        **f** Our optimised interval guidance for Imagenet

**g** Naive interval guidance for Imagenet        **h** Our optimised interval guidance for Imagenet

**i** Naive interval guidance for Imagenet        **j** Our optimised interval guidance for Imagenet

Figure 11: Visual Comparison of guidance for the Imagenet dataset [Deng et al., 2009, Ryu, 2024].
All samples were originally generated in $512 \times 512$ resolution.

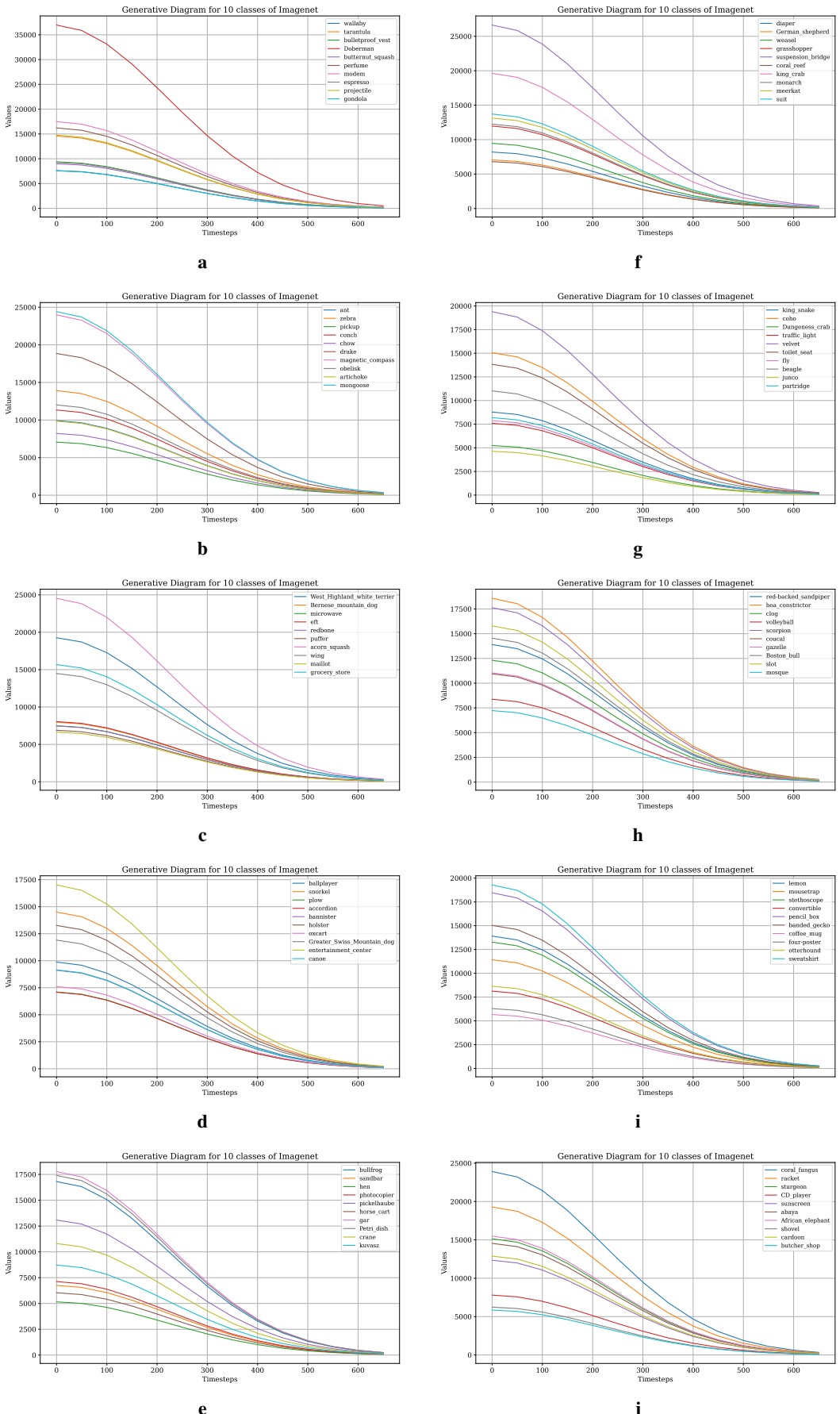

49

Figure 12: Merger-transition subplots for ImageNet (ten classes shown in two parallel columns).

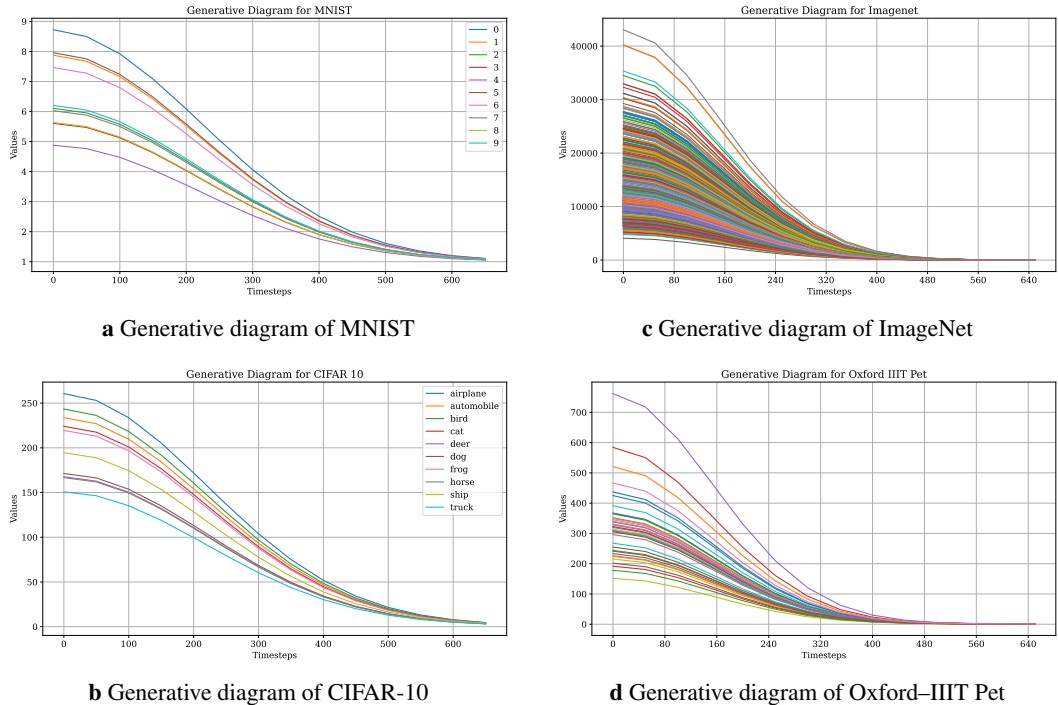

**a** Generative diagram of MNIST

**c** Generative diagram of ImageNet

**b** Generative diagram of CIFAR-10

**d** Generative diagram of Oxford–IIIT Pet

Figure 13: Merger–transition measure obtained from the intersection time of the principal eigenvalues of the class-covariance matrices.

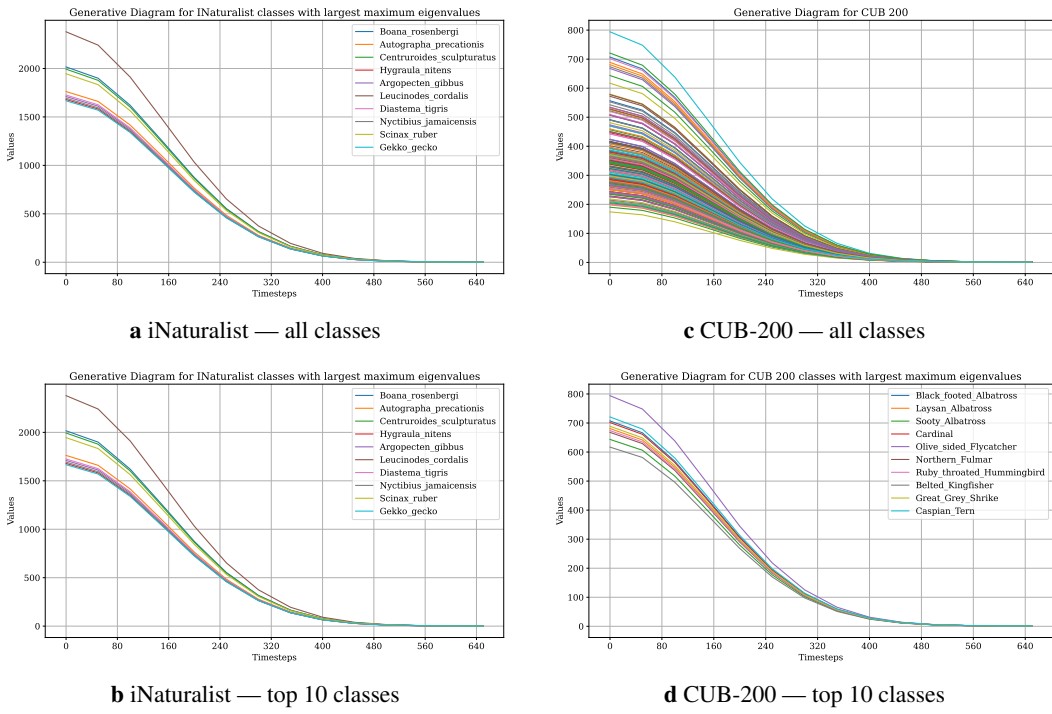

**a** iNaturalist — all classes

**c** CUB-200 — all classes

**b** iNaturalist — top 10 classes

**d** CUB-200 — top 10 classes

Figure 14: Merger–transition measures obtained from the intersection times of the principal eigenvalues for iNaturalist and CUB-200. The bottom row shows that the classes with the ten largest eigenvalue magnitudes upper-bound all other transitions.

## C.4 Rare-class generation

**Interpolation-based interval guidance.** In Section 4.3 we observed that augmenting Algorithm 2 with an *interpolation correction*—akin to ILVR [Choi et al., 2021]—gives the best fidelity for CUB-200 and iNaturalist-2019. The full procedure is listed in Algorithm 3; it differs from standard Interval Guidance only in lines 5–7.

---

**Algorithm 3** Interpolation-based interval guidance (class $c$)

---

**Require:**   • $x_T \sim \mathcal{N}(\mathbf{0}, \mathbf{I})$                                                                                          *latent to be denoised*
   • $\hat{x}_0 \sim p_0$ with shape$(x_T) =$ shape$(\hat{x}_0)$                                                            *class-$c$ exemplar*
   • guidance weights CFG$_w$, CFG$_0$
   • interpolation schedule $\eta = \{\eta_t\}_{t=0}^{T-1}$, where $0 \leq \eta_t \leq 1$
1: **for** $t = T - 1, \ldots, 0$ **do**
2:    **if** $t_{\text{start},c} < t < t_{\text{end},c}$ **then**                                                            ▷ guidance window
3:       $x_t \leftarrow \text{CFG}_w(x_{t+1}, c)$
4:       $\hat{x}_t \leftarrow \text{FWD}(\hat{x}_0, t)$
5:       $x_t \leftarrow \eta_t x_t + (1 - \eta_t) \hat{x}_t$
6:    **else**
7:       $x_t \leftarrow \text{CFG}_0(x_{t+1}, c)$
8:    **end if**
9: **end for**
10: **return** $x_0$

---

Note that in Algorithm 3, CFG$_w$ applies classifier-free guidance with strength $w$; CFG$_0$ disables conditioning. FWD$(\hat{x}_0, t)$ generates the *forward-noised* version of the exemplar at step $t$. The convex update in line 5 nudges the current latent towards the exemplar's trajectory, counteracting class drift.

**Choosing the interpolation schedule $\eta$.** Because guidance corrections are most valuable late in the reverse chain, we derive $\eta_t$ from the noise schedule $\beta_t$: $\eta_t = s(\beta_t / \max_{u<T} \beta_u)$, $0 < s \leq 1$. The scale factor $s$ is found by a binary search over $[10^{-4}, 10^{-2}]$: larger values degrade sharpness, while smaller ones have negligible impact.

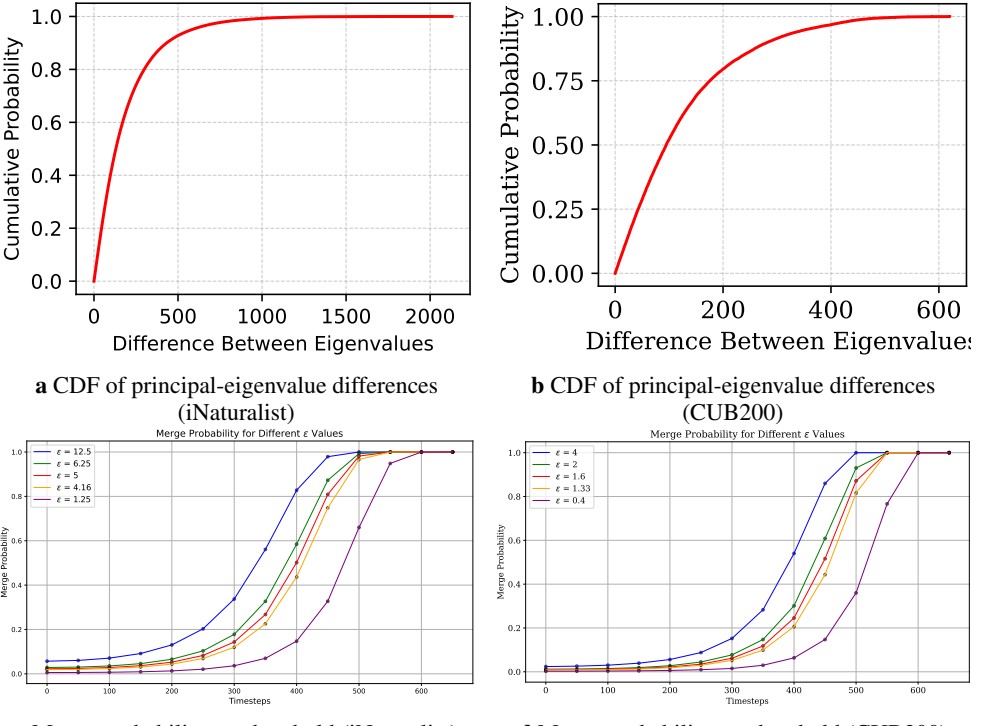

**a** CDF of principal-eigenvalue differences (iNaturalist)

**b** CDF of principal-eigenvalue differences (CUB200)

**c** Merge-probability vs. threshold (iNaturalist)

**d** Merge-probability vs. threshold (CUB200)

Figure 15: Comparison of eigenvalue-difference distributions and merge-probability curves for iNaturalist vs. CUB200.

**Experimental settings.** For all runs, we fix $\epsilon = 5$ on iNaturalist and $\epsilon = 2$ on CUB-200 when computing merger statistics. Figures 14a and 14c visualise per-class merger probabilities, with complementary CDF and eigenvalue trajectories in Figures 15a to 15d. Top-10 eigenvalue plots appear in Figures 14b and 14d. Qualitative comparisons between Algorithm 3 and naïve Stable Diffusion, using identical random seeds, is given in Figures 16 and 17.

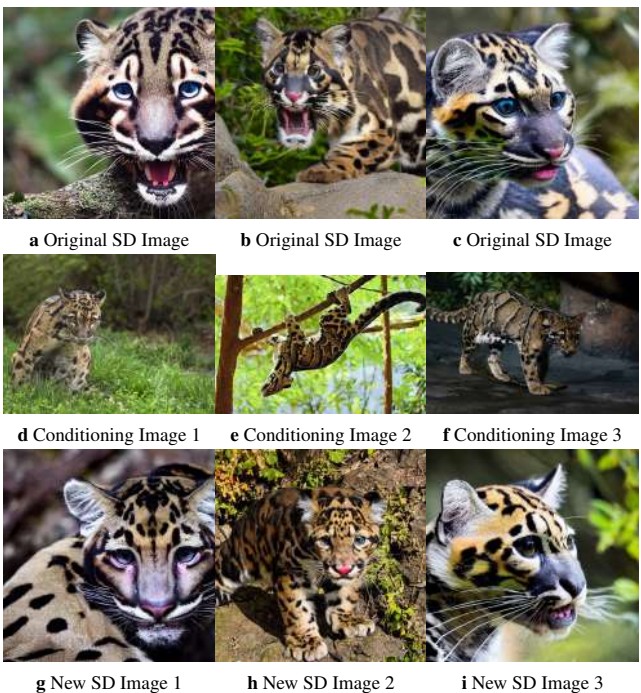

Figure 16: Visual comparison of generation algorithms for stable diffusion for the iNaturalist class prompt "clouded leopard walking".

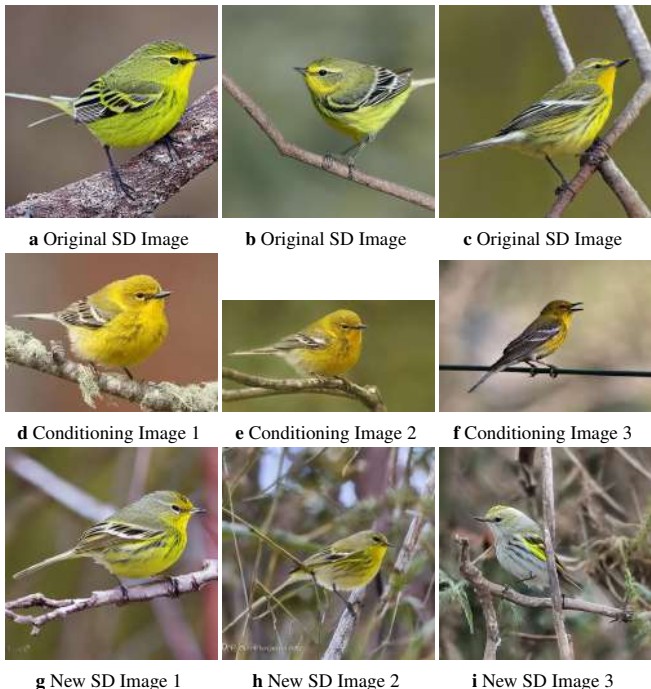

Figure 17: Visual comparison of generation algorithms for stable diffusion for the CUB-200 prompt "pine warbler."

## C.5 Zero-shot classification

In Li et al. [2023], a pretrained *class-conditional* diffusion network $f_\theta$ is repurposed as a classifier by averaging softmax logits over forward-diffused replicas of the query image. We keep that spirit but (i) restrict the average to the *class-specific guidance window* $[t_{\text{start},\lambda}, t_{\text{stop},\lambda}]$ identified in Section 4.2, and (ii) attach an importance weight $w(t)$ to every timestep $t$. Setting $w(t) = 1/T$ and the bounds $t_{\text{start},\lambda} = 0, t_{\text{stop},\lambda} = T$ recovers the estimator of Li et al. [2023].

Throughout this section, $\Lambda = \{1, \ldots, K\}$ is the label set; $\varepsilon_n \overset{\text{iid}}{\sim} \mathcal{N}(0, I_d)$ are noise seeds ($n = 1, \ldots, N$); $\text{FWD}_t(x, \varepsilon)$ denotes the *forward* map that produces the noisy latent $z_t$ at step $t$; $w(t) \geq 0$ is a probability mass on the integer interval $[t_{\text{start},\lambda}, t_{\text{stop},\lambda}]$.

---

**Algorithm 4** Zero-shot class probability $p_\lambda(z)$

---

**Require:** query $z \sim p_0$, class label $\lambda \in \Lambda$, time window $t_{\text{start},\lambda} \leq t \leq t_{\text{stop},\lambda}$, weights $\{w(t)\}$ summing to 1, noise seeds $\{\varepsilon_n\}_{n=1}^N$
1: $p_\lambda \leftarrow 0$
2: **for** $t = t_{\text{start},\lambda}, \ldots, t_{\text{stop},\lambda}$ **do**
3:     **for** $n = 1, \ldots, N$ **do**
4:         $z_t \leftarrow \text{FWD}_t(z, \varepsilon_n)$
5:         $s_\lambda \leftarrow -\left\| f_\theta(z_t, \varepsilon_n, \lambda) - \varepsilon_n \right\|^2$
6:         $p_\lambda \leftarrow p_\lambda + \dfrac{w(t)}{N} \dfrac{\exp(s_\lambda)}{\sum_{\mu \in \Lambda} \exp\left(-\| f_\theta(z_t, \varepsilon_n, \mu) - \varepsilon_n \|^2\right)}$
7:     **end for**
8: **end for**
9: **return** $p_\lambda$

---

Let $\alpha_t$ be the signal coefficient of the VP process (2.4). The signal-to-noise ratio is $\text{SNR}(t) = \alpha_t^2/(1 - \alpha_t^2)$. We consider three discrete weight laws: *Uniform:* $w(t) = 1/(t_{\text{stop},\lambda} - t_{\text{start},\lambda} + 1)$. *Inverse-SNR:* $w(t) \propto \text{SNR}(t)^{-1}$ with $t_{\text{start},\lambda} = 0$. *Truncated inverse-SNR:* same as previous but restricted to $t \geq 20$ (empirically, very early steps degrade performance as the score model is not exact at these time scales [Chen et al., 2022a, Karras et al., 2022]). We fix $N = 250$ for all of our experiments following Li et al. [2023].

### C.5.1 Binary classification via linear probes

To understand why non-uniform weights help, we study binary accuracy along the forward chain with *no* diffusion model involved. Pick two ImageNet classes $\{\lambda, \mu\}$ at random and sample $\min\{\text{card}(\lambda), \text{card}(\mu), 10000\}$ number of images for each class. At each step $t$, we extract the noisy embedding $z_t$ (VP schedule) and train a linear MLP on $80\%$ of the embeddings; the rest forms the test set. We repeat the experiment 20 times and report mean $\pm$ s.d.

Figure 18 shows that test accuracy rises sharply and peaks near the *merger time* of $\lambda$ and $\mu$, after which it is not defined since the embeddings corresponding to $\lambda, \mu$ become practically indistinguishable. Averaging the accuracies with the three weight laws yields the means in Table 11; using the single best $t$ (the merger time) achieves the highest score but is undefined for the multi-class case, however, the general trend for the other strategies that are valid for the multi-class setting remains the same.

| Weighting strategy | Avg. binary accuracy ↑ |
|---|---|
| Uniform | $0.72 \pm 0.03$ |
| Inverse-SNR | $0.79 \pm 0.06$ |
| Trunc. inv. SNR | $0.82 \pm 0.04$ |
| Merge-time probe | $\mathbf{0.90 \pm 0.02}$ |

Table 11: Binary accuracy for random ImageNet pairs.

Each forward step convolves the data with a Gaussian kernel, progressively smoothing non-linear features. Just before the classes merge, the representation is *simpler* yet still separates the two manifolds, making linear decision boundaries easiest to learn. This explains why inverse-SNR weighting, which emphasises mid-to-late timesteps—beats uniform weighting, and provides an additional reason why further truncation gives a small extra gain. Thus, merger-aware weighting focuses the zero-shot estimator on the most discriminative region of the diffusion trajectory, narrowing the accuracy gap to dedicated representation learners such as CLIP.

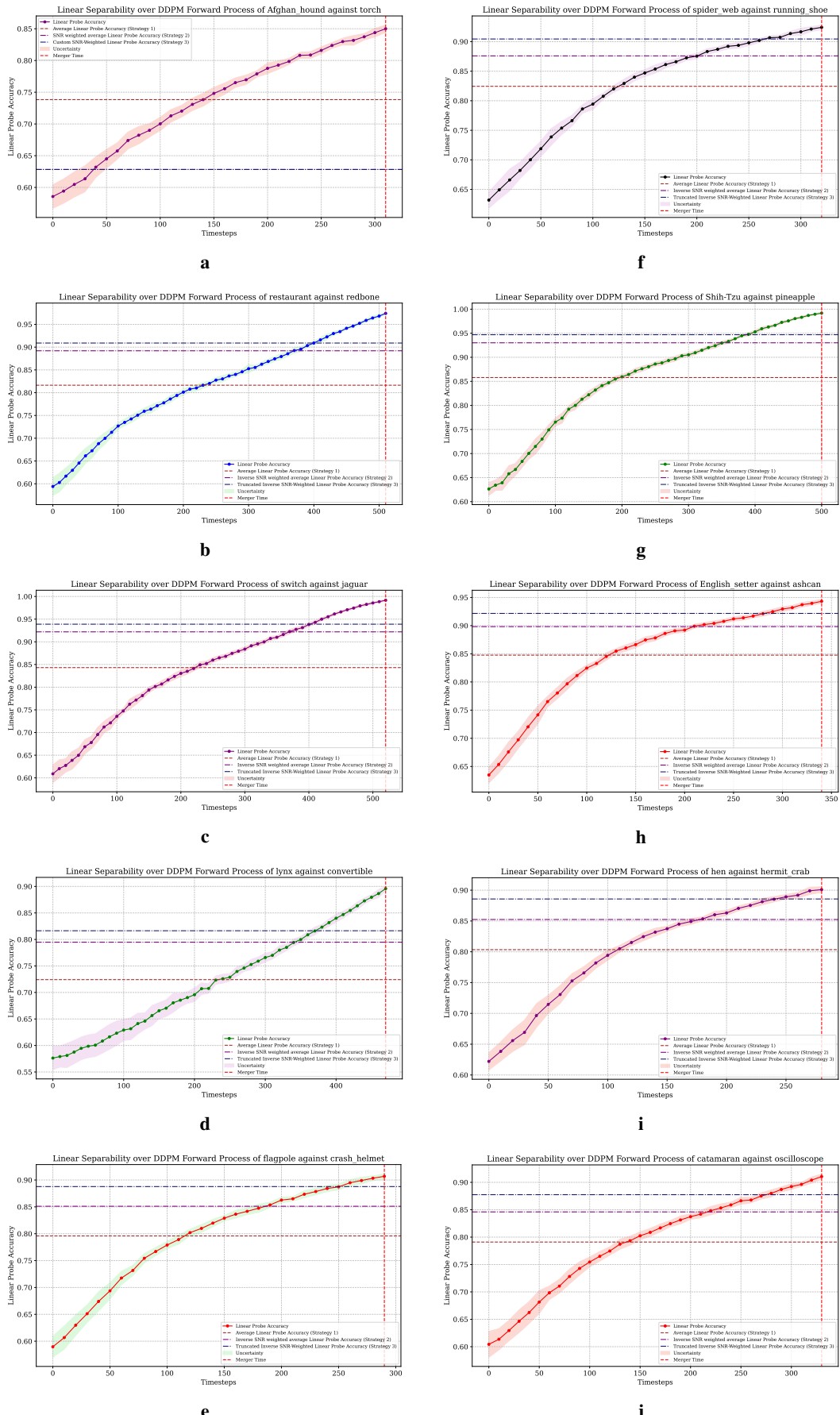

Figure 18: Linear-probe accuracy through the forward diffusion process. Later timesteps hold greater discriminative information between the two classes.

## C.6 Zero-shot style transfer

In [Meng et al., 2021] the parameter $t_{\text{stop},z}$ is estimated by a grid search over the interval $0.1T, 0.2T, \ldots, T$ for an entire dataset for each style which we follow for our baseline implementation using the PSNR metric. $t_{\text{stop},z}$ estimated in this way for both datasets were observed to lie in the range $0.3T, 0.4T, 0.5T$ across styles. For our implementation $t_{stop,z}$ is estimated class-wise as the smallest merger time of the class(es) to which $z$ belongs.

---

**Algorithm 5** Zero-shot Style Transfer

---

**Require:** query $z \sim p_0$, noise seed $\epsilon_n$, maximum time $t_{\text{stop},z}$
1: $z_{t_{\text{stop},z}} \leftarrow \text{FWD}_{t_{\text{stop},z}}(z, \varepsilon_n)$
2: **return** $z^* \leftarrow \text{BWD}_{\theta, t_{\text{stop},z}}(z_{t_{\text{stop},z}})$

---

We use open-source fine-tuned stable diffusion models available on Hugging Face for all of our experiments. These models are trained on the artistic styles of Studio Ghibli Miyazaki and Ghibli [2014], Van Gogh Roojen [2019] and the styles of the animation game Elden Ring Software and Entertainment [2022] and series Arcane Production and Games [2021][10]. Note that some samples from the OxfordIIITPet Dataset may be randomly censored by the automatic filter present in these models. We observed that this happens mostly for dog images, where some breeds have samples with their mouth open. This can be mitigated to some extent by center cropping and realigning; however, if preprocessing fails, we discard such images. Table 5 in Section 4.5 of the main paper has results for the Studio Ghibli and Van Gogh styles while Table 12 has results for the Elden Ring and Arcane styles.

| Style | Elden Ring | | Arcane | |
|---|---|---|---|---|
| Models/Metrics | PSNR ($\uparrow$) | MSE ($\downarrow$) | PSNR ($\uparrow$) | MSE ($\downarrow$) |
| SD Edit (OxfordIIITPets) | $24.19 \pm 0.72$ | $0.09 \pm 0.006$ | $25.17 \pm 0.42$ | $0.09 \pm 0.005$ |
| **Ours** (OxfordIIITPets) | $\mathbf{28.14 \pm 0.69}$ | $\mathbf{0.03 \pm 0.002}$ | $\mathbf{28.35 \pm 0.72}$ | $\mathbf{0.03 \pm 0.006}$ |
| SD Edit (AFHQv2) | $26.08 \pm 0.37$ | $0.06 \pm 0.003$ | $26.49 \pm 0.34$ | $0.05 \pm 0.004$ |
| **Ours** (AFHQ v2) | $\mathbf{27.56 \pm 0.37}$ | $\mathbf{0.04 \pm 0.009}$ | $\mathbf{28.23 \pm 0.18}$ | $\mathbf{0.03 \pm 0.001}$ |

Table 12: Style Transfer results for Elden Ring and Arcane styles

Visual results are in Figures 19 and 20 for the AFHQv2 and OxfordIIITPet datasets, respectively. We also show visual proof of our core assumption from Section 4.5 elaborated in Appendix B.3.4 through visual plots of the evolution of the fourier transforms through the forward process of images mainly differing only in style, in Figure 21. Here we set a gap of $0.1T$ between the two images. These plots empirically show that for images differing only in style but with the same essential structure, their fourier transforms are *close* and this fact extends to their noised versions as well due to the nonlinear decay in Brownian Motion (Appendix B.3.3).

---

[10]Disclaimer: All referenced trademarks, copyrighted characters, and original artworks remain the property of their respective owners. Algorithm 5 was utilised for research purposes only and is not intended for commercial use or to infringe upon the intellectual property rights of the original creators.

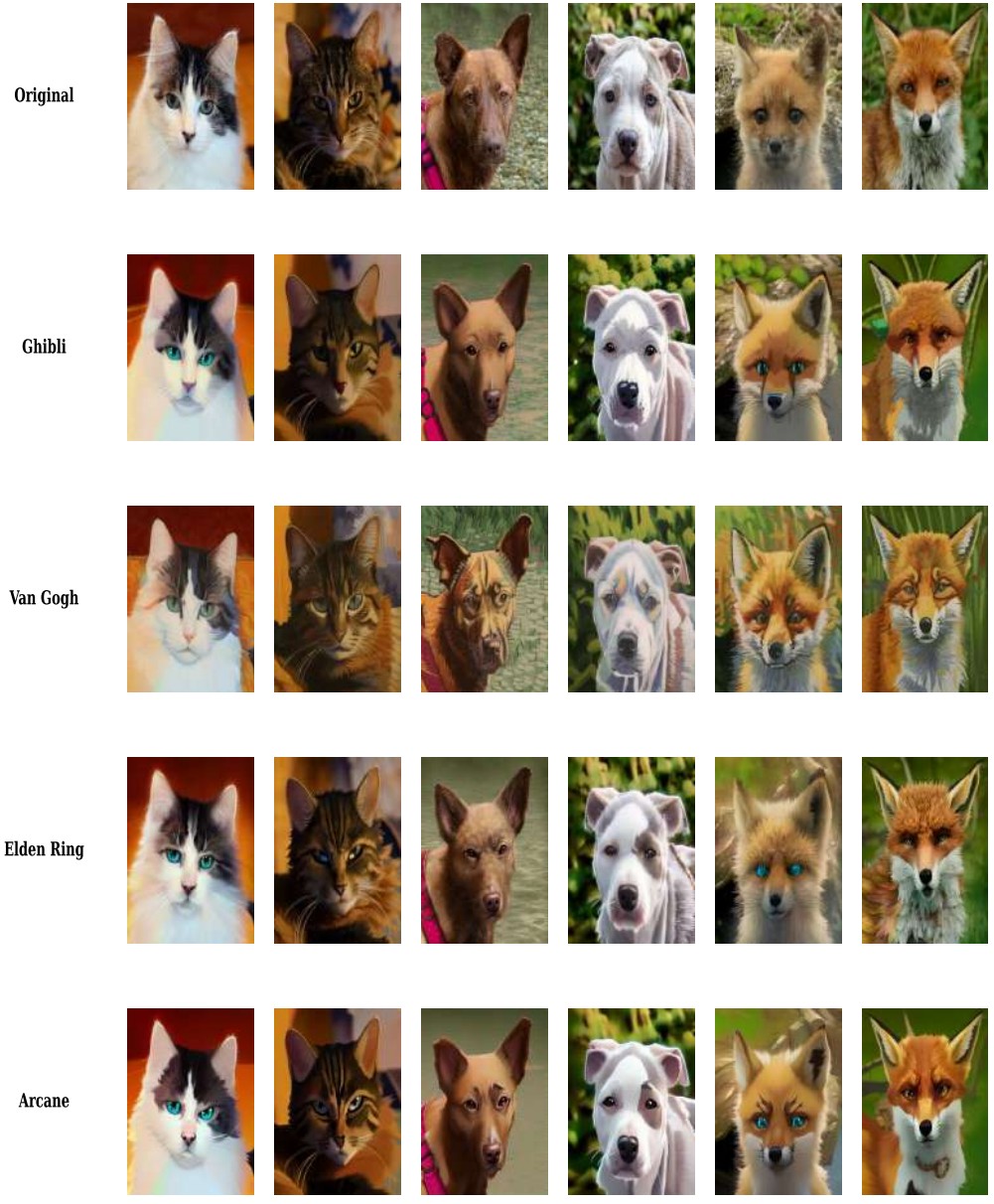

Figure 19: Zero Shot Transfer results on AFHQ v2

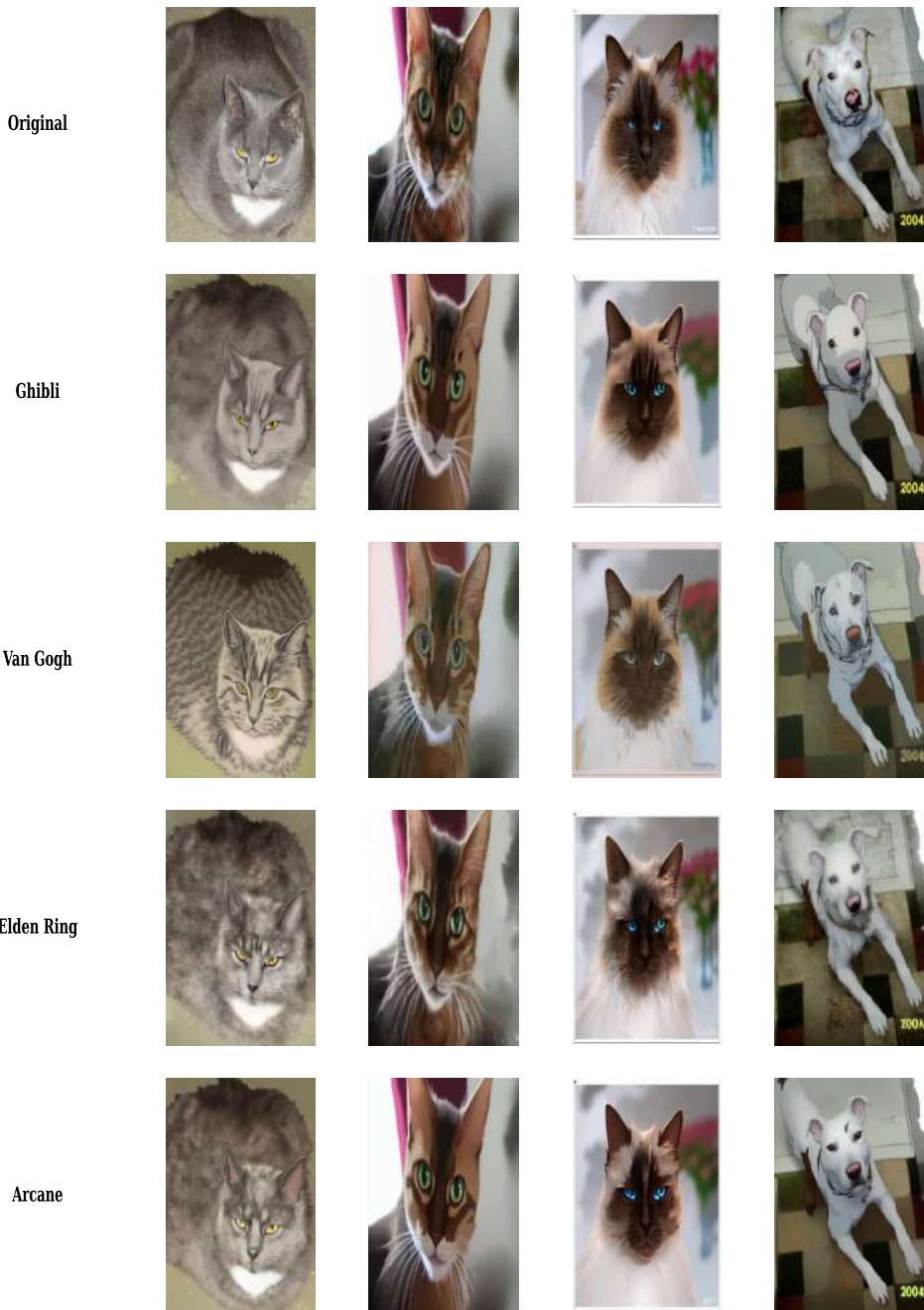

Figure 20: Zero Shot Transfer results on OxfordIIITPet

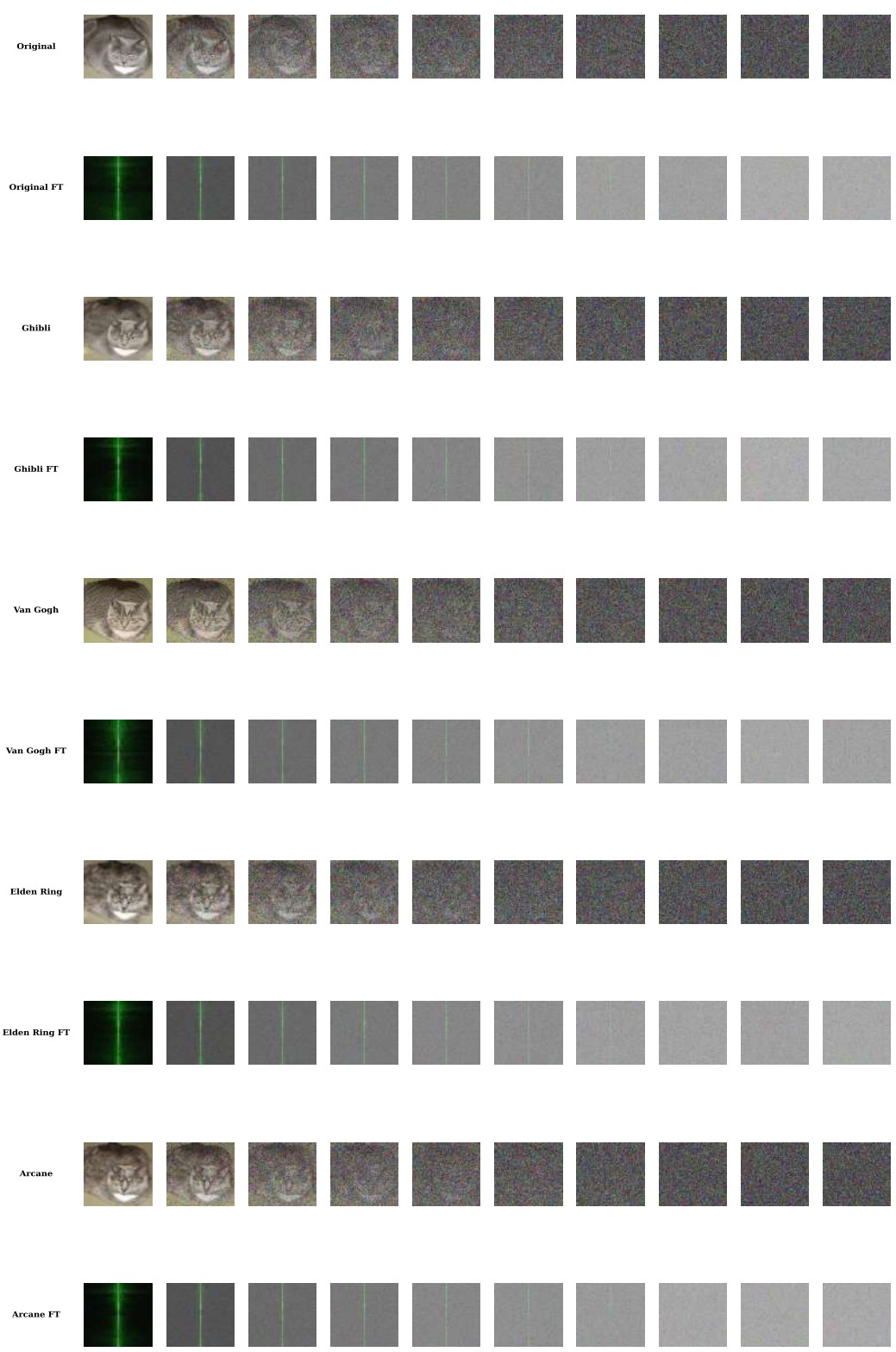

Figure 21: Fourier transforms and decay in Fourier spectra are close for structurally similar data

# D    Limitations and future work

**Modelling assumptions.**    Our analysis is built on three structural premises: (i) a variance–preserving (VP) noise schedule, (ii) isotropic Brownian perturbations, and (iii) the Fourier-regularity bound of (4.7). These hold for the canonical SDE/ODE pairs [Song et al., 2021, Ho et al., 2020], but have not yet been extended to non-VP schedules (e.g. EDM [Karras et al., 2022]) or to anisotropic diffusions. Relaxing any of the three assumptions remains open.

**Moment truncation.**    Empirically, the first few centered moments suffice to expose the phase boundaries, yet difficult data (heavy-tailed, multimodal) may demand higher orders. Reliable estimation of tensor moments beyond order four is computationally expensive and lacks sharp concentration inequalities.

**Memory overhead.**    Computing covariances $\Sigma_{k,t} \in \mathbb{R}^{d \times d}$ for many classes incurs $O(Kd^2)$ memory. Random projections, sketching, or block-diagonal approximations could mitigate this without degrading detection power.

**Modalities beyond 2-D images.**    All experiments target natural images. How merger-based guidance interacts with 3-D diffusion, video or audio, or with text diffusion is unexplored.

**Training-time usage.**    We use cross-fluctuations *post hoc*. Injecting merger times into the loss—as a curriculum on noise levels or as a regulariser enforcing class separation—may accelerate or stabilise training; we leave this to future work.

# E    Related work

**Statistical physics of diffusion models**    Thermodynamic transitions in diffusion models have been studied in prior work, such as Raya and Ambrogioni [2024], which identifies a mixing time transition akin to ours in Section 4.1, but for a narrow class of initial distributions. This is similar to the Curie-Weiss transition in ferromagnetic systems Kivelson et al. [2024], with an analytical estimate. We provide an alternative derivation of this dependency for a broader class of sub-Gaussian data in Appendix B.3.1, but our approach in Section 4.1 is more general, relying solely on the assumption that the supports of the data and the isotropic Gaussian are *essentially disjoint*. Biroli and Mézard [2023], Biroli et al. [2024] extend Raya and Ambrogioni [2024] by modeling data as a Gaussian mixture and show three distinct *phases* in a class-conditional setup similar to Section 4.2. Our framework further builds on these ideas in Appendix B.3.6, where we showcase a general framework relying on discrete *lattice transitions* more common in quantum systems.

**Fast sampling for diffusion models.**    DDIM [Song et al., 2020], IDDPM [Nichol and Dhariwal, 2021], DPM-Solver [Lu et al., 2022], and EDM [Karras et al., 2022] reduce the reverse step count; early-exit criteria such as ours are orthogonal and in principle could be combined with any of them. We leave such extensions to future work. As discussed earlier, the occurrence of this criterion has also been demonstrated by [Raya and Ambrogioni, 2024, Biroli and Mézard, 2023, Biroli et al., 2024] through a different analysis based on symmetry breaking of a potential function; we show that our framework extends such considerations in Appendix B.3.6.

**Theoretical lenses on diffusion.**    Hyper-contractivity [Saloff-Coste, 1994, Chen et al., 2022a], mixing-time bounds [Levin and Peres, 2017], and classical coupling [Aldous and Fill, 2002] traces back to Markov-chain theory. We recast these ideas as *cross-fluctuation mergers*, bridging discrete and continuous settings in Appendix B.1.4.

**Conditional guidance.**    Score distillation [Poole et al., 2022], ILVR [Choi et al., 2021], classifier-free guidance [Ho and Salimans, 2022a], and Interval Guidance (IG) [Kynkäänniemi et al., 2024] dominate conditional generation. Our merger-aware IG trims the interval search from per-sample to per-class with no extra hyper-parameters with details in Section 4.2 and Appendix C.3

**Zero-shot classification using diffusion networks.**    Li et al. [2023] showed diffusion backbones encode class information. Importance weighting by intermediate times with a cutoff at merger times

boosts their zero-shot accuracy at equal compute (Section 4.4). We also demonstrate that near the merger times, the diffusion process demonstrates near-perfect discriminability Appendix C.5.1.

**Rare-class synthesis.** Long-tail generation typically relies on fine-tuning [Bansal et al., 2023, Samuel et al., 2024]. Our fluctuation-based guidance is tuning-free and alleviates the mode dropping at inference time by utilizing optimised IG based guidance (Section 4.3, Appendix C.4)

**Zero-shot style transfer using diffusion models** The intriguing property of style transfer by learning a trained VP-SDE only on the target style after utilizing the forward process to corrupt inputs from the source style was initially shown in Meng et al. [2021] and has become standard practice across setups Rombach et al. [2022]. To our knowledge, we are the first to provide a theoretical foundation for this approach in terms of Fourier regularity that manifests in observed transitions (Section 4.5, Appendix B.3.4) while showing that setting the noising parameter based on merger times can boost fidelity (Section 4.5, Appendix C.6)

