# OpenReview forum: "Cross-fluctuation phase transitions reveal sampling dynamics in diffusion models"
_NeurIPS.cc/2025/Conference — NeurIPS 2025 poster_

### Official Review · Reviewer_yAPh · 2025-06-30

**Clarity:** 2
**Significance:** 3
**Originality:** 3
**Rating:** 4
**Confidence:** 3

**Summary:**

The paper introduced a study of score-based models using cross-fluctuations. The aim of the paper is to show that in the sampling process there are several transitions between distinct states. The paper illustrates that this states can be detected with the cross-fluctuation formalism and be used to speed-up the sampling and improve zero-shot use-cases.

**Questions:**

See Strengths and Weaknesses.

**Ethical Concerns:**

["NO or VERY MINOR ethics concerns only"]

**Final Justification:**

See the answer to authors rebuttal.

**Limitations:**

See Strengths and Weaknesses.

**Paper Formatting Concerns:**

No issues with the formatting.

**Quality:**

2

**Strengths And Weaknesses:**

The paper is not really accessible to all the community. Even if the result is a purely mathematical one, we still have to remember that we are submitting the work to share the knowledge to most of the community. Perhaps, before diving into the mathematical formalism, I always recommend to give an highly level overview on the method and intuition behind the discovery. Due to the high-density of mathematical formalism without broader intuition, the paper is really hard to read. Half-way through the paper, I lost completely the scope of the paper.

Said so, I don't really understand what a "transition" is in this use-case. What does actually measure that there is a phase transition? Phase transition between what and what? In Physics, phase-transitions can be first or second order with their properties. Here, the use of transition between phase is used since the beginning but I haven't seen any definition on the states and/or transitions. In the algorithm, are you sampling with two score-based models simultaneously and measure the cross-fluctuations between the two? Is it the code available?

I believe that this kind of approach is novel, I've never seen these concepts applied to score-based models. Experimental results show improvements in FID and inference time. However, I'd ask the authors to give a higher-level intuition in the main paper and part of the mathematical definition to the appendix. At the current state, I see a high-potential paper with lack in interpretability for most of the research community.

---

> ### Author Rebuttal · Authors · 2025-07-31
>
> We would like to express our sincere gratitude to the reviewer for their thoughtful and constructive feedback. We address these points carefully, one by one, below.
>
> ---
>
> ### **High-level intuitive overview**
>
> Before addressing the specific technical points, we begin with the core intuition:
>
> **Core insight**: In diffusion sampling, structured data (e.g., class-separated) gradually transitions into noise. This process unfolds through **discrete "merger events"** where formerly distinguishable groups suddenly become statistically indistinguishable. Think of this as watching ice cubes melt: separate cubes soften gradually, then suddenly merge into uniform water.
>
> **What we measure**: We detect **cross-fluctuations**—the moment when the statistical "fingerprints" of different data groups converge. These moments mark **optimal intervention points** for early stopping, guidance, and other sampling strategies.
>
> **Practical value**: Instead of relying on expensive grid searches, our method automatically pinpoints the most effective windows for intervention based on these fluctuations.
>
> ---
>
> ### **Defining phase transitions in our context**
>
> We acknowledge that the terminology around "phases" and "transitions" needs clarification. In our framework:
>
> **Phase transition**: A discrete point during forward diffusion where two previously distinguishable data clusters become statistically indistinguishable. These are detected as sharp discontinuities in the cross-fluctuation statistic.
>
> **States**: Each "phase" corresponds to a statistical regime:
>
> - **Structured phase**: Classes remain distinguishable, retaining unique moment structures
> - **Merged phase**: Classes become statistically identical
> - **Transition point**: A discrete timestep at which the regime switch occurs
>
> #### Relation to physical systems
>
> While not the main focus, these phase transitions are **emergent phenomena**, driven by finite precision in numerical computation. Classical theory predicts a single mixing event as $t \to \infty$, but for $d$-dimensional Sub-Gaussian data, this occurs practically at time scale $\mathcal{O}(\log d)$ (see Proposition 15, Appendix B).
> Our framework uncovers **intermediate, computationally accessible transitions**—moments where merging occurs before the theoretical limit, enabling real-world intervention strategies.
>
> (**Click for plot**) [Merger phase diagram](https://postimg.cc/3kQMG8tN)
>
> This plot reveals the **phase structure governing the choice of $\epsilon$**, showing four distinct operational regions. The **optimal region** allows detection of the largest number of mergers ($1 < K \leq N-1$ for $N$ classes).
> Values of $\epsilon$ that are too small fall below numerical resolution; values too large cause early overmerging. Our chosen threshold of $\lambda_{\max}/400$ lies within this optimal region, where the chance of premature merging at $t=0$ is typically $\mathcal{O}(10^{-2})$ for standard datasets.
>
> This emergent behavior can be interpreted, in physics terms, as a **Renormalization Flow** induced by discretization. While a full treatment is outside our current scope, Lemma 17 (Appendix B.14) provides a comparison between the **classical single-phase prediction** and the **multiple discrete regimes** observed under finite precision. We prove that a single theoretical phase **splits** into a rich sequence of detectable merger events.
>
> ---
>
> ### **Visualization of the merger process**
>
> To support intuition, we include a visual hierarchy of the merger process:
>
> (**Click for plot**) [Merger visualization](https://postimg.cc/N9hRsXdp)
>
> This diagram shows how initially disjoint events merge over time during diffusion. Merger points indicate when discriminative information vanishes. The resulting tree structure reveals the **natural statistical hierarchy** among events in the dataset.
>
> ---
>
> ### **Algorithm clarification**
>
> The reviewer raised a concern regarding simultaneous sampling with two models. This reflects a common misunderstanding. Our approach:
>
> - **Uses a single forward diffusion model**
> - Tracks how **different data subsets** (e.g., classes) evolve under this process
> - Requires no separate models or retraining
>
> **Cross-fluctuation computation** compares the evolved states of different data subsets. When their statistics converge (i.e., similarity approaches 1), we label the groups as merged.
>
> **Practical implementation**: Algorithm 1 only uses **forward passes** through the training data. It involves no reverse sampling, retraining, or hyperparameter search.
>
> Theoretical support for this setup is given in Theorem 9 (Appendix B.4), under standard regularity conditions.
>
> ---
>
> ### **Methodology details**
>
> - **Measurement process**:
>   We define cross-fluctuations $\mathcal{M}^{(n)}_{\rho}(\Omega_1, \Omega_2) \in [0,1]$, where values near 1 indicate statistical indistinguishability.
>
> - **Detection criterion**:
>   We declare a **merger transition** when the expected moment structures of two groups converge within a threshold $\epsilon$. Sharp rises in $\mathcal{M}^{(n)}_{\rho}$ toward 1 mark transition points. These transitions serve as **precise markers** for optimal interventions like early stopping or guided sampling—**without** any grid search.
>
> - **Computational efficiency**:
>   Our method is highly efficient. It relies only on **forward diffusion** and **covariance estimation**. The latter can be scaled using block-diagonal approximations or random projections (Appendix D), without compromising detection quality.
>
> ---
>
> ### **Addressing remarks on accessibility**
>
> We agree that the current presentation could better serve a broader audience. Based on the reviewer's feedback, we will implement the following changes:
>
> - **Structural reorganization**:
>   We will lead with intuitive overviews and visualizations before introducing mathematical details.
>
> - **Concrete examples**:
>   Section 2.2 will be expanded to include simple, visualizable use cases that show cross-fluctuation dynamics in action.
>
> - **Appendix relocation**:
>   Technical derivations will be moved to appendices to declutter the main text while preserving theoretical completeness.
>
> - **Practical emphasis**:
>   We will highlight the method as a plug-in toolkit (Section 3), usable without deep theoretical background.
>
> ---
>
> ### **Experimental validation**
>
> **Performance**:
> As noted by the reviewer, our experiments show significant improvements in FID and inference efficiency. Our approach yields **orders-of-magnitude gains** over grid search-based methods.
>
> **Generality**:
> We validate across **five use cases**: early stopping, class conditioning, rare class generation, classification, and style transfer. This underscores the broad applicability of our phase transition framework.
>
> ---
>
> ### **Summary**
>
> We sincerely thank the reviewer for their constructive feedback. We fully agree that presenting our framework more intuitively will broaden its impact without compromising rigor.
> **We are confident that these improvements will significantly enhance clarity, accessibility, and practical utility of the revised submission.**

---

> > ### Comment · Reviewer_yAPh · 2025-08-05
> >
> > I would like to thank the authors for the effort they have put in the rebuttal. In light of their response, some parts are more clear. This highlights however that the core initial submitted papers needed further clarification (in terms of definitions of key concepts and visual and intuitive overview). With a few changes, also proposed by the authors, the paper could be accessible to a broader audience making it more impactful in the long term.
> >
> > In the good faith that the authors will make these changes, I’m going to raise my score.

---

### Official Review · Reviewer_fbWE · 2025-07-01

**Clarity:** 2
**Significance:** 3
**Originality:** 3
**Rating:** 4
**Confidence:** 2

**Summary:**

The paper analyzes sampling dynamics in score-based diffusion models through cross-fluctuations, an nth-order central moment that measures when two sets of trajectories become indistinguishable under the forward noise process. It shows the connection between phase transitions with the trajectory  discontinuities of this statistic. For variance-preserving SDEs, the authors derive a closed-form expression that can be evaluated without additional simulations. These theoretical results lead to several methods that can detect merger points and safely skip the earliest reverse steps, yielding faster sampling and other guidance tricks without retraining.

**Questions:**

1. The Main Contributions section states that this work establishes "a connection between ensemble based techniques in statistical mechanics as well as classical mixing-coupling results for Markov chains, using a generalized notion of phase transitions". Could you provide a high-level explanation of this result, and comment on how novel or significant it is within the statistics literature?
2. Typo: line 292 reads "methodlologies" but should be "methodologies".

**Ethical Concerns:**

["NO or VERY MINOR ethics concerns only"]

**Final Justification:**

I thank the authors for their detailed response that address all my questions. I will maintain my positive score.

**Limitations:**

yes

**Quality:**

3

**Strengths And Weaknesses:**

Strengths:
1. The paper introduces a fresh mathematical viewpoint for studying diffusion models by bringing tools from fluctuation theory into the analysis.
2. The study is thorough, moving from formal results to practical tests, and the theory-driven algorithm shows promising empirical gains.

Weaknesses:
My main concern is presentation. To me, the key contribution is the new theoretical lens, while the experiments mostly serve as illustrations. The paper does not present clearly which theoretical results are formally proved and how to read them intuitively. It would help to make it explicit which parts are provable theory and which parts are theory-inspired heuristics.

---

> ### Author Rebuttal · Authors · 2025-07-31
>
> We are sincerely grateful to the reviewer for highlighting our work’s “fresh mathematical viewpoint” and for offering such constructive suggestions to improve the clarity of our presentation.  We have addressed the reviewer's comments point by point below.
>
>
> ### **Mathematical context and distinguishing formal proofs from heuristics**
>
> **Mathematical foundation**: Cross-fluctuations are nth-order centered moment statistics computed over trajectory ensembles within event sets (partitions of the data space). The framework measures structural similarity between events through their moment interactions. We track the nromalized cross-fluctuation $\mathcal{M}^{(n)}_{\rho}(\Omega_1, \Omega_2) \in [0,1]$ with values near $1$ indicating statistical indistinguishability.
>
> **Formally proven results**:
> - **Theorem 12** (Appendix B.7): Moment-TV inequality establishing that fluctuation moment proximity bounds total variation distance between probability distributions under regularity conditions.
> - **Theorem 10** (Appendix B.5): Topological equivalence between fluctuation-based distance d_M and Frobenius distance d_F on expected fluctuation tensors.
> - **Theorem 11** (Appendix B.5): Diffusion time geometry theorem proving that cross-fluctuation merger sequences determine the metric structure of evolving marginals, with extensions to hierarchical partitions.
>
> **Theoretically inspired heuristics**:
> - The criterion for $\epsilon$ parameter selection (validated through extensive experimentation in Appendix C rather than theoretical derivation).
> - Algorithm 1 implementation details for practical merger detection.
> - Application-specific guidance window calculations (detailed in Appendix C).
>
> ### **More details on the theoretically inspired selection of $\epsilon$**
>
> #### Quantifying merger events - Phase spectrum analysis
>
> (**Click for plot**) [Merger phase diagram](https://postimg.cc/3kQMG8tN))
>
> This plot reveals the fundamental phase structure governing $\epsilon$ selection across four distinct operational regions. The optimal region enables detection of the maximum number of distinguishable mergers ($1 < K \leq N-1$ for $N$ classes), while regions with $\epsilon$ values too low or too high either are impossible due to a lack of numerical precision or fail to detect class distinctions. Our $\lambda_{max}/400$ criterion operates within this optimal detection region as the merger probability for standard datasets is of the order $10^{-2}$ with this criterion.
>
>
> #### Merger feasibility frontiers - Theoretical trade-offs
>
> (**Click for plot**) [Merger feasibility frontiers](https://postimg.cc/Mctggyqf))
>
> This analysis demonstrates the theoretical trade-off between merger time $t^{\star}$ and $\epsilon$ across different fluctuation orders ($n = 2,3,4,5$). Higher-order fluctuations expand the feasible parameter space, allowing merger detection across broader $\epsilon$ ranges at lower timesteps while collapsing faster due to non-linear decay induced by the diffusion process. The plot confirms that second-order fluctuations provide stable discrimination for practical applications.
>
>
> ### **Connection to statistical mechanics and Markov chains**
>
> **Mathematical framework**: Our approach addresses fundamental limitations in extending coupling theory to continuous state spaces. In discrete Markov chain theory, the coupling time $t_{cpl}(\epsilon) = \min\{t \geq 0 : \max_{\alpha,\beta} d_{TV}(S_{\alpha,t}, S_{\beta,t}) \leq  \epsilon\}$ measures when chains ($S_{\alpha,t},S_{\beta,t}$) starting from different initial distributions become indistinguishable. However, total variation distance lacks discriminative power in continuous spaces, where it often equals $1$ for distinct absolutely continuous distributions.
>
> **Generalized coupling framework**: We establish a moment-based generalization through our cross-fluctuation statistic. As detailed in Appendix B.6, we may define the generalized coupling time $t_{gen}(ε) = \min\{t \geq 0 : \max_{\alpha,\beta} \mathcal{M}^{(n)}_\rho(\Omega_{\alpha,t}, \Omega_{\beta,t}) \geq  1-\epsilon\}$, which coincides with classical coupling time for discrete state spaces while providing meaningful discrimination in continuous settings.
>
> **Theoretical bridge**: Theorem 12, Appendix B.7 establishes that moment proximity bounds total variation distance under regularity conditions. This enables practical merger detection in high-dimensional continuous spaces where direct TV computation is intractable.
>
> **Statistical physics connection**: The discrete phase transitions we identify in cross-fluctuations connect to ensemble theory in statistical mechanics (Appendix B.14). We distinguish thermodynamic transitions (discontinuities in classical derivatives) from lattice transitions (finite-difference discontinuities with threshold $\tau$), where our merger events represent lattice transitions in the moment space.
>
> **Significance**: This work provides a computational framework for extending coupling concepts to modern generative models operating in high-dimensional continuous spaces, where traditional TV-based approaches face fundamental mathematical and computational challenges.
>
> ### **Organizational improvements**
>
> We acknowledge the presentation concerns and will implement several organizational improvements based on reviewer feedback. These include restructuring the main results into a dedicated theorem section, consolidating theoretical foundations in reorganized appendices, and providing clearer delineation between proven results and practical heuristics. These changes will enhance accessibility while maintaining mathematical rigor.
>
> ### **Other Improvements**
>
> We acknowledge the typo on line 292 ("methodlologies" $\rightarrow$ "methodologies") and will correct this in the final version, along with additional proofreads to catch other related typos.
>
> ### **Summary**
>
> Our work establishes theoretical connections between discrete Markov chain coupling concepts and continuous diffusion dynamics through fluctuation-based phase transitions. The formal theorems in Appendix B provide mathematical foundations that enable practical merger detection in high-dimensional continuous spaces. While parameter selection, such as $\epsilon$, represents an empirically validated approach rather than a theoretical derivation, the underlying mathematical framework provides principled grounding for controlling diffusion sampling dynamics across multiple applications.
>
> We sincerely thank the reviewer for their thoughtful and constructive feedback. **We are confident that the reviewer’s suggestions can be seamlessly integrated into the revised script**.

---

> > ### Comment · Reviewer_fbWE · 2025-08-04
> >
> > I thank the authors for their response and maintain my positive score.
> >
> > That said, I would like to note that including URL links in the rebuttal is not permitted.

---

### Official Review · Reviewer_Z71p · 2025-07-02

**Clarity:** 2
**Significance:** 3
**Originality:** 3
**Rating:** 4
**Confidence:** 2

**Summary:**

This paper introduces a novel framework for controlling sampling in diffusion models through the lens of fluctuation theory. The central insight is that during forward diffusion, different classes (data events) exhibit distinct fluctuation behaviors, which eventually converge (or "merge") as the process approaches pure noise. The authors define a normalized cross-fluctuation metric   to quantify distinguishability between events and use it to detect phase transitions — points at which classes become statistically indistinguishable.
They apply this framework to: accelerated sampling via early stopping, improved class-conditional guidance, enhanced rare-class generation, zero-shot classification using fluctuation signatures.

**Questions:**

Sec 4.2: to what extent are the results sensitive to the threshold epsilon? It would have been interesting to give the results for various epsilons.

Sec 4.3: similar question: I understand that the criterion is whether the rare class has merged with any other class. Would the results be sensitive to a change of this criterion, e.g. the merging of the rare class with m classes with m>1?

Sec 4.4: would this method generalize to hierarchical or fine-grained classification settings (e.g., sub-breeds of dogs, rather than mnist classes)? How scalable is the covariance-based comparison in high-dimensional spaces?

**Ethical Concerns:**

["NO or VERY MINOR ethics concerns only"]

**Final Justification:**

kept my score

**Limitations:**

The authors did not really discuss the limitations of this work.

**Quality:**

2

**Strengths And Weaknesses:**

Strengths

- originality: The use of fluctuation metrics as a lens to analyze diffusion dynamics is novel. Rather than relying on learned metrics or external classifiers, the method uses intrinsic model statistics.
- applicability: The paper provides a practical algorithm (Algorithm 1). The method is evaluated across diverse settings: class-conditional sampling, rare class recovery, long-horizon compositional prompts, and even zero-shot classification.
- significance: This approach has wide implications; among which acceleration of diffusion, improvement of rare-class generation, guiding conditioning more precisely, and reveal structure without training.
- clarity (mostly): The paper is well-organized and mostly clear. That said, some parts, especially the explanation of how fluctuation metrics translate to guidance windows — require careful reading or visual aids.

Weaknesses

- clarity: The concept of fluctuations, merger times, and their link to class indistinguishability is dense and may be hard to parse without prior familiarity. The paper would benefit from intuitive visualizations or simplified examples early on.
- Metrics/ interpretation of the numerical results for some applications: While precision/recall/coverage are well-adapted for generative models, the way numerical results illustrate the claims of the authors is not always clear. For instance, it is not clear how they indicate a greater rare-class coverage. the paper could be clearer about how this, especially since it doesn’t directly report class-wise sample counts.
- Uncertainty in Merging Behavior: The notion that class k's merger time is based on indistinguishability from any other class (rather than all classes) is conservative but may underestimate useful guidance intervals. Discussion of this tradeoff is absent.

---

> ### Author Rebuttal · Authors · 2025-07-31
>
> We thank the reviewer for acknowledging the originality and significance of our work, and for pointing out where the exposition and clarity should be improved. We respond to the points raised below, with the key aim of suggesting what we are doing to improve the clarity and accessibility of the work.
>
> ### Addressing concerns on clarity
>
> We acknowledge that the mathematical treatment of fluctuation theory may be dense. To provide an intuitive understanding, we present a visualization capturing what we are measuring: a hierarchy of mergers that reveals how members of a dataset partition relate to each other.
>
> (**Click for plot**) [Merger visualization](https://postimg.cc/N9hRsXdp)
>
> This temporal hierarchy illustrates how distinct, mutually disjoint events merge as the diffusion process evolves, with merger points indicating when discriminating information vanishes.
>
> ## Response to specific questions about §§4.2, 4.3, and 4.4
>
> ### **Section 4.2: Sensitivity to epsilon threshold**
>
> **Selection criterion**: We establish $\epsilon = \frac{\lambda_{max}}{400}$, where $\lambda_{max}$ represents the largest principal eigenvalue magnitude across class covariance matrices. This provides a principled, data-driven selection criterion that adapts automatically to dataset characteristics.
>
> **Empirical validation**: As detailed in Appendix C.3, Table 9 demonstrates the relationship between $\epsilon$ values, merger probabilities (probabilities of two random classes being merged at $t = 0$ with this $\epsilon$), and associated FID performance:
>
> | $\epsilon$ | Merge probability (t=0) | FID (↓) |
> |---|-------------------------|---------|
> | 200 | 0.027 | 3.06±0.19 |
> | 100 | 0.013 | 2.86±0.15 |
> | 80 | 0.010 | 2.92±0.17 |
> | 67 | 0.009 | 2.90±0.11 |
> | 20 | 0.002 | 2.88±0.14 |
>
> **Merger probability analysis**: Figures 7,12 (Appendix C.3) present curves showing how the fraction of merged classes evolves over time t for different ε values across our choice of datasets. These curves demonstrate that our criterion yields stable merger dynamics with merger probabilities at $t=0$ being of the order $10^{-2}$. We also present a sample curve for the Imagenet case (from Figure 7, Appendix C.3)
>
> (**Click for plot**) [Merger probability curves](https://postimg.cc/0z0pkjTs)
>
> ### **Additional theoretical analysis**
>
>
> **Quantifying merger events - Phase spectrum analysis**
>
> (**Click for plot**) [Merger phase diagram](https://postimg.cc/3kQMG8tN)
>
> This plot reveals the fundamental phase structure governing $\epsilon$ selection across four distinct operational regions. The optimal region enables detection of the maximum number of distinguishable mergers ($1 < K \leq N-1$ for $N$ classes), while regions with $\epsilon$ values too low or too high either are impossible due to a lack of numerical precision or fail to detect class distinctions. Our $\lambda_{max}/400$ criterion operates within this optimal detection region as the merger probability for standard datasets is of the order $10^{-2}$ with this criterion.
>
>  **Merger feasibility frontiers - Theoretical trade-offs**
>
> (**Click for plot**) [Merger feasibility frontiers](https://postimg.cc/Mctggyqf)
>
> This analysis demonstrates the theoretical trade-off between merger time $t^{\star}$ and $\epsilon$ across different fluctuation orders ($n = 2,3,4,5$). Higher-order fluctuations expand the feasible parameter space, allowing merger detection across broader $\epsilon$ ranges at lower timesteps while collapsing faster due to non-linear decay induced by the diffusion process. The plot confirms that second-order fluctuations provide stable discrimination for practical applications.
>
> ### **Section 4.3: Merger criterion sensitivity**
>
> We appreciate the reviewer’s suggestion to explore the sensitivity of our merger criterion—namely, changing from merging as soon as any single class becomes indistinguishable to requiring merger with $m>1$ classes. Our conservative choice (merging upon indistinguishability with any other class) identifies the earliest point at which statistical separation is lost, and thus defines the most appropriate guidance‐window boundary. While one could certainly extend the windows by insisting on merger across multiple classes, doing so carries the risk of operating beyond the region in which class distinctions remain meaningful.
>
> The reviewer asks about sensitivity to changing the merger criterion from merging with any single class to merging with $m>1$ classes. Our conservative approach (merger with any other class) captures the earliest point of statistical indistinguishability, which represents the optimal guidance window boundary. Relaxing this criterion to require merger with multiple classes would extend guidance windows, but risks operating beyond meaningful class distinction.
>
> We provide empirical validation for this design choice in Appendix C.6, where Table 11 and Figure 15 demonstrate that linear separability between randomly chosen ImageNet class pairs increases monotonically up to the merger time. This confirms that the binary merger criterion identifies the point of maximum statistical distinguishability. We attach a sample plot from Figure 15 for reference.
>
> (**Click for plot**) [Linear separability](https://postimg.cc/2qqQhZ2N)
>
> ### **Section 4.4: Hierarchical classification and scalability**
>
> **Generalization to higher hierarchies**: Our mathematical framework assumes only a partition of distinct events in the data space, enabling generalization to finer hierarchies. We refer the reviewer to Theorem 11 in Appendix B.5, which formalizes that partitions can be refined to greater detail down to singletons with well-defined mergers.
>
> **Scalability**: The covariance-based comparison scales as $O(Kd^{2})$ for $K$ classes and $d$ dimensions. For high-dimensional spaces, this can be mitigated through random projections or block diagonal approximations without degrading detection power, as discussed in Appendix D.
>
> ### **Metrics interpretation and rare-class coverage**
>
> **Standardness of metrics**: We employ precision, recall, density, and coverage as standard metrics for rare class synthesis, consistent with established evaluation protocols for long-tailed datasets. CLIP similarity provides additional validation for semantic quality.
>
> **Rare-class coverage**: The numerical results demonstrate improved rare-class coverage through higher recall and coverage metrics. Our method ensures rare classes receive appropriate guidance windows before merging with dominant classes, preventing their collapse during generation.
>
> ### **Limitations**
>
> We acknowledge several limitations of our approach. Memory overhead scales as $O(Kd^{2})$ for computing class covariances, though this can be mitigated through approximation techniques. The method has been validated primarily on 2D images and the standard VP schedule, and extension to other modalities and different noise schedules requires further investigation.
>
> ### **Summary**
> To summarize: our criterion provides principled parameter selection with demonstrated stability across the tested range. The phase structure analysis reveals the theoretical foundations underlying this robustness, while our mathematical framework naturally extends to hierarchical classifications. These contributions establish a theoretically grounded and practically reliable approach for controlling diffusion sampling dynamics.
>
> We trust that the new experiments and analyses effectively address the reviewer’s concerns regarding parameter sensitivity, criterion robustness, and scalability. **We sincerely appreciate the insightful suggestions and will incorporate them fully in the revised manuscript**. Should any questions or additional concerns remain, we would be delighted to discuss and resolve them. Thank you very much for your time and thoughtful review.

---

> ### Comment · Reviewer_Z71p · 2025-08-01
> **Comments**
>
> Thank you for your answer.
>
> I appreciate the additional experiments and visualizations.
>
> Yet I would temper the use of vocabulary "principled" (for epsilon - I generally understand "principled" as theoretically justified) and section "theoretical analysis" (for phase diagrams that are not supported by strong theoretical results), because neither the manuscript nor the rebuttal provide formal theoretical results or derivations to derive these. I would suggest the authors to be careful with their presentation of their results in the revised version if the paper is accepted as this vocabulary can be misleading.
>
> I keep my positive opinion on the paper and maintain my score.

---

> > ### Author Response · Authors · 2025-08-05
> > **Thanks**
> >
> > Thanks!
> >
> > We agree with the modulation of the term "principled" and will adjust our description accordingly.

---

### Official Review · Reviewer_Sxsz · 2025-07-02

**Clarity:** 3
**Significance:** 3
**Originality:** 3
**Rating:** 4
**Confidence:** 3

**Summary:**

This paper presents a novel framework for understanding the sampling dynamics of diffusion models using cross-fluctuation, a concept from statistical physics. The authors propose a useful operator to capture the discrete transition during the sampling of diffusion models and demonstrate the effectiveness of the proposed method on five applications, including early stopped sampling, CFG sampling, rare-class generation, classification, and style transfer.

**Questions:**

I don't have further questions.

**Ethical Concerns:**

["NO or VERY MINOR ethics concerns only"]

**Limitations:**

Yes

**Paper Formatting Concerns:**

I don't find any major formatting issues in this paper.

**Quality:**

3

**Strengths And Weaknesses:**

Strengths:
- The paper is well-organized, and the application of the concept of cross-fluctuation on diffusion models is novel.
- The proposed method provides a practical criterion, which can be computed with only the forward process, to capture the discrete phase transition in the sampling dynamics of diffusion models.
- Such a criterion is shown to be practically useful in various scenarios.

Weaknesses:
- The most crucial hyperparameter, the $\epsilon$ in Equation (3.1), requires more detailed illustration. The choice of $\epsilon$ determines the performance of the proposed method in all the included application scenarios. However, the criterion for choosing a proper $\epsilon$ seems to lack discussion, except for class-conditional sampling, where a criterion based on the largest eigenvalue is proposed. If a proper $\epsilon$ is sensitive to the dataset or task, it will still take a source-extensive sweep search to find the optimal one.

---

> ### Author Rebuttal · Authors · 2025-07-31
>
> We thank the reviewer for acknowledging the novelty of our cross-fluctuation framework and its practical efficiency across multiple applications. To address the remark regarding the selection of $\epsilon$, we consolidate and expand our sensitivity analysis as follows:
>
> (1) we recall our rule for selecting $\epsilon$ without costly grid search (Appendix C.3);
> (2) we reiterate how this rule governs the probability of full dataset merging at each step (Appendix C.3);
> (3) we introduce a new phase diagram illustrating how varying $\epsilon$ leads to distinct qualitative behaviors, spanning the spectrum from $\epsilon = 0$ (infinite precision) to $\epsilon \to \infty$ (zero precision);
> (4) we further extend the analysis by showing how admissible $\epsilon$ ranges shift under higher-order fluctuations beyond the second-order (covariance) case.
>
> ### **(1) Principled selection of $\epsilon$.**
>
> **Selection criterion**:
> Instead of running an expensive grid search, we use a simple rule of thumb: set $\epsilon = \frac{\lambda_{\max}}{400}$, where $\lambda_{\max}$ is the largest eigenvalue magnitude among class covariance matrices. We arrived at this choice by inspecting our datasets and plotting the CDF of pairwise **differences** between the top eigenvalues of unperturbed class covariances. We then chose $\epsilon$ so that $P(\mathrm{difference} \leq \epsilon)$ is small typically around $10^{-2}$. This helps avoid trivial mergers between classes at $t = 0$. Figure 7a in Appendix C.3 shows a representative CDF for ImageNet that illustrates this behavior.
>
>
> [Difference between eigenvalues for Imagenet classes](https://postimg.cc/qznfZp35)
>
>
> **Empirical validation**: (please also see Appendix C.3, Table 9). We demonstrate the relationship between $\epsilon$ values, merger probabilities (probabilities of two random classes being merged at $t = 0$ with this $\epsilon$), and the associated FID.
>
> | $\epsilon$ | Merge probability (t=0) | FID (↓) |
> |---|-------------------------|---------|
> | 200 | 0.027 | 3.06±0.19 |
> | 100 | 0.013 | 2.86±0.15 |
> | 80 | 0.010 | 2.92±0.17 |
> | 67 | 0.009 | 2.90±0.11 |
> | 20 | 0.002 | 2.88±0.14 |
>
>
> ### **2) Merger probability analysis.**
>
> Figures 7b,7d, and 12 b, 12d (Appendix C.3) present curves showing how the fraction of merged classes evolves over time $t$ for different values of $\epsilon$ across our choice of datasets. These curves demonstrate that our criterion yields stable merger dynamics with merger probabilities at $t=0$ being of the order $10^{-2}$. We also present a sample curve for the Imagenet case (from Figure 7b, Appendix C.3)
>
> (**Click for plot**) [Merger probability](https://postimg.cc/0z0pkjTs)
>
>
> ### **(3) Phase structure of $\epsilon$ selection.**
>
> To further substantiate the choice of \$\epsilon\$, we have carried out additional theoretical analyses to explore the phase structure underlying the merger dynamics:
>
> (**Click for plot**) [Merger phase diagram](https://postimg.cc/3kQMG8tN)
>
> The plot above reveals the fundamental phase structure governing $\epsilon$ selection across four distinct operational regions. The optimal region enables detection of the maximum number of distinguishable mergers ($1 < K \leq N-1$ for $N$ classes), while regions with $\epsilon$ values too low or too high either are impossible due to a lack of numerical precision or fail to detect class distinctions. Our $\lambda_{\max}/400$ criterion operates within this optimal detection region as the merger probability for standard datasets is of the order $10^{-2}$ with this criterion.
>
> ### **(4) Merger feasibility frontiers - dependency on order of fluctuation.**
>
> (**Click for plot**) [Merger feasibility frontiers](https://postimg.cc/Mctggyqf)
>
> This analysis demonstrates the theoretical trade-off between merger time $t^{\star}$ and $\epsilon$ across different fluctuation orders ($n = 2,3,4,5$). Higher-order fluctuations expand the feasible parameter space, allowing merger detection across broader $\epsilon$ ranges at lower timesteps while collapsing faster due to non-linear decay induced by the diffusion process. The plot confirms that second-order fluctuations provide stable discrimination for practical applications.
>
>
> ### **Summary**
>
> We hope the above discussion addresses the reviewer's concerns, and we would be happy to engage in any discussion that may be helpful in further clarification. **We will incorporate the reviewer suggestions fully in the revised manuscript**.

---

### Decision · Program_Chairs · 2025-09-17

**Decision:**

Accept (poster)

**Comment:**

* Summery

The authors introduce tools from statistical physics to analyse the sampling dynamics of diffusion models and their underlying dynamics.
Using fluctuation theory and fluctuation dispersion style tools, they can separate and distinguish between different regions of events and define the transition between such regions as a phase transition. The method is then explored, both through theoretical arguments and insights, as well as empirical experiments. All of the above is leveraged to improve accelerated sampling through early stopping, enhancing class-conditional guidance, and class generation, and performing zero-shot classification.

* Strengths and reasons for acceptance

The idea is novel and, in a sense, builds upon the original diffusion models presented in "Deep Unsupervised Learning using Nonequilibrium Thermodynamics." Since this kind of fluctuation theory is what ties equilibrium and [non-equilibrium processes](https://www.pnas.org/doi/10.1073/pnas.1918386117), the paper is very well written and puts a major effort into being self-contained. While theoretical in nature, the authors also provide a practical algorithm for implementation and demonstrate it across several tasks related to training and using diffusion models. In addition to the above, this is one of the first methods to examine the model data independently in this manner, without requiring additional training or external classifiers.
With all of the above, I would recommend this paper for acceptance into the conference.

* Weaknesses
The paper is quite dense, and discerning the originality of ideas can be challenging; more specifically, the clarity of the core concepts is difficult to grasp. The biggest point of weakness and issue with the paper we should be critical of is the hyperparameters, and specifically the $\epsilon$ parameter. I would have been happier with a bit more insight into it and guidance on how to choose it.